# Possible biases in scaling-based estimates of glacier change: A case study in the Himalaya

Argha Banerjee[1], Disha Patil[1], and Ajinkya Jadhav[1]

[1]ECS, IISER Pune, India

**Correspondence:** Argha Banerjee (argha@iiserpune.ac.in)

**Abstract.** Approximate glacier models are routinely used to compute the future evolution of mountain glaciers under any given climate-change scenario. A majority of these models are based on statistical scaling relations between glacier volume, area, and/or length. In this paper, long-term predictions from a scaling-based models are compared with those from a two-dimensional shallow-ice approximation (SIA) model. We derive expressions for climate sensitivity and response time of glaciers assuming a time-independent volume-area scaling. These expressions are validated using a scaling-model simulation of the response of 703 synthetic glaciers from the central Himalaya to a step-change in climate. The same experiment repeated with the SIA model, yields about two times larger climate sensitivity and response time than those predicted by the scaling model. In addition, the SIA model obtains area response time that is about 1.5 times larger than the corresponding volume response time, whereas scaling models implicitly assume the two response times to be equal to each other. These results indicate the possibility of a low bias in the scaling model estimates of the long-term loss of glacier area and volume. The SIA model outputs are used to obtain parameterisations climate sensitivity and response time of glaciers as functions of ablation rate near the terminus, mass-balance gradient, and mean thickness. Using a linear-response model based on these parameterisations, we find that the linear-response model outperforms the scaling model in reproducing the glacier response simulated by the SIA model. This linear-response model may be useful for predicting the evolution of mountain glaciers on a global scale.

## 1 Introduction

In the coming decades, shrinking mountain glaciers will contribute significantly to global eustatic sea-level rise (e.g., Radić et al., 2014; Hock et al, 2019; Marzeion et al., 2020), and impact the hydrology of glacierised basins worldwide (e.g., Huss and Hock, 2018; Immerzeel et al., 2020). The reliability of the predicted changes in global sea-level and those in the regional hydrology of various river basins is, thus, intimately tied to the accuracy of the predicted total ice-loss from mountain glaciers for any given climate scenario.

Instantaneous (annual) glacier surface mass balance can be calculated readily using climate model outputs. In contrast, any prediction of the long-term evolution of a glacier requires simulating the slow (decadal) changes in glacier area and geometry. Ideally, this is to be done by solving the dynamical ice-flow equations (e.g., Hutter, 1983). However, the numerical cost of such a computation on a global scale is high, even if simplified approximate descriptions of the ice-flow equations, like, shallow-ice approximation (SIA) (Hutter, 1983) or its higher order variants were to be used (Egholm et al., 2011; Clarke et al., 2015). One-

dimensional SIA-based modelling tools are promising developments in this regard (Maussion et al., 2019; Zekollari et al., 2019; Rounce et al., 2020). The uncertainties associated with various input parameters, e.g., an uncertain glacier bedrock, limit the benefit of using the physically-based ice-flow models (Farinotti et al., 2016). Consequently, a majority of the recent estimates of the global to regional scale evolution of mountain glaciers relies on low-dimensional approximate parameterisations of glacier dynamics (e.g., Radić et al., 2014). The results from these simplified models have provided critical inputs for multimodel ensemble-averaged estimates of future sea-level rise (Hock et al, 2019; Marzeion et al., 2020), assessments of regional to global vulnerability to sea-level rise (e.g., Kulp and Strauss, 2019), and understanding the co-evolution of glaciers, river runoff and climate in glacierised regions like High Mountain Asia (e.g., Zhao et al., 2014; Zhang et al., 2015; Kumar et al., 2019).

While some of the approximate parameterisations of glacier dynamics are empirical prescriptions for adjusting the hypsometry of the transient glaciers (Raper and Braithwaite, 2006; Huss et al., 2010; Huss and Hock, 2015), a majority of them are primarily based on a statistical volume-area (or volume-area-length) scaling relation. This volume-area scaling equation relates glacier volume $V$ to glacier area $A$ as,

$$V = cA^\gamma, \tag{1}$$

where $\gamma$ is a dimensionless scaling exponent, and $c$ is a scale factor (Bahr et al., 2015). This relation was established empirically (e.g., Chen and Ohmura, 1990), and subsequently proved using dimensional analysis (Bahr et al., 1997, 2015). The derivation utilised the empirical sub-linear scaling of glacier width and ablation rate with the glacier length (Bahr, 1997).

Theoretically, the scaling exponent $\gamma$ is time-independent, and can be expressed as $\gamma = 1 + \frac{m+1}{m+n+3}$ (Bahr et al., 2015). Here, $n$ is the power-law exponent of Glen's rheology of ice (Glen, 1955), and $m$ is the scaling exponent of ablation rate with glacier length (Bahr, 1997). For an individual glacier, the scale-factor $c$ captures the control of all the glacier-specific factors (except area) on its volume (Bahr et al., 2015). There is no available theoretical prescription for obtaining the value of $c$ for an arbitrary glacier. $c$ may be calibrated for a particular glacier based on available independent measurements of area and volume over an epoch, but its time dependence can be accessed only with a detailed model simulation (Bahr et al., 2015). For a large enough ensemble, glacier area typically spans a few orders of magnitude. However, the corresponding $c$ values vary over a relatively restricted range (Bahr et al., 2015). This allows an approximate statistical description of any set of glaciers using eq. 1, where a single best-fit $c$ and a fixed $\gamma$ is used (Bahr et al., 2015). Such a best-fit scaling relation provides a fairly accurate estimate of the total ice volume of a large set of glaciers, but the corresponding predictions for the individual glaciers have relatively large uncertainties (Bahr et al., 2015). Note that there is no theoretical constraint for $c$ to be time-independent for a given set of non-steady glaciers (Bahr et al., 2015).

It is the above statistical interpretation of the scaling relation, where a best-fit time-invariant $c$ and a constant $\gamma$ is used to describe an ensemble of glaciers, that is exploited in the scaling-based approximate models of glacier dynamics (e.g., Radić et al., 2007). Hereinafter, we refer to the models based on such an approach (e.g., Radić et al., 2007), as "scaling models". As the present study investigates the possibility of biases in scaling model predictions of glacier evolution, we restrict ourselves to the above statistical interpretation of the scaling relation.

The performance of scaling models in simulating the transient glacier response have previously been tested against various dynamical ice-flow models (e.g., SIA, higher order approximations, or Stokes' model) in one to three dimensions using both idealised (Radić et al., 2007; Adhikari and Marshall, 2012) and realistic geometries (Radić et al., 2008; Farinotti and Huss, 2013). The uncertainties introduced by a scaling-model parameterisation of the evolution of glaciers with realistic geometries were considered by Farinotti and Huss (2013). The spirit of the present study is quite similar to that of Farinotti and Huss (2013), except that we are investigating the possible intrinsic biases of scaling models in a situation where the parameters ($c$ and $\gamma$) are known accurately. The specific objectives of the present study are,

1. To obtain analytical predictions for climate sensitivity and response time of glaciers in a scaling model.

2. To compare the climate sensitivity and response time of a large number of synthetic glaciers with realistic geometries, as obtained from a scaling model and a 2-d SIA model.

3. To investigate the possibility of long-term biases in scaling model estimates of changes in glacier area and volume with respect to corresponding SIA results.

4. To find convenient parameterisations of glacier response properties obtained from the SIA simulations, and develop an accurate linear-response model.

Note that a linear-response model introduced in the last objective is a low-complexity model obtained in the limit of a relatively small deviation around a steady state (e.g., Oerlemans, 2001). To apply this model to a large number of glaciers, the response time and climate sensitivity need to be specified for each of them. A lack of accurate and numerically-convenient parameterisations of these dynamical properties may have limited its application (Harrison et al., 2001; Lüthi, 2009; Bach et al., 2018). Here, we aim to obtain parameterisations of the glacier response properties as functions of a few easily accessible properties of the glaciers, using results from 2-d SIA simulations of a large ensemble of synthetic glaciers with realistic geometries.

The paper is organised as follows. First, we theoretically derive the glacier-response properties within a time-invariant scaling assumption (sect. 2.1 and 3.1). Then, we compare the performance of a representative scaling model (Radić et al., 2007) with that of a 2-dimensional SIA model, in simulating the response of 703 idealised Himalayan glaciers in the Ganga basin to a hypothetical step rise in equilibrium line altitude (ELA) (sect. 2.2 and 3.2). We use the response properties obtained from the scaling model to test the above analytical expressions for glaciers-response properties. The corresponding SIA results are used to obtain parameterisations for the linear-response properties of realistic glaciers. The accuracy of the scaling model and a linear-response model in reproducing the SIA-derived long-term loss of total glacier area and volume is assessed for the above 703 glaciers. The performance of the linear-response model is also tested for an independent set of 164 glaciers in the western Himalaya without any further calibration. We also discuss the applicability of the linear-response model for actual computation of future glacier loss for a set of transient glaciers forced by any arbitrary time-variation ELA (sect. 3.3).

## 2  Methods

### 2.1  Theoretical methods

For a theoretical analysis of the glacier-response properties implied by a scaling model, we consider a set of hypothetical glaciers that respond to a warming climate such that the volume-area scaling relation (eq. 1) is valid, and $c$ is a given time-invariant constant. Then, the fractional changes in area and volume of these glaciers, in the limit of small changes, are related as follows.

$$\Delta V \approx c\gamma A^{\gamma-1}\Delta A = \gamma\frac{V}{A}\Delta A = \gamma h\Delta A, \tag{2}$$

where $\Delta V$ and $\Delta A$ are the changes in area and volume, and the mean ice thickness is $h = V/A$. The above equation is the basis of the scaling models of glacier evolution (e.g., Radić et al., 2007). We have derived analytical expressions for glacier response time and climate sensitivity starting from this equation, essentially following the line of arguments by Harrison et al. (2001).

### 2.2  Numerical methods

We simulated the response of an ensemble of synthetic clean glaciers with realistic geometries to a hypothetical step-change in ELA using three different methods (scaling, SIA, and linear-response models). For this exercise, we considered all the 814 glaciers larger than 2 km$^2$ in the Ganga basin, the central Himalaya (Supplementary fig. S1). The ice-free bedrock for each glacier was obtained using available ice-thickness estimates (Kraaijenbrink et al., 2017) and surface-elevation data (ASTER GDEM, V003). The following idealised elevation-dependent linear mass-balance profile was used,

$$b(z) = Max\{\beta(z - E), b_0\}. \tag{3}$$

Here $\beta$ is the balance gradient, $z$ is the surface elevation, and $E$ is the equilibrium-line altitude (ELA). $b_0$ is a cutoff on maximum accumulation taken to be 1.0 m/yr. The choice of $\beta$ is described later. In our mass-balance model, we neglected complicating factors like supraglacial debris cover and its effects on ablation, and the avalanche contribution to accumulation (Laha et al., 2017). The debris effects are expected to modify the scaling relations as well (Banerjee, 2020). Overall, the simulated glaciers cannot be considered faithful copies of the actual Himalayan glaciers. Rather, they constituted an ensemble of synthetic clean glaciers with realistic geometries (e.g., Farinotti and Huss, 2013) to be used here for a comparative study of the performances of the three models.

#### 2.2.1  A 2-d SIA model

The ice-flow dynamics was implemented within a two-dimensional SIA (Hutter, 1983) as a numerically efficient non-linear diffusion problem (Oerlemans, 2001). While SIA may not be the best method for simulating valley glaciers due to its limitation in describing ice-flow influenced by longitudinal stresses and/or steep bedrock slopes (Le Meur et al., 2004), there is enough evidence in the literature that SIA does a reasonable job of describing both the steady and transient dynamics of valley glaciers

(e.g., Vieli and Gudmundsson, 2004; Le Meur et al., 2004; Radić et al., 2008). The contribution of sliding to the flow was neglected here for simplicity.

The value of Glen's flow-law exponent was assumed to be 3 (e.g., Oerlemans, 2001). For the sake of simplicity, we did not tune any of the model parameters to match the observed ice-thickness and/or flow velocity on any of these glaciers. The only exception was ELA which was tuned to obtain the initial steady state as described below. In order to avoid possible dependence of the results on any specific choice of parameters, we picked the parameters related to mass balance and flow from random distributions. The rate constant of Glen's law was picked randomly from the set $\{0.5, 0.6, ..., 1.4, 1.5\} \times 10^{-24}$ $Pa^{-3}s^{-1}$ for each of the glaciers. This range of values is comparable to those used to model mountain glaciers previously (Radić et al., 2008). The balance gradient $\beta$ was also picked randomly from the set of values $\{0.005, 0.006, ..., 0.009, 0.010\}$ $yr^{-1}$ for each glacier. This range of $\beta$-values is comparable to the observed mass-balance gradients in the Himalaya (e.g., Wagnon et al., 2013).

The model was integrated using a linearised implicit finite-difference scheme (Hindmarsh and Payne, 1996), with a no-slip boundary condition at the ice-bedrock interface and a no-flux boundary condition at the domain boundary. An iterative conjugate-gradient method was employed within the implicit scheme, with a spatial grid-size of $100\,m \times 100\,m$ and time steps of 0.01 years. To avoid the known problem of a possible violation of mass conservation in SIA on steep terrains (Jarosch et al., 2013), we smoothed the bedrock with a centrally-weighted $3 \times 3$ moving-window averaging. In addition, the conservation of ice was explicitly monitored by tracking the total accumulation and ablation on the glacier surface, and the ice flux out of the glacier boundary in the ablation zone. The cumulative net gain of ice matched the total ice in the domain to within one part per $10^9$ at any time $t$. Only on three glaciers (out of the total of 814), a violation of conservation due to steep bedrock was observed, and these three were not considered in our analysis (supplementary figure S2). One more glacier had to be removed where an erroneously mapped truncated tributary led to an unrealistic piling up of ice (Supplementary fig. S2).

The SIA simulation was run starting with an empty bedrock, with the initial $E$ being the median elevation. The simulation was continued until an approximate steady state was reached such that the absolute value of the net specific balance was less than $10^{-4}$ m/year. Subsequently, $E$ was moved up or down, and the simulation was repeated until the extent of the steady state was similar to the present glacier extent (RGI, 2017) (Supplementary fig. S2). Once the desired steady state was found (See supplementary fig. S3 for a few examples), the glaciers were perturbed by a 50 m step rise in ELA. Subsequently, the annual values of area and volume were recorded for the next 1000 years (Supplementary fig. S4). The mean and standard deviation of the modelled ELA for these 810 glaciers were 5480 and 445 m, respectively.

Out of the total 810 simulated glaciers from the Ganga basin, on 98 glaciers the fractional change in glacier area at $t = 1000$ was more than 50%, and these were excluded from the analysis. This was necessary as a linear-response model can only be applied to glaciers with small relative changes (Oerlemans, 2001). We confirmed that the nature of our results does not depend on the precise value of this cutoff (Supplementary fig. S6). An additional 9 glaciers had response time larger than 500 years and they were removed. This was done to avoid a possible overestimation of the response time whenever its magnitude was comparable to or larger than the total simulation period of 1000 years (supplementary fig. S7). The removal of these 9 glaciers led to a reduction in the number (total area) of simulated glaciers by only $\sim 1\%(\sim 2\%)$.

Finally, we were left with an ensemble of 703 synthetic Himalayan glaciers (Supplementary fig. S1), with area in the range of $2.2-156.0$ km$^2$ (a median value $5.5$ km$^2$). The steady glaciers modelled with SIA had, on the average, 1.25 times larger area and 1.66 times larger ice-thickness (supplementary figs. S3, S8) compared to the corresponding estimates of Kraaijenbrink et al. (2017). The higher thickness of the modelled glaciers can be ascribed to a larger modelled area, a steady mass balance, and an uncalibrated SIA model. The total area and volume of these 703 synthetic glaciers were 6865 km$^2$ and 847 km$^3$, respectively. This set covered 86% of the total 810 glaciers number-wise, and 89% area-wise. The distributions of glacier area and mean slope for the two sets of 810 and 703 synthetic glaciers are shown in supplementary fig. S8.

### 2.2.2 Scaling model

The response of the above set of 703 steady-state glaciers to a 50 m instantaneous rise in ELA was also computed with a scaling model (Radić et al., 2007). The SIA-derived initial steady-state volume, area, and hypsometry (with the bin size of 25 m) for each of the glaciers were used as the starting point. For any of the modelled glaciers, the scaling and SIA models used the same mass-balance parameters. At any time $t$ during the evolution, the mass-balance function (eq.3) was summed over the instantaneous glacier hypsometry to obtain the net volume loss for that particular time step. The corresponding area loss was then obtained using Eq. 2. The reduction in the area was assumed to have taken place in the lowest elevation band/s of each glacier (Radić et al., 2007). The scaling exponent was fixed at $\gamma = 1.286$ because of the assumed linear mass-balance profiles of the simulated glaciers (i.e., $m = 1$). The annual-resolution time series of area and volume were recorded for 1000 years for each of the glaciers.

### 2.2.3 Glacier response properties

For each of the 703 glaciers, the time series of volume and area as obtained using the SIA and scaling models, were separately fitted to linear-response forms (e.g., eq. 9 below) to obtain the corresponding best-fit values of the four linear-response parameters (the climate sensitivities and the response times of area and volume) for each of them (supplementary fig. S4).

Note that applying a step change in ELA of a steady-state glacier to obtain the step-response function is a standard prescription for obtaining glacier response properties (Oerlemans, 2001; Vieli and Gudmundsson, 2004; Harrison et al., 2001; Bach et al., 2018). Within a linear-response assumption, the step-responses of volume and area have an exponential form (e.g., eq. 9 below). The asymptotic exponential decay time is the response time of the glacier, and the asymptotic magnitude of the decay is the climate sensitivity. Because of the deviations of the simulated response from a pure exponential decay (supplementary fig. S4), the best-fit response time may be slightly different from the $e$-folding time, which has been used in some of the previous studies (e.g., Vieli and Gudmundsson, 2004; Bach et al., 2018). However, we take the best-fit asymptotic decay time to be the response time. By definition, it minimises the deviation between the predictions of the SIA and linear-response models, and thus, improves the performance of the latter in reproducing SIA results to some extent. We confirm that the difference between the above two definitions of the response time is small.

The best-fit linear-response properties obtained from the scaling model results for the 703 glaciers were used to verify the corresponding theoretical expressions obtained from scaling theory (eqs. 8, 11, 12, 13 below). The best-fit response times and

climate sensitivities obtained from the SIA simulations of the 703 glaciers were used to fit for empirical relations that are motivated by the corresponding expressions derived from the scaling theory. These fitted forms would allow estimation of the response properties of any given glaciers as functions of properties like mean thickness, mass balance gradient, and so on. All the above fits were performed in a log-log scale, and $R^2$ of the fits were noted.

### 2.2.4 A linear-response model

The best-fit empirical parameterisations for climate sensitivity and response time obtained by fitting the SIA results as described above (given in eqs. $14-17$ later), were used to run a linear-response model simulation for any given glacier. This model was applied to simulate the response of the above 703 synthetic Himalayan glaciers to a 50 m step-change in ELA at $t = 0$. We emphasise that for the linear-response model, we did not use the best-fit the response properties of the individual glacier derived from the SIA simulations. Rather, the parameterisations of the same obtained by fitting the SIA-derived response properties (given in eqs. $14-17$ later) were utilised. These parameterisations thus allow the model to be applied to any other set of Himalayan glaciers without the need for simulating them with SIA first.

To assess the uncertainty of the linear-response model output, the uncertainty of each of the fit parameters was set equal to the corresponding standard error, and the 95% uncertainty band for the linear-response model outputs were generated using a Monte Carlo method.

To test the applicability of the above linear-response model that was calibrated using SIA results for the 703 central Himalayan glaciers, the same model was applied to a different set of 204 glaciers from the western Himalaya. The parameterisations developed for the central Himalayan glaciers as discussed above (given in eqs. $14-17$ later) were used to estimate the response properties of each of these western Himalayan glaciers using input values of corresponding mass-balance gradient, mean thickness and ablation rate near the terminus. For these western Himalayan glaciers, SIA and scaling model simulations were also performed following the procedures as detailed above. The glaciers showing more than 50% change at the 500 year mark in the corresponding SIA simulations were left out as before, and the time series of total area and total volume of 164 western Himalayan glaciers obtained using the three different models were compared.

## 3 Results and Discussions

### 3.1 Theoretical results

Below, we derive some relevant consequences of the time-invariant scaling assumption, including expressions for the climate sensitivity and response time of glacier area and volume. These results are expected to be generally valid for all scaling models those are based on eq. 2.

### 3.1.1 The rates of area and volume change

Eq. 2, which was derived from eq. 1 assuming a time-independent $c$, implies,

$$\dot{V} = \gamma c A^{\gamma-1} \dot{A} = \gamma h \dot{A}. \tag{4}$$

Here $\dot{V}$ and $\dot{A}$ denote the corresponding rates of change of glacier volume and area, respectively. If the net specific balance is $\delta b$ (in m/yr), then the annual rate of volume loss $\dot{V} = \delta b A$. This, together with eq. 4, implies,

$$\dot{A} = \frac{\delta b}{\gamma h} A \tag{5}$$

$$= \frac{\delta b}{\gamma c} A^{2-\gamma}. \tag{6}$$

Thus, in the scaling models, the rate of change of area scales with the glacier area with an exponent $(2-\gamma)$. This is consistent with empirical observations for real glaciers as well (Banerjee and Kumari, 2019). As the scale factor $\frac{\delta b}{\gamma c}$ in the right-hand side (RHS) of eq. 5 is proportional to the net specific mass balance, this may be a convenient way of obtaining mean regional thinning rates from relatively straightforward remote-sensing measurements of the rate of area change. However, the accuracy of this relation is contingent on the validity of the assumption of a time-independent $c$.

### 3.1.2 Area response time

To compute the area response time, let us consider a constant perturbation, i.e., a step change in ELA applied to a steady glacier for time $t \geq 0$ (e.g., Oerlemans, 2001). Let's denote the corresponding instantaneous net negative balance at $t=0$ by $\delta b_0 A$, the asymptotic ($t \to \infty$) shrinkage of glacier area by $\Delta A_\infty \equiv A(0) - A(t \to \infty)$, and that of ice volume by $\Delta V_\infty$. Then, we have (Harrison et al., 2001),

$$\Delta A_\infty b_t + \beta \Delta V_\infty \approx -\delta b_0 A. \tag{7}$$

Here $b_t$ is the ablation rate near the terminus. The area response time of the glacier can be expressed as $\tau_A \approx \Delta A_\infty / \dot{A}$. Therefore, using the above expressions for $\dot{A}$ (Eq. 5) and $\Delta A_\infty$ (Eq. 7), we obtain,

$$\tau_A = -(\frac{b_t}{\gamma h} + \beta)^{-1} \equiv \tau^*. \tag{8}$$

Here the symbol $\tau^*$ is a convenient shorthand notation for the time scale $-(\frac{b_t}{\gamma h} + \beta)^{-1}$. In the above derivation, $\Delta V_\infty$ that appears in eq. 7 is eliminated with the help of eq. 2. Eq. 8 is comparable with the expression of area response time as given by Harrison et al. (2001), or Lüthi (2009).

### 3.1.3 Volume response time

The instantaneous change in volume ($\Delta V(t)$) for a steady glacier perturbed by a small step change in ELA at $t=0$ is given by,

$$\Delta V(t) = \Delta V_\infty (1 - e^{-t/\tau_v}), \tag{9}$$

where $\tau_v$ is the volume response time and $\Delta V_\infty$ is the volume sensitivity (e.g., Lüthi, 2009). Now, $V(t), V(0)$, and $V(t \to \infty)$ appearing in eq. 9 can be expressed in terms of $A(t), A(0)$, and $A(t \to \infty)$, respectively, with the help of corresponding scaling relations (eq. 1). This, in the limit of a small fractional changes in area, yields,

$$\Delta A(t) = \Delta A_\infty (1 - e^{-t/\tau_v}). \tag{10}$$

Comparing the above two equations, and using eq. 8 one obtains,

$$\tau_A = \tau_V = \tau^*. \tag{11}$$

Thus, all scaling models implicitly assume the area and volume response times of a glacier to be equal to each other. However, it is known that for mountain glaciers area response time is larger than the volume response time within an SIA model (Oerlemans, 2001; Vieli and Gudmundsson, 2004). Therefore, the assumed equality of the two response times in scaling models (eq. 11) contradicts the existing SIA results. This is an intrinsic bias that is present in any scaling model.

After a step change in ELA, as the ablation zone shrinks, the initial net negative balance of a glacier gradually decays to zero over a period determined by the corresponding response time. A longer area response time in SIA implies that this reduction in the ablation zone is slower here than that in a scaling model. A corresponding feedback of a larger ablation zone on the net mass balance should then lead to a higher long-term volume loss in an SIA model than that in a scaling model. This indicates the possibility of a low bias in scaling model estimates of the climate sensitivity of volume, or equivalently, that in the long-term changes in glacier volume due to any rise in ELA.

### 3.1.4 Climate sensitivity of area and volume

An expression for the climate sensitivity of glacier area ($\Delta A_\infty$), which is the asymptotic change in area due to a change in ELA by $\delta E$, is obtained by eliminating $\Delta V_\infty$ from eq. 7 using eq. 2,

$$\frac{\Delta A_\infty}{A} = \frac{\tau^* \beta \delta E}{\gamma h} \equiv \alpha^*. \tag{12}$$

Here we have used the definition of $\tau^*$ (Eq. 8), and that of $\delta b_0 \approx \beta \delta E$ for a step change in ELA by $\delta E$. The RHS of the above equation is denoted by $\alpha^*$ for convenience.

The corresponding expression for $\frac{\Delta V_\infty}{V}$ is then obtained using Eq. 2,

$$\frac{\Delta V_\infty}{V} = \gamma \alpha^*. \tag{13}$$

Again, Eq. 13 is comparable to the expression of volume sensitivity as derived by (Harrison et al., 2001), where the authors used an arbitrary thickness scale $H$, instead of the denominator of $\gamma h$ appearing in the definition of $\alpha^*$ above.

Strictly speaking, the climate sensitivity of area and volume with respect to a change in ELA should be defined as $\frac{\Delta A_\infty}{\delta E}$ and $\frac{\Delta V_\infty}{\delta E}$, respectively. However, in this paper, we use $\Delta A_\infty$ and $\Delta V_\infty$ as the corresponding sensitivities to simplify the notation.

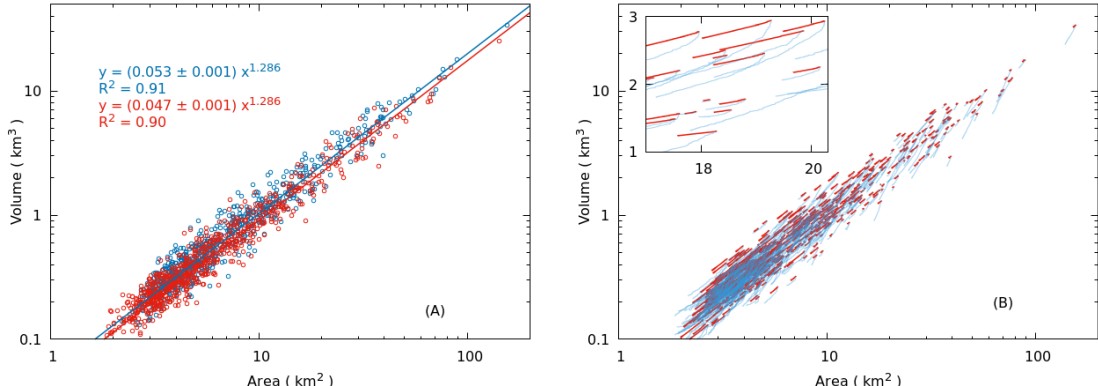

**Figure 1.** A) Glacier volume as a function of area for the 703 Himalayan glaciers simulated with SIA at $t = 0$ yr (blue circles), and at $t = 500$ yr (red circles) are plotted along with the corresponding best-fit scaling relations (blue and red solid lines). The corresponding fitted functions, and $R^2$ values are shown in blue and red texts, respectively. B) The trajectories of the 703 glaciers in the $V - A$ plane as simulated with SIA (thick red lines) and scaling (thin blue lines) models. The inset is a zoomed-in version of the same plot, but with a linear scale.

## 3.2 Numerical results

### 3.2.1 Volume-area scaling and a time-dependent scale factor in the SIA model

Following eq. 1, a power-law relation between the area and volume of the 703 glaciers with an exponent $\gamma = 1 + \frac{m+1}{m+n+3} = 1.286$ is expected as $m = 1$ and $n = 3$. The ensemble of glaciers modelled with SIA did conform to above power-law relation

$V = cA^{1.286}$ at any time $t$ with a single best-fit $c$. The scale factor slowly decreased with time. For example, fig. 1a shows the power-law fits at $t = 0$ and $t = 500$ years ($R^2 = 0.9$), where the best-fit $c$-values were $0.053 \pm 0.001$ and $0.47 \pm 0.001$ km$^{3-2\gamma}$, respectively. This implies a $\sim 11\%$ reduction in $c$ for the ensemble over the period of 500 years after the step-change in ELA was applied. A time-dependent $c$ is consistent with the theoretical arguments of Bahr et al. (2015).

The slow and systematic decline in $c$ for the ensemble of shrinking glaciers simulated with the SIA model contradicts the

basic assumption of scaling models that $c$ is time-invariant. A decreasing $c$ would mean eq. 2 is violated, with $\frac{\Delta V}{V} = \gamma \frac{\Delta A}{A} + \frac{\Delta c}{c}$. Note that all three fractional changes involved in this relation are negative. Therefore, for any given $|\Delta A|$, the corresponding $|\Delta V|$ is going to be larger in SIA model than that in a scaling model where $\frac{\Delta c}{c}$ is assumed to be zero (eq. 2). Even though the decline in $c$ is only about 11%, it may be associated with a stronger low bias in the long-term change predicted by scaling models. This is because a larger volume change in SIA would lead to a thinner glacier, and a corresponding surface-elevation

feedback to mass balance is likely to amplify the corresponding long-term mass loss over time.

The dependence of the glacier-specific scale factor on the mean slope is known (Bahr et al., 2015) and has been incorporated in modified scaling relations where volume is a power-law function both area and slope (e.g., Grinsted, 2013; Zekollari and Huybrechts , 2015). For the simulated 703 glaciers, the mean slope increases with time as the area is lost preferentially from

the gently-sloping lower ablation zone. For example, the median slope of the 703 simulated glaciers reduced from 0.41 at $t = 0$ to 0.37 at $t = 500$ years. This $\sim 10\%$ reduction in slope is expected to lead to a $\sim 5\%$ decline in $c$ (Bahr et al., 2015) . So, at least part of the time dependence of $c$ for transient glaciers in SIA simulation is explained by the slope-dependence of $c$. However, there may be other factors contributing to the decline in $c$ for the transient glaciers as discussed below.

### 3.2.2 Area and volume response times

The theoretical predictions for glacier area and volume response time (eq. 11) worked rather well for the scaling model results (figs. 2C, and 2D), with best-fit relations of $\tau_V = (0.914 \pm 0.002)\tau_A$ with $R^2 = 0.99$, and $\tau_A = (1.066 \pm 0.008)\tau^*$ with $R^2 = 0.80$.

For SIA simulations, the data showed that $\tau_A > \tau_V$, and that the two response times were still proportional to each other (fig. 3C: $\tau_V = (0.687 \pm 0.004)\tau_A$, with $R^2 = 0.94$). Also, $\tau_A$ was proportional to $\tau^*$ to a good approximation (fig. 3D: $\tau_A = (2.56 \pm 0.04)\tau^*$, with $R^2 = 0.53$). Interestingly, the value of the proportionality constant in the latter relation as obtained from SIA was about 2.4 times larger than the corresponding value obtained in the scaling model. This underlines the relatively large underestimation of area response time by the scaling model. Similarly, the volume response time was about 1.8 times larger in the SIA simulation than the corresponding scaling model value. This implies that for a given ELA perturbation, the glacier response is much faster in the scaling model compared to that in the SIA model for the ensemble of 703 synthetic glaciers.

Apart from the overall underestimation of area and volume response times by the scaling model, another serious limitation of scaling models that emerges from the above analysis is that here the area and volume response times are equal to each other (eq. 11, and fig. 3C). In contrast, the SIA model predicted $\tau_A \approx 1.5\tau_V$. The ratio of the two response times obtained from the 2-d SIA model here is generally consistent with earlier results based on 1-d flowline models (Oerlemans, 2001; Vieli and Gudmundsson, 2004). The equality of the two response times in the scaling model led to a linear trajectory in $V - A$ plane for the transient glaciers (fig. 1B). In comparison, a relatively larger area response time, together with a slow initial changes in area (supplementary figs. S4, S10), led to curved $V - A$ plane trajectories for individual transient glaciers in the SIA model simulations. In particular, a slowly changing area means the $V - A$ trajectories bend downward causing $c$ to reduce for the transient ensemble (fig 1). Moreover, At the early stages of response, glaciers simulated by a scaling model lose area much quicker than those simulated by an SIA model (fig. 1B). The associated net mass-balance feedbacks then lead to a subdued long-term volume response in the scaling model, and a comparatively stronger volume response in the SIA model, just as predicted in sect. 3.1.3.

### 3.2.3 The climate sensitivity of glacier area and volume

For the 703 glaciers simulated by the scaling model, the fitted asymptotic fractional changes in area and volume, or equivalently, the corresponding (fractional) climate sensitivities, were proportional to each other (fig. 2A: $\frac{\Delta V_\infty}{V} = (1.383 \pm 0.003)\frac{\Delta A_\infty}{A}$, with $R^2$=0.99). Here, the best-fit constant of proportionality is close to, but about 8% larger than $\gamma = 1.286$ as predicted by eq. 2.

In contrast, the SIA simulations obtained $\frac{\Delta V_\infty}{V} = (1.93 \pm 0.02)\frac{\Delta A_\infty}{A}$, with $R^2$=0.85 (fig. 3A). In this case, the constant of proportionality was $\sim 1.5\gamma$, compared to the corresponding value of $\sim \gamma$ in the scaling model. This larger value of the ratio of

the two climate sensitivities in the SIA model is consistent with the observed decline in $c$ for the transient glaciers simulated with this model (fig. 1). Note that no theoretical prediction is available for the ratio of asymptotic fractional changes in volume and area in an SIA model.

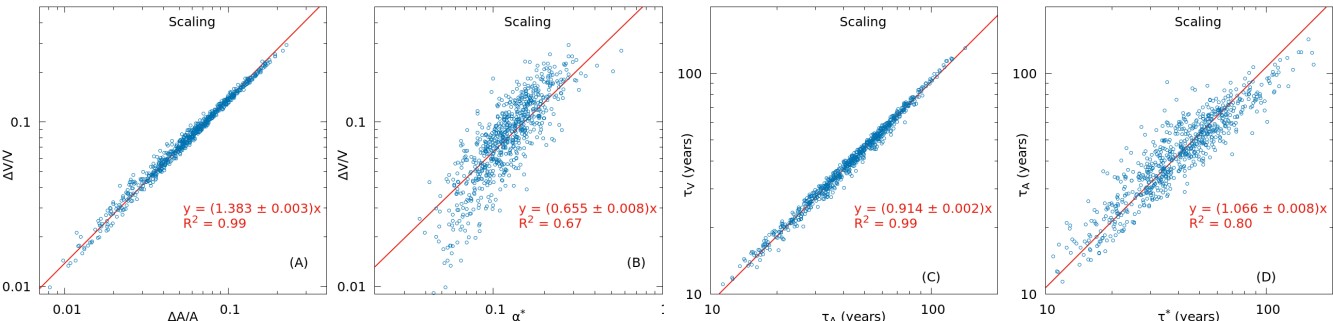

**Figure 2.** Scaling model simulations of the 703 synthetic Himalayan glacier show that, (A) the best-fit (fractional) climate sensitivities of area and volume are proportional to each other, (B) The climate sensitivity of volume is proportional to $\alpha^* \equiv \frac{\beta \delta E \tau^*}{\gamma h}$, (C) The response times associated with glaciers area and volume are approximately equal, and (D) the volume response time is approximately equal to $\tau^* \equiv -(\frac{b_t}{\gamma h} + \beta)^{-1}$. In all the above plots, the corresponding best-fit curves are shown with red lines. The fit parameters and $R^2$ of the fits are also given. These numerical trends are consistent with theoretical results derived in sect. 3.1.

Fig. 2B shows that in the scaling model, climate sensitivity of glacier volume is proportional to $\alpha^*$ ($\frac{\Delta V_\infty}{V} = (0.655 \pm 0.008)\alpha^*$, with $R^2 = 0.67$). This is in line with eq. 13, except that the constant of proportionality is significantly less than $\gamma$. A similar proportionality between the SIA-derived best-fit $\frac{\Delta V_\infty}{V}$ and $\alpha^*$ is shown in fig. 3B, with $\frac{\Delta V_\infty}{V} = (1.71 \pm 0.03)\alpha^*$. However, in this case the fit is relatively noisy with $R^2 = 0.48$.

The above relations suggest that the climate sensitivity of volume in the SIA simulation was about 2.6 times larger than that in the scaling model. Similarly, the climate sensitivity of glacier area obtained from the SIA model was also about 1.9 times larger than that obtained from the scaling model. This trend of a relatively large (by about a factor of about 2) underestimation of climate sensitivity of glacier volume and area by the scaling model is consistent with the effects of a relatively faster shrinkage of the ablation zone in the early stages of the response as discussed in 3.1.3 and 3.2.2.

### 3.2.4 The total glacier loss estimated using the three models

Starting with an initial volume (area) of 847 km$^3$ (6865 km$^2$), the 703 glaciers simulated by SIA lost a total of 194 km$^3$ (726 km$^2$) of volume (area) in 500 years due to the step-rise in ELA by 50 m. As shown in fig 4, both the scaling and the linear-response models underestimated the long-term change in total area in this experiment, with estimated area changes of 334 and 623 km$^2$, respectively. The scaling-model prediction for area change was only 46% of the corresponding SIA estimate, while the linear-response model estimate was 86% of that of SIA. Similar trends were seen for the magnitudes of estimated volume change as well, with the respective scaling and linear-response model estimates being $\sim 31\%$ and $\sim 75\%$ of the corresponding SIA prediction (fig 4). We confirmed that the nature of the above results does not depend on the chosen cut-off of 50% change

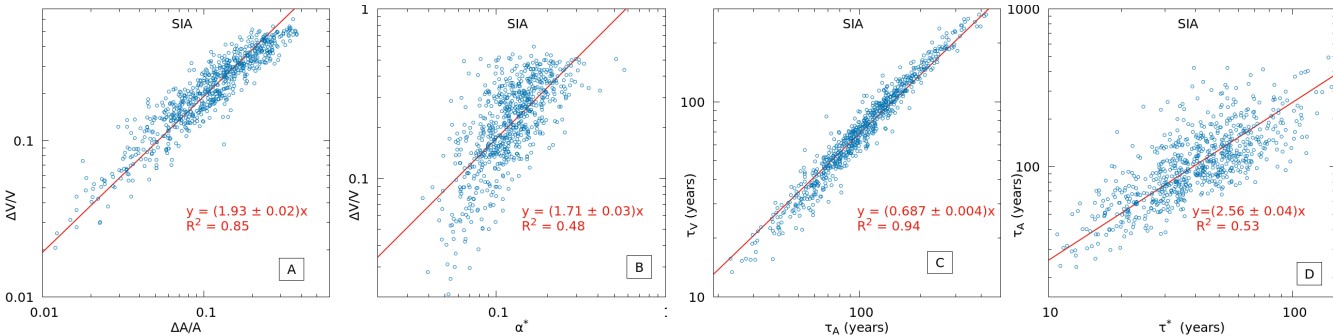

**Figure 3.** Results from the SIA simulations of the 703 synthetic Himalayan glacier show that, (A) The climate sensitivities of area and volume are proportional to each other, (B) The climate sensitivity of glacier volume is proportional to $\alpha^* = \frac{\beta\delta E\tau^*}{\gamma h}$, (C) The response times associated with glaciers area and volume are proportional to each other, and (D) The volume response time is proportional to $\tau^* = -(\frac{b_t}{\gamma h} + \beta)^{-1}$. The fitted functions are shown with red lines. The corresponding fit parameters and $R^2$ of the fits are also given. See text for detailed discussions.

that was used to select the 703 glaciers (Supplementary fig. S6). In fact, with a smaller cut-off, the linear-response model estimates were even closer to the corresponding SIA estimates (Supplementary fig. S6). This is expected as linear-response models are derived in the limit of small fractional changes (Oerlemans, 2001).

The low-bias in the long-term changes of glacier area and volume computed with the scaling model is consistent with the underestimation of corresponding climate sensitivities by this model (sect. 3.2.3). On shorter time scales of multiple decades, an underestimation of response times by about a factor of 2 (sect. 3.2.2) partly compensates for a corresponding underestimation of the climate sensitivities (sect. 3.2.3), and the deviations between the SIA and the scaling model are not that prominent (fig. 4). The biases in the scaling model become clearer over multiple centuries (fig. 4).

Note that, depending on the details of the scaling and SIA models compared, or the set of glaciers simulated, the actual magnitude of the biases in scaling-model derived climate sensitivity, response time, and long-term glacier change could be different from these here. However, based on the theoretical arguments and numerical evidence presented, similar qualitative trends are expected if the above exercise were to be repeated with a more detailed model and/or for a more realistic set of glaciers.

The above results indicate the possibility of a negative bias in scaling model estimates of future changes in mountain glaciers, and the corresponding contribution to sea-level rise. As an example, let us consider a recent comparison (Hock et al, 2019) of projected end-of-the-century sea-level rise contribution of glaciers from 6 different models, with 5 of them being based on some form of scaling. In that intercomparison study, the hypsometric-adjustment-based model (Huss and Hock, 2015) consistently predicted the largest fractional change of global glacier volume and area under various climate scenarios (Table 3 of Hock

et al (2019)). In another recent comparison, similar trends are seen as far as global-scale fractional volume loss by 2100 are concerned (Figure S17−S20 of Marzeion et al. (2020)), although on a regional scale there are differences. However, it is difficult to draw a definite conclusion about any potential bias in scaling models from the above-mentioned studies as there are wide differences among the model runs in terms of the initial conditions, climate forcing, and mass-balance parameterisations

used. An intercomparison of the models where the same set of glaciers, with the same initial geometry and volume, are simulated under the same mass-balance forcing - similar to the strategy used in the present study - is necessary to identify possible biases in the existing scaling models.

The above results show that the linear-response model outperformed the scaling model, producing a closer match with the SIA results for the 703 synthetic glaciers from the Gangetic Himalaya. However, this linear-response model was calibrated using the SIA results for the same set of glaciers. Therefore, this match is not enough to establish the effectiveness of the linear-response model. To confirm the improved performance of the linear-response model compared to that of the scaling model, we applied both the models to simulate a different set of 164 glaciers in the western Himalaya (supplementary fig. S1). The best-fit linear-response properties obtained from SIA simulation of the 703 central Himalayan glaciers were first fitted to obtain four equations (eqs. $14-17$) that relate the response properties to $\beta, \gamma, h$ and $b_t$ as described before. The same equations were used to estimate the response properties of each of the 164 western Himalayan glaciers as required for the linear-response model simulations. In this independent experiment, the linear-response model again outperformed the scaling model in reproducing the corresponding SIA results (supplementary fig. S9). This confirms that the linear-response model, along with eqs. $14-17$, can be used for computing long-term glacier changes accurately.

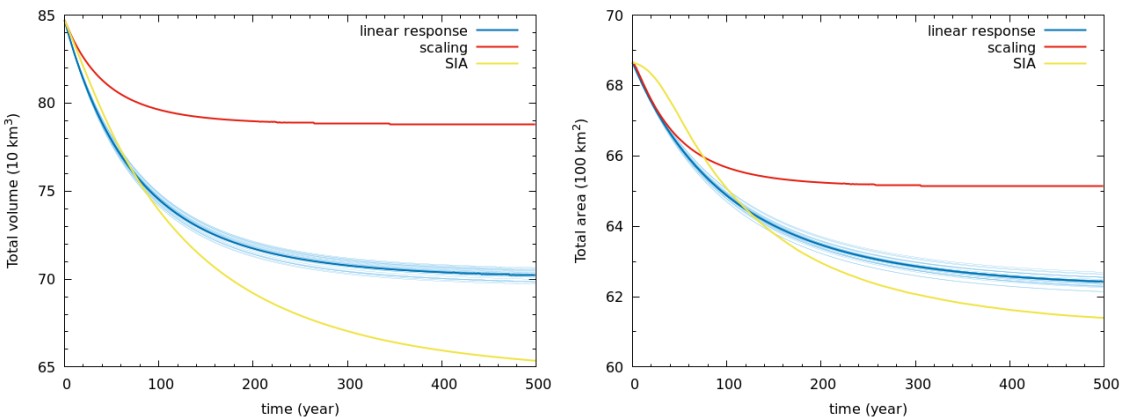

**Figure 4.** The evolution of the total (A) volume, and (B) area of the ensemble of 703 Himalayan glaciers simulated with three different methods: SIA, scaling, and linear-response models. The uncertainty bands for the linear response model results are also shown. See text for details.

### 3.3 The effects of glacier geometry

Can the biases in the scaling model described above, be artefacts arising out of some peculiarities of the geometry of the specific set of glaciers being simulated, and are not relevant in general for scaling model computations of global-scale mass loss of mountain glaciers? To rule out this possibility, we simulated the response of a set of highly idealised synthetic glaciers using both a flowline model (Banerjee, 2017) and the above scaling model (Radić et al., 2007). Note that this flowline model included sliding as well. All of these synthetic glaciers have the same constant-width, the same linear bedrock with a constant

slope, and the same linear mass-balance profile. Only the ELA was varied between glaciers. Even for this highly idealised set of glaciers, the scaling model estimates for the evolution of total area and volume showed biases compared to those obtained from the flowline model (supplementary fig. S9), and these biases were qualitatively very similar to those depicted in figs. 1 and 4. Again, the scaling model predicted relatively smaller climate sensitivities, a relatively faster area response, and a low-bias in the long-term changes, compared to corresponding flowline-model estimates (supplementary fig. S9).

The above flowline-model experiment provides an additional piece of evidence that the scaling-model biases discussed in this paper are, in general, expected to be present in scaling model simulations of any set of glaciers. We re-emphasise that even though the biases are expected to be qualitatively similar to those presented here, the magnitudes of the biases are likely to depend on the detailed characteristics (related to geometry, flow, and mass-balance processes) of the glaciers studied and the models used.

## 3.4 The linear-response model, and its application to real glaciers

As described above, we have used results from the 2-d SIA model simulations of the response of 703 synthetic Himalayan glaciers to a 50 m step change in ELA, to obtain the following best-fit parameterisations of the glacier response properties (i.e., $\frac{\Delta V_\infty}{V}, \frac{\Delta A_\infty}{A}, \tau_A$ and $\tau_V$).

$$\frac{\Delta V_\infty}{V} = (1.71 \pm 0.03)\alpha^*, \tag{14}$$

$$\frac{\Delta V_\infty}{V} = (1.93 \pm 0.02)\frac{\Delta A_\infty}{A}, \tag{15}$$

$$\tau_A = (2.56 \pm 0.04)\tau^*, \tag{16}$$

$$\tau_V = (0.687 \pm 0.004)\tau_A. \tag{17}$$

Here as defined before, $\tau^* \equiv -(\frac{b_t}{\gamma h} + \beta)^{-1}$, $\alpha^* \equiv \frac{\beta \delta E \tau^*}{\gamma h}$, and $\delta E = 50$ m.

With the estimated glacier-specific response properties obtained from eqs. 14−17, it is possible to compute the evolution glacier volume and area accurately for any glacier and for any arbitrary ELA forcing function. For this, the following general solution of the linear-response equation is used.

$$\Delta V(t) = \Delta V(0)e^{-t/\tau_V} + \frac{\Delta V_\infty}{\tau_V \delta E} \int_0^t \Delta E(t')e^{-(t-t')/\tau_V}\,dt' \tag{18}$$

Here $\Delta E(t)$ is the given (arbitrary) ELA forcing function. This equation simply states that any continuous ELA change can be interpreted as the sum total of a series of discrete impulses, and the corresponding net response is given by a superposition of suitably delayed responses due to each of the impulses. An analogous expression can be obtained for the area evolution just by replacing all the $V$'s in the above equation with $A$'s.

Note that the above formulation does not require the initial state to be steady. As long as the glacier is close to a steady state, a linear-response theory will be a good approximation (Oerlemans, 2001). However, an additional initial condition, i.e., the value of $\Delta V(0)$, is needed to apply the linear-response model to transient glaciers. $\Delta V(0)$ is the initial departure from a steady

state, and can be obtained from the observed rate of volume loss ($\dot{V}$) simply as, $\Delta V(0) = -\tau_V \dot{V}$. Thus, the linear-response model can be used to evolve the area and volume of a real set of glaciers for any arbitrary time-dependent ELA forcing given the initial rates of change of volume and area, initial thickness, mass-balance gradient, and melt rate near glacier terminus.

Since the above parameterisation of linear-response properties (eqs. 14−17) are derived from SIA simulations of an ensemble of Himalayan glaciers, when applying them to any other glacierised region in the world, it may be necessary to simulate a few tens of glaciers (having a representative range of area and slope) from that region using SIA first and confirm the accuracy of the above parameterisations.

Due to the noise present in the fits (fig. 3), the linear-response model predictions for an individual glacier would have significant uncertainties. However, for a large set of glaciers, the linear-response model provides accurate estimates of the total area and volume evolution (fig. 4, supplementary figs. S6 and S9).

### 3.5   Limitations of the present study

Because of the idealised descriptions of ice flow and the mass-balance profile (as discussed in sect. 2.2), and the absence of model calibration to match the available observed data of surface velocity, ice thickness, recent mass balance, etc., the glaciers simulated here are not faithful copies of the Himalayan ones. For a set of more realistic glaciers, the magnitudes of the corresponding biases in scaling-model derived climate sensitivity and response time could be different from those obtained here. However, based on the theoretical arguments and numerical evidence presented, similar qualitative trends are expected if the above exercise were to be repeated for a more realistic model that includes higher-order mechanics, a more realistic mass-balance model, and so on. Similarly, The parameterisations for the linear-response properties given here are obtained from 2-d simulations of 703 synthetic Himalayan glaciers with some idealisations (sect. 2.2) and without any tuning of model parameters. The fit-parameters in eqs. 14-17 may be different for a different set of glaciers. The parameterisations may also change if a more detailed and calibrated model of the same glaciers is used. However, the protocol used here to obtain the parameterisation for linear response-properties can be directly applied without any change for any set of glaciers and for any ice-flow/mass-balance model. While applying the linear response model to any other region, it may be useful to obtain the response properties of a few tens of representative glaciers using flow-model simulations and check if any recalibration of the parameterisation as given in eqs. 14-17 is necessary.

### 4   Summary and Conclusions

We performed a theoretical analysis of the response of mountain glaciers within a time-independent scaling assumption. In addition, the step-response of 703 steady-state synthetic Himalayan glaciers with realistic geometries and idealised mass-balance profiles were simulated with three different models: a scaling model, a 2-d SIA model, and a linear-response model. The results obtained are as follows.

- Analytical expressions for climate sensitivity and response time of glacier area and volume are derived within a time-independent scaling assumption. These expressions are validated using results from the scaling model simulation of the ensemble of 703 glaciers.

- The response of the glaciers simulated with the 2-d SIA model reveals that the initial steady states and the transient states follow the volume-area scaling relation, with the best-fit scale factor reducing slowly with time.

- For the ensemble of glaciers studied, the scaling model obtains relatively smaller climate sensitivities of glacier area and volume by a factor of about 1.9 and 2.6, respectively, compared to those obtained from the SIA model. This results in a low bias in the long-term changes predicted by the scaling model.

- For the ensemble of glaciers studied, the scaling model underestimates volume (area) response time by a factor $\sim 1.8$ (2.4) compared to the corresponding SIA estimates.

- For the scaling model, $\tau_A \approx \tau_V$, and $\frac{\Delta V_\infty}{V} \approx \gamma \frac{\Delta A_\infty}{A}$. In contrast, for the SIA simulations, $\tau_A \approx 1.5\tau_V$ and $\frac{\Delta V_\infty}{V} \approx 1.5\gamma \frac{\Delta A_\infty}{A}$.

- The relatively larger ratio of the two response times in the SIA simulations, along with an initial slow change in the area, leads to curved $V - A$ trajectories, a decreasing $c$, and a relatively larger long-term volume loss for the transient glaciers due to a corresponding mass-balance feedback.

- A linear-response model based on the parameterisations of SIA-derived response properties helps reduce the biases in the long-term glacier changes predicted by the scaling model for the idealised central Himalayan glaciers. The improved performance of this model is validated on an independent set of 164 glaciers in the western Himalaya.

Based on the theoretical arguments and numerical evidence presented here, it is possible that qualitatively similar biases may generally be present in the long-term glacier changes computed with scaling models. However, the actual magnitudes of such biases in scaling models may be different from those obtained here for a set of synthetic Himalayan glaciers with idealised mass-balance profiles. Possible biases in scaling models may, in turn, lead to a low bias in the corresponding estimates of the long-term sea-level rise contribution from shrinking mountain glaciers. On a multidecadal scale, a faster response due to shorter response times in the scaling model can compensate for the effects of smaller climate sensitivities to some extent. However, the low biases in scaling model derived changes in glacier area and volume are likely to become apparent over longer time scales of multiple centuries. The linear-response model presented above could potentially be useful in predicting the long-term global glacier change and sea-level rise due to its accuracy and numerical efficiency.

*Code availability.* The glacier model codes are available on the repository: https://github.com/Disha-Patil/glacier_models.

*Author contributions.* AB designed the study, did the theoretical analysis, and wrote the paper. AJ and DP wrote the codes. AJ, DP, and AB ran the simulations. All the three authors contributed to the analysis of the simulated data and discussions.

*Competing interests.* We declare that there are no competing interests.

*Acknowledgements.* The authors acknowledge valuable inputs from reviewer Eviatar Bach, the anonymous reviewer, and editor Valentina Radić. The SIA code was developed with support from MoES grant no. MoES/PAMC/H&C/80/2016-PC-II. AJ was supported by MoES grant no MoES/PAMC/H-&C/79/2016-PC-II. Deepak Suryavanshi has contributed to the initial development of the SIA code.

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
