# Peer review of "Possible biases in scaling-based estimates of glacier change: A case study in the Himalaya"

_The Cryosphere, 2019_

## Referee Comment (RC1) · Eviatar Bach (Referee) · 16 Jan 2020

The paper is interesting, well-written, and has the potential to be useful for understanding biases in scaling-based projections of glacier volume evolution. However, I have a few major concerns which prevent me from recommending publication.

Major issues:

1. For the linear-response model based projections, the authors write that they fit the four parameters (area and volume sensitivities and response times) for each glacier based on the SIA data. They then validate the projections obtained using these pa-

rameters on the same SIA data. This is using the same data for fitting and validation, so it is not surprising that it replicates the data fairly well. Testing this method requires validation data that is not part of the fitting. A possible way to do this would be to only use a portion of the time-series of each glacier to fit, and validate on the rest (for example, fit on the first 50 years, project into the future, and validate on the 450 remaining years).

There is this this sentence which I was not clear on: "We have verified the linear-response model obtained by fitting the SIA simulation results for the ensemble of 551 central Himalayan glaciers, similarly outperforms the scaling-based method for another set of 143 glaciers from the western Himalaya (figure A2)." Were the parameters obtained for the central Himalayan glaciers somehow extrapolated to the western Himalayan ones? Or were the parameters fit for every western Himalayan glacier as well? It is not clear from the description. If the authors use an extrapolation method, it would be important to describe it.

2. The linear-response method is being proposed as an alternative to scaling-based methods for projecting glacier volume evolution. However, I am not clear on how this would be implemented in practice. The climate sensitivities $\Delta V\_inf$ and $\Delta A\_inf$ characterize the response of an initially steady-state glacier to a perturbation in the ELA. How can this be used to project evolution of a glacier that is already transient, and in a situation where it is not a single perturbation in the ELA, but that the ELA is continually rising?

3. Furthermore, it seems that the linear-response method would require a relatively long time-series of the area and volume evolution of each glacier in order to fit the parameters, which is often not available. I would like to see a discussion of the data requirements and feasibility for use in sea-level projections.

4. Although it is true that there are a priori arguments for what the scaling exponent gamma should be, in practice it can be quite different, even for simulated glaciers (e.g.,

Radić et al., 2007). So for use of the scaling-based method as a statistical projection method it would be more fair, in order to compare to the linear-response method, to estimate the scaling exponent from the SIA runs. Although Radić et al. (2007, 2008) showed that the volume evolution over 100 years is not very sensitive to the exponent, this may be different for the 500-year simulations. Also, how were the constants of proportionality c determined for each glacier?

In fact, we could expect that fitting the exponent and constant of proportionality individually for each glacier, as would be possible if a sufficiently long-time-series data were available for every glacier, would considerably reduce the bias of the scaling-based method.

Other issues: 1. The authors remove some glaciers from consideration in several parts of the paper, such as those that had fractional changes of more than 50% over 500 years, and those with response times higher than 300 years. Also, in another part of the paper, glaciers with large values of $\Delta A\_inf/A$ are removed, and another cut-off on $\Delta V\_inf/V$ is imposed. I don't see an adequate justification of why these were removed, and doing so biases the results. 2. "The minor differences are due to the time-invariant scaling assumption made here." Please clarify in more detail what is the difference between your derivations and those of Harrison (2001). 3. In Fig. 1B, scaling the SIA results by 10 for visual comparison is confusing. It's also hard to distinguish which are the thick and thin lines.

Minor issues: L9: "glacier" -> "glaciers" L51: "have" -> "has" L80: "using" -> "using a" L120: Missing parenthesis L166: "appearing" -> "appearing in" L175: "till" -> "until" L185: Should this be 551 glaciers? L191: "receives" -> "receive" L194: "These" -> "these" L258: "possible a" -> "a possible" L261: Missing parenthesis, "an" -> "and" L295: "are" -> "is" L299: "intruding" -> "introducing"

---

## Author Comment (AC1) · 29 Jan 2020

Argha Banerjee, Ajinkya Jadhav, and Disha Patil

We thank referee Eviatar Bach for his critical comments (RC1) on the manuscript. Below we provide point-by-point reply to his comments. The referee comments are highlighted with *red italicised fonts*.

*The paper is interesting, well-written, and has the potential to be useful for understanding biases in scaling-based projections of glacier volume evolution. However, I have a few major concerns which prevent me from recommending publication.*

**Major issues:**

*1. For the linear-response model based projections, the authors write that they fit the four parameters (area and volume sensitivities and response times) for each glacier based on the SIA data. They then validate the projections obtained using these parameters on the same SIA data. This is using the same data for fitting and validation, so it is not surprising that it replicates the data fairly well. Testing this method requires validation data that is not part of the fitting. A possible way to do this would be to only use a portion of the time-series of each glacier to fit, and validate on the rest (for example, fit on the first 50 years, project into the future, and validate on the 450 remaining years).*

*There is this this sentence which I was not clear on: "We have verified the linear-response model obtained by fitting the SIA simulation results for the ensemble of 551 central Himalayan glaciers, similarly outperforms the scaling-based method for another set of 143 glaciers from the western Himalaya (figure A2)." Were the parameters obtained for the central Himalayan glaciers somehow extrapolated to the western Himalayan ones? Or were the parameters fit for every western Himalayan glacier as well? It is not clear from the description. If the authors use an extrapolation method, it would be important to describe it.*

Let us clarify that our workflow is as follows:

- The response of 551 Central Himalayan glaciers to a 50 m step-change in

ELA is modeled with SIA to generate time series of area and volume.

- The area and volume evolution curves are fitted to obtain $\Delta V_\infty, \Delta A_\infty, \tau_A$, and $\tau_V$ for each of the glaciers.

- The response coefficients of the set of 551 glaciers obtained above are used to arrive at the following best-fit parameterisations. $\frac{\Delta V_\infty}{V} = (1.65 \pm 0.03)\alpha^*$
  $\frac{\Delta A_\infty}{A} = (1.87 \pm 0.02)^{-1}\frac{\Delta V_\infty}{V}$
  $\tau_A = (2.67 \pm 0.04)\tau^*$
  $\tau_V = (0.647 \pm 0.003)^{-1}\tau_A$
  where, $\tau^* = -(\frac{b_t}{\gamma h} + \beta)^{-1}$ and $\alpha^* = \tau^*\beta\delta E/\gamma h$.

- These four expressions are then used to run the linear-response model first for the same 551 glaciers (main fig 6), and subsequently, for 143 western Himalayan glaciers (main fig A2). (We emphasise that for modeling the second set of 143 glaciers, the four equations stated above, the ones that were obtained from results for the set of 551 glaciers, were used without any further calibration).

We agree with the referee that a favourable comparison between linear-response model and SIA results for the set of 551 central Himalayan glaciers is not an independent validation of the linear response model. However, the reasonable performance of the exact same parameterisations for 143 western Himalayan glaciers can be considered an independent validation of the linear-response model presented here. We propose to move the figure A2 to the main text to highlight this.

To avoid the possibility of any confusion, we shall tabulate these expressions and summarise the rationale in the discussion section of the revised manuscript. We shall also discuss how this model can be applied to any other set of glaciers, and details like the data requirements etc. as suggested by the referee in his comments below.

*2. The linear-response method is being proposed as an alternative to scaling-based methods for projecting glacier volume evolution. However, I am not clear on how this would be implemented in practice. The climate sensitivities $\Delta V_\infty$ and $\Delta A_\infty$ characterize the response of an initially steady-state glacier to a perturbation in the ELA. How can this be used to project evolution of a glacier that is already transient, and in a situation where it is not a single perturbation in the ELA, but that the ELA is continually rising?*

Linear response model is excepted to be a good approximation even if the initial state is a transient one, as long as the glacier being modelled is close to a steady state. It is not essential that the initial state is strictly a steady state. However, an additional initial condition is needed to apply linear-response model to a transient state. A continuous ELA change can be implemented as the sum total of a series of discrete steps every year (say). The net response is given by

a superposition of suitably delayed response profiles due to each of the steps. The above statements follow from the general solution of the linear-response equation,

$$\Delta V(t) = \Delta V(0)e^{-t/\tau} + \frac{\Delta V_\infty}{\delta E} \int_0^t \Delta E(t')e^{-(t-t')/\tau}dt'.$$

Here, $\frac{\Delta V_\infty}{\delta E}$ is the climate sensitivity of glacier volume. $\Delta V(0)$ is the initial departure from a steady state that can be obtained from the observed rate of volume change as $\Delta V(0) = -\tau \frac{dV(0)}{dt}$. A similar expression can be written down for area evolution as well. We propose to include these details in the discussion section of the revised manuscript.

*3. Furthermore, it seems that the linear-response method would require a relatively long time-series of the area and volume evolution of each glacier in order to fit the parameters, which is often not available. I would like to see a discussion of the data requirements and feasibility for use in sea-level projections.*

In principle, no further tuning is required as the paramterisation of the linear response properties of mountain glaciers have been obtained with the help of the ensemble of SIA glaciers. One only needs a method to compute balance gradient ($\beta$), melt rate near terminus ($b_t$), and mean thickness ($h$) for each of the glacier to obtain the response properties. However, since the parameterisation is obtained for an ensemble of Himalayan glaciers, there is a need to validate the parameterisations for various other regions in the world where the mass-balance profiles or the typical glacier geometry could be different. That can be achieved with similar SIA-based tw-dimensional simulation of a large enough ensemble of glaciers.

To apply the model on a global scale, data on glacier area and volume, and their rate of changes are required at the initial epoch to start the model run. Additionally, a mass-balance model is required to compute $\beta$ and $b_t$ for each of the glacier. Another requirement is a prescription to determine the change in terminus elevation as volume and area changes. Most of these inputs are quite similar to that needed for any analogous scaling-based model (eg, Radic et al., 2007). Again, we shall include these details in the discussion section of the revised manuscript.

*4. Although it is true that there are a priori arguments for what the scaling exponent gamma should be, in practice it can be quite different, even for simulated glaciers (e.g., Radic et al., 2007). So for use of the scaling-based method as a statistical projection method it would be more fair, in order to compare to the linear-response method, to estimate the scaling exponent from the SIA runs. Although Radic et al. (2007, 2008)showed that the volume evolution over 100 years is not very sensitive to the exponent,this may be different for the 500-year simulations. Also, how were the constants of proportionality c determined for each glacier? In fact, we could expect that fitting the exponent and constant of*

*proportionality individually for each glacier, as would be possible if a sufficiently long time-series data were available for every glacier, would considerably reduce the bias of the scaling-based method.*

In the manuscript we followed the prescription of Bahr et al. (2015) and fitted for only $c$, setting $\gamma$ equal to its theoretical value. We agree that in practice $\gamma$ and $c$ both are to be fitted for. For our synthetic glaciers simulated with SIA such a fit led to best-fit $\gamma$ that was essentially the same as the theoretically predicted value - likely due to the idealisation involved in the models.

We have used a single fit-parameter $c$ for all glaciers (Bahr et al., 2015). However, we emphasise that scaling evolution only requires the value of $\gamma$, and does not require that of $c$ due to the assumed time-invariance (eq. 2 in the manuscript).

We disagree with the referee that fitting for $c$ and $\gamma$ for individual glaciers using available long-term observations can cure the bias in the scaling-based methods. It was clearly demonstrated in this manuscript that $c$ is time-dependent. Therefore a constant $c$ assumption would always lead to some bias in the long run. However, over shorter time scales, the suggested prescription may be useful - particularly if the drift in $c$ is negligible.

**Other issues:**

*1. The authors remove some glaciers from consideration in several parts of the paper, such as those that had fractional changes of more than 50% over 500 years, and those with response times higher than 300 years. Also, in another part of the paper, glaciers with large values of $\Delta A_\infty / A$ are removed, and another cut-off on $\Delta V_\infty / V$ is imposed. I don't see an adequate justification of why these were removed, and doing so biases the results.*

We agree with the reviewer that rejecting several modeled glaciers at various stages is confusing. The criticism is well taken and our arguments and proposed changes as detailed below.

First of all, the cut-off on fractional changes in area/volume (assumed to be 50% in the manuscript) are necessary as linear-response theory only works for relatively small deviations. We checked that our results is more or less independent of the actual value of the cut-off. For example, the model performance for the set of glaciers with fractional changes of less than 20% are shown in Figure 1 below.

We propose to remove the cut-offs used while fitting for glacier response properties (the excluded points were denoted by black circles in main fig 2, 3 and 5). This lead to small insignificant changes in the best-fit coefficients appearing in the expressions for linear-response properties.

A cut-off of $\tau_A < 300$ years (Please note that as $\tau_A > \tau_V$, it is enough to put a cut-off on only $\tau_A$) was applied as our simulation period is 500 years, and response times much larger than that may not be estimated accurately. The actual value of the cut-off is somewhat arbitrary though. For example, we had

[Figure]

Figure 1: A comparison of model performance for the set of 246 glaciers with fractional changes of less 20%.

verified that setting the cut-off at, say 500 years, does not affect the linear-response parameterisations within a few percent of so. The nature of model outputs is insensitive to the precise cut-off value (see Figure 2 below). However, to be on the safe side we have used a response-time cutoff of 300 years. It can

[Figure]

Figure 2: A comparison of model performance for the set of 582 glaciers with response time less than 500 years.

be seen Figure 3A below that without the best-fit $\tau_A$ for individual glaciers are likele overestimated when they are larger than 500 years or so. As a result the linear relationship between $\tau_A$ and $\tau_V$ breaks downhere. This is also confirmed by Figure 3B below, where we have shown the best-fit exponential decay lead to larger estimates for $\tau_A$ when a small part of the intial profile is fitted for. Based of these arguments, we believe an appropriate cutoff on response time - one that is less than 500 years or so - is necessary to obtain accurate parameterisation of the same.

We propose to include all the details mentioned here related to cutoffs applied as supplementary material with the revised manuscript.

[Figure]

Figure 3: (A) For glaciers with $\tau_A$ comparable or larger than simulation period of 500 years, best-fit $\tau_A$ values are overestimated. This leads to a breakdown of the linear relationship between $\tau_A$ and $\tau_V$. (B) As an illustration, we show fits to the timeseries of area for one of the 551 glacier over the first 200, 300, 400, and 500 years (solid lines) which obtain response time of $7 \times 10^6$, 557, 312, and 257 years, resepctively.

*2. "The minor differences are due to the time-invariant scaling assumption made here." Please clarify in more detail what is the difference between your derivations and those of Harrison (2001).*

The differences arise out of our use of scaling to relate $\Delta V$ and $\Delta A$, namely, $\Delta V = \gamma h \Delta A$, that is used to obtain the expression of $\tau_v = -(\frac{b_t}{\gamma h} + \beta)^{-1}$.

In contrast, Harrison et al. (2001) used $\Delta V = H \Delta A$ where $H$ is a "thickness scale" to get, $\tau_v = -(\frac{b_t}{H} + \beta)^{-1}$.

Since sensitivities are proportional to response time the same factor of $\gamma$ also appears in our expression of climate sensitivities. That is not the case for corresponding expressions of Harrison et al. (2001).

*3. In Fig. 1B, scaling the SIA results by 10 for visual comparison is confusing. It's also hard to distinguish which are the thick and thin lines.*

We proposed to replace the figure with the following one (see Figure 4 below) where we plotted results for 10 glaciers and did not scale the SIA results.

**Minor issues: ...**

We thank the referee for pointing out the typographical errors which we shall correct in the revised version of this manuscript.

[Figure]

Figure 4: Evolution of 10 randomly chosen glacier in the $V - A$ plane for SIA and scaling based models.

---

## Referee Comment (RC2) · Anonymous Referee #2 · 2 Mar 2020

**Review of 'Possible biases in scaling-based estimates of mountain-glacier contribution to the sea level' by Banerjee et al.**

Submitted to 'The Cryosphere Discussions', discussion started on 9 January 2020

In this manuscript, Banerjee and colleagues simulate the evolution of 551 glaciers from the Ganga basin (Himalaya). They start from glaciers in steady state resembling present-day glaciers and force them with a stepwise change in equilibrium line altitude until they evolve into a new steady state over a 500-yr time period. These simulations are performed with a V-A scaling method, with a linear response model and with a 2-D flow model based on the shallow ice approximation (SIA). The authors find that the V-A scaling method, when subject to a time-invariant scaling, underestimates the modelled future loss compared to the simulations performed with the SIA model. From this finding, they suggest that relying on V-A scaling is problematic for studies focusing on the future sea-level contribution from glaciers as this contribution is thus likely to be underestimated.

Some of the ideas put forward in this manuscript are interesting, but the manuscript has several problems (some of which are major in my opinion). My main concerns relate to:

[1] **State-of-the art**. The authors compare the outcome of V-A scaling with results from a SIA model. V-A has indeed been used in some important regional- to global studies in the past (e.g. Marzeion et al., 2012; Radić et al., 2014) due its computational efficiency. Moreover, with spatial estimates of ice thickness lacking for individual glaciers at the time, V-A methods offered a good alternative to estimate the volume of a glacier (and its changes through time). However, increasing computational performance and new glacier-specific inventories on e.g. ice thickness (Huss & Farinotti, 2012; Farinotti et al., 2019) and mass balance (e.g. Brun et al., 2017; Braun et al., 2019; Dussaillant et al., 2019; Zemp et al., 2019), now allow for far more sophisticated methods to simulate the dynamic evolution of glaciers. This includes methods based on imposing observed geometry changes in which the glacier geometry is explicitly accounted for (e.g. Huss & Hock, 2015; Rounce et al., 2020a, 2020b) and more recently also flowline models in which glacier dynamics (i.e. mass transfer within a glacier) are included when projecting glacier changes at regional to global scales (Maussion et al., 2019; Zekollari et al., 2019). When reading this manuscript, it seems like V-A scaling is a state-of-the art approach, and that you compare it to something more sophisticated (2-D SIA model). This comparison would have been very relevant a few years ago, when V-A scaling was state-of-the art (I am for instance thinking about the excellent work realized by Surendra Adhikari during his PhD; see e.g. Adhikari & Marshall, 2012), but has, in my opinion, lost some of its interest by now. With the new glacier-specific ice thickness estimates and other information derived from remote sensing becoming widely available (outlines, surface topography, ice thickness derived from this), the importance of V-A scaling methods is now strongly reducing and is likely to continue doing so so in the near future (see e.g. discussion by Haeberli, 2016). I do therefore have some reservations whether the 'The Cryosphere' is the ideal medium to share these (somewhat outdated?) findings. This concern is furthermore strengthened by my doubts about the experimental setup and the validity of your main conclusions as elaborated in the following points.

[2] **The experimental setup**:
   a. Comparing different methods and models is always quite complicated. This is especially the case when considering 'real' cases (glaciers with real geometries in your case). A study such as the one presented here would have greatly benefited from an idealized setup, which would have made comparisons more straightforward and allowed to disentangle differences between simulations obtained from V-A scaling and those relying on 2-D SIA modelling: see e.g. Leysinger Vieli and Gudmundsson (2004) and Adhikari and Marshall (2012). Here a 'selection' of

glaciers is considered, due to some 'problems' occurring when considering all glaciers in the region (see point 2b), which makes it even questionable how representative these are for this given region. With idealized glacier geometries, you could have explored the effect of glacier size, surface slope,... on the discrepancies between V-A based results and SIA modelled results more carefully.

b. Several arbitrary steps and decisions are made in the manuscript. A few examples of decisions that are hard to understand / seem not well funded:

   o l. 181-182: you exclude glaciers with a large change in area over the 500-year time period? Why? This seems arbitrary, but you must have a reason for this. Moreover, how this this influence your results? This makes the sample less representative...

   o l. 182-183: why do you exclude glaciers with long response times? Again, this makes your sample less representative (you probably exclude a certain type of glaciers, likely those that are gently sloping: see e.g. Haeberli & Hoelzle, 1995). Is this because these glaciers are not in steady state after 500 years? If so, you should simply run your experiments for longer and not exclude these glaciers.

   o Figure 1: you show '200 randomly chosen glaciers': why? Should show them all!

   o l.249-250: 'In fig. 2b, about 30 data points,...were not included in the fit': why? You mention something about possibly creating a bias in the linear fit in the next sentence, but I do not see where this would result from / what the problem could be.

c. The Setup of your SIA model is not fully clear.

   o You mention that for > 100 cases 'our algorithm for finding a steady-state similar to present extent did not converge or the final steady state glacier geometry was not realistic': how is this possible? How can a simple SIA solution not 'converge' to steady state (in fact, even analytical solutions may exist that do not even require running the SIA model to find the steady state: see e.g. Jouvet & Bueler, 2012)? And what do you consider 'not being realistic'? Which boundary conditions did you use to ensure mass conservation (e.g. to ensure specific ice-free regions do not become ice-covered)? You mention that mass conservation was monitored (l. 162-163): but how do you do this (this is not so straightforward to do...)? Did you check that the integrated SMB over your glacier is zero for the steady states (which it should be)? Would also be good if you could consider some benchmark experiments (e.g. Jarosch et al., 2013) to make sure your model is mass conserving.

   o Why do you randomly pick the values for the rate factor in Glen's flow law (not 'Glenn' + add a reference to the original studies, e.g. Glen, 1955)? The value of the rate factor will have a large influence on the local ice thickness and on thus the glacier volume. By picking this randomly: could be 'off' quite a lot from the 'reference/observed' volume of the glacier. Why do you not match this to the reference volume from every glacier that you have from Kraaijenbrink et al. (2017)? Is this also not problematic when working with single values for $c$ and $\gamma$ later in your analyses for all glaciers (e.g. for the best fits): you make some glaciers too thin and some too thick.

d. Lack of in-depth analyses. Often you seem to be perplexed by some findings yourself and leave important questions unanswered, which is unsatisfying for the reader. This questions the thoroughness of your approach, e.g.:

   o l. 186-187: '...we did not do a detailed glacier-by-glacier analysis of the reason behind the failure of the algorithm'... Well, you should do this! May be something intrinsically wrong with your setup (e.g. in terms of mass conservation, boundary conditions; see 2c). If this is the case, this is likely to have direct consequences for your results and for some of your conclusions...

   o l. 247-248: 'We do not have a clear explanation of this effect as yet': ...

- l.256: '**Again**, we do not have a theoretical argument for such a power-law behavior and did not explore this further here': ...
- l.304-305: '..., it remains to be investigated if the results described here depend on the regional characteristics of glaciers to some extent': ...

[3] The **main conclusion** drawn your manuscript, and which also appears in the title, is that using V-A scaling methods (with 'time-invariant scaling') are likely to underestimate the future sea level contribution from glaciers.

a. I am not sure that the material you presented is convincing enough to support this statement and that the experimental setup is adequate (see previous point).

b. Another major concern that I have is: if this would be the case: why do we not see this when comparing outcomes of V-A scaling estimates compared to more sophisticated methods relying on retreat parameterizations (Huss & Hock, 2015) or flowline models (Maussion et al., 2019)? The first phase of the GlacierMIP project (Hock et al., 2019), in which future large-scale glacier simulations from the literature were compared, did not reveal a tendency for V-A scaling methods to underestimate the contribution to sea-level rise (SLR). Also in the second phase of the GlacierMIP experiments, in which several ice dynamic (vs. V-A) were included and in which coordinated experiments were performed, no clear tendency can be seen when considering V-A scaling vs. methods in which the glacier geometry (and in some cases also ice dynamics) are explicitly considered. From the material at hand, I would rather tend to believe the outcomes from GlacierMIP than the main conclusions put forward here when it comes to the implications of using V-A scaling for future sea level projections.

c. You draw your main conclusion (that the loss from V-A scaling with time-invariant scaling is underestimated vs. SIA) from two steady states: an initial one and a final one. You present your results like transient results (e.g. in plots, when describing response times, in section 4.1. describing that *c* is time-dependent and decreases with time, in section 4.4.,....etc.), but in the end, **it boils down to the fact that the volume of the final steady state with time-invariant V-A scaling is 'too large'** (compared to the SIA). Due to this, the transient volume loss when evolving to this steady state is underestimated (always with respect to SIA results). **The main question that you thus need to address is: why is the V-A scaled final steady state too big?** I am not an expert in V-A scaling, but I would find it surprising that this issue has not been addressed in other V-A scaling studies and that no solutions to this problem have been formulated. In the end, from my understanding, what happens is that many glaciers that reduce in size lose their lowest part, which are often the most gently sloping parts of the glacier and where the highest ice thickness is thus found (in most ice thickness reconstructions this clearly appears, where in the end, a large part of the reconstruction results from the negative correlation between the surface slope and the local ice thickness; see Farinotti et al., 2017). It is thus to be expected that the V-A scaling that you use to create the initial steady state does not hold for the final one. This is something that would need to be explored in more detail, and for which studies in which the volume scaling also uses information from other glacier characteristics (e.g. the glacier slope) could be useful (Grinsted, 2013; Zekollari & Huybrechts, 2015; see e.g. Fig. 9a in the latter, which summarizes the main point made here).

[4] **Unclarities** in the manuscript. I found the text difficult to follow and quite often had to re-read sentences several times before being able to grasp their meaning. A few examples include:

a. l. 8-9: '..and validate them with results from scaling-based simulation of the ensemble of glacier'

b. l.84-85: '…are then empirically extended in order to obtain accurate parameterisations the linear-response properties of the SIA-simulated glaciers'

c. l.86-87: 'The linear-response model the long-term total shrinkage of glaciers as predicted by the scaling-based method (Radić et al., 2007), and the linear-response model are compared with the corresponding response'

d. ….etc. See also comments on specific sections below.

This makes it tedious to go through the manuscript. Furthermore, there a substantial number of grammatical errors, some of which (but not all) have already been pointed out by the first reviewer. Also, many figures cannot be interpreted/read independently, without having to refer to the caption. It would be good if all essential information (e.g. meaning of colors used, R^2 values, equations,…etc.) could be directly included in the figure.

Some other **comments for specific sections** (non-exhaustive list and not focusing on grammatical errors)

o 1. Introduction: 'methods solving the dynamical ice-flow equations' → 'numerical cost of such a computation on a global scale is prohibitive': well is not really the case anymore. In general: would be good to acknowledge regional- to global studies in which ice flow is explicitly accounted for (Clarke et al., 2015; Maussion et al., 2019; Zekollari et al., 2019).

o 1.2. Motivation for the present study: difficult to follow the first paragraph: be more specific when you refer to $c$ and $\gamma$ and do not continuously mix with other terminology 'time invariant scaling-based parameterisation', '…given the known violation of the time-invariant scaling assumption'.

o 2. Quite abstract and thus very difficult to go through for someone who is not an expert in V-A scaling. Could make it less technical by for instance adding some additional information that links the various parts.

o 3.1.: 2-dimensional SIA model:

 • l.152-154: where did you get the ice thickness from? From Kraaijenbrink et al. (2017) directly? As the ice thickness is quite crucial in your story (it determines the volume…), why did you not consider the consensus estimate of Farinotti et al. (2019), which is freely available?

 • SIA: refer to the original work by Hutter (1983) also.

 • You neglect basal sliding (l. 161). Justification? Could refer to other studies where this is done, like e.g. Gudmundsson (1999) and Clarke et al. (2015).

 • l.168-178: this is related to the SMB, which you apply in all cases (i.e. also for the linear-response model and the V-A scaling, right?). Not sure this section is correctly placed here in the '2-dimensional SIA model' section.

 • l.183: through several exclusion you keep 68% of the initial glaciers… How much does this represent in terms of glacier volume and glacier area when compared to the total glacier sample?

 • l.188-196: you explain some simplifications related to debris cover, avalanche and sliding have been made and that this may influence your results. Well, you have made much larger simplifications than this: e.g. linear SMB profiles with strongly imposed max. SMB, steady state assumptions for glaciers,… → not even worth mentioning these more detailed simplifications in my opinion. With all these simplifications, would have been better to opt for idealized setup likely (see main comment 2a).

 • l. 194: 'These simplifications do not weaken our study': not sure you can judge on this yourself…

o Section 3.2.:

 • l.204: 'was fixed at… because…': don't understand the causality (i.e. link between cause and consequence).

- Figure 1: SIA-derived volumes are scaled by a factor 10: why? Does not really make sense and unclear when just looking at the figure without reading the caption... Axes should be correct in the figure and not only for a part of the data you show.. Also illustrates the unclarity in the figures mentioned in main comment 4 (problem that figures cannot be interpreted without referring to their caption).
- Section 3.3.:
  - l.208-210: complicated way to say that you consider e-folding time scales. Would reformulate this and add references for this to e.g. Leysinger Vieli & Gudmundsson (2004).
- Section 4.1.:
  - l.225: $V=cA^{1.286}$: not sure I understand. Does this statement apply for the initial and/or final steady state volumes? And can all the volumes be described with this single relationship? Is the fact that quite different rate factors are used not a problem for this (see main comment, point 2c)?
  - l.227 + l.230 + l.232: here you mention that $c$ is time dependent. Not sure you can say that it is time dependent: simply results from the fact that final steady state volume for V-A scaling is 'overestimated' (vs. SIA). As a result the evolution to this steady state is different. See main comment 3c for this.
  - l.235-237: relates to main comment 3c again. If you do not modify the V-A scaling, then problems will arise when considering the same glacier that is much smaller in a warmer climate (when rising the ELA in your case): you typically lose the lower parts where most volume is and volume will thus be 'overestimated'. Is this not accounted for in some way in future glacier evolutions based on V-A scaling? As a part of this discussion, studies in which V-A scaling is extended with other glacier characteristics (such as the surface slope; Grinsted, 2013; Zekollari & Huybrechts, 2015) would be good to include. Such relationships which could prove to remain valid over time, even without changing scaling and exponents.
- Section 4.2.:
  - l. 243: 'This is exactly what is seen in Fig. 2b, which shows...': I cannot directly see this...
  - l.244: 'change in c to the tune of ~13%': what does this mean?
  - l.255: 'The above figure': will depend where your figure comes in final manuscript...
- Section 4.3:
  - I was wondering what the point is that you want to make with this section? It is known from literature that volume responds faster than area (e.g. Oerlemans, 2001; Leysinger Vieli & Gudmundsson, 2004).
  - l.260-264: relationship between volume and area response times. How does this compare to the relationship others have found in the literature?
- Section 4.4.:
  - l.271-272: 'with most of the changes taking place during the first couple of centuries': this is not a result/finding.. This directly results from the e-folding time-scale when forcing a steady state glacier with an instantaneous forcing in SMB.
  - l.273: 'underestimates the long-term change': not about reaction/response. This is direct consequence of fact that final steady state volume is too large (see main comment 3c)
  - l.279-280: '...suggests that there might be significant negative biases of mountain glacier contribution to sea-level rise as computed by scaling-based methods' (+ section 4.5, l.300-302): well, do not see this in GlacierMIP phase 1+2... Is a very strong statement to make and should be sure that it is well-founded.
- Section 4.5:

- l.296-297: 'More detailed studies that relaxes some of the above mentioned assumptions are needed...': not sure what you mean by this. Would also make sense that you dig into this: e.g. by focusing on real transient response vs. comparing two steady states (what you do now and then translate into an analysis of the transient response resulting from this: see main comment 3c).
- l.299: 'intruding more scatter in the fits': what does this mean?

o Summary and Conclusions:
  - l.309-310: scale factor reduces over time. Well, not sure the time dimension is adequate here. Boils down to having a final steady state that would require a smaller value for $c$: see main comment 3c.
  - l.324: computational efficiency. OK, still important, but is not really a limitation anymore, due to which V-A scaling becomes less important (and also driven by the release of new datasets with regional- to global spatial coverage at individual glacier level: see main comment 1).
  - Code availability: for which models is the code available? Seems to suggest that the SIA code is not available. Not sure if this fully agrees with the policies of The Cryosphere: see www.the-cryosphere.net/about/data_policy.html

**References**

Adhikari, S., & Marshall, S. J. (2012). Glacier volume-area relation for high-order mechanics and transient glacier states. *Geophysical Research Letters*, *39*, L16505. https://doi.org/10.1029/2012GL052712

Braun, M. H., Malz, P., Sommer, C., Farias-Barahona, D., Sauter, T., Casassa, G., et al. (2019). Constraining glacier elevation and mass changes in South America. *Nature Climate Change*, *9*, 130–136. https://doi.org/10.1038/s41558-018-0375-7

Brun, F., Berthier, E., Wagnon, P., Kääb, A., & Treichler, D. (2017). A spatially resolved estimate of High Mountain Asia glacier mass balances from 2000 to 2016. *Nature Geoscience*, *10*, 668–673. https://doi.org/10.1038/NGEO2999

Clarke, G. K. C., Jarosch, A. H., Anslow, F. S., Radić, V., & Menounos, B. (2015). Projected deglaciation of western Canada in the twenty-first century. *Nature Geoscience*, *8*, 372–377. https://doi.org/10.1038/ngeo2407

Dussaillant, I., Berthier, E., Brun, F., Masiokas, M., Hugonnet, R., Favier, V., et al. (2019). Two decades of glacier mass loss along the Andes. *Nature Geoscience*. https://doi.org/10.1038/s41561-019-0432-5

Farinotti, D., Brinkerhoff, D. J., Clarke, G. K. C., Fürst, J. J., Frey, H., Gantayat, P., et al. (2017). How accurate are estimates of glacier ice thickness? Results from ITMIX, the Ice Thickness Models Intercomparison eXperiment. *The Cryosphere*, *11*, 949–970. https://doi.org/10.5194/tc-11-949-2017

Farinotti, D., Huss, M., Fürst, J. J., Landmann, J., Machguth, H., Maussion, F., & Pandit, A. (2019). A consensus estimate for the ice thickness distribution of all glaciers on Earth. *Nature Geoscience*. https://doi.org/10.1038/s41561-019-0300-3

Glen, J. W. (1955). The Creep of Polycrystalline Ice. *Proceedings of the Royal Society A: Mathematical, Physical and Engineering Sciences*, *228*(1175), 519–538. https://doi.org/10.1098/rspa.1955.0066

Grinsted, A. (2013). An estimate of global glacier volume. *The Cryosphere*, *7*(1), 141–151. https://doi.org/10.5194/tc-7-141-2013

Gudmundsson, G. H. (1999). A three-dimensional numerical model of the confluence area of Unteraargletscher, Bernese Alps, Switzerland. *Journal of Glaciology*, *45*(150), 219–230. https://doi.org/10.3189/002214399793377086

Haeberli, W. (2016). Brief communication: On area- and slope-related thickness estimates and volume calculations for unmeasured glaciers. *The Cryosphere Discussions*, (January), 1–18. https://doi.org/10.5194/tc-2015-222

Haeberli, W., & Hoelzle, M. (1995). Application of inventory data for estimating characteristics of and regional climate-change effects on mountain glaciers: a pilot study with the European Alps. *Annals of Glaciology*, *21*, 206–212. https://doi.org/10.3189/S0260305500015834

Hock, R., Bliss, A., Marzeion, B., Giesen, R. H., Hirabayashi, Y., Huss, M., et al. (2019). GlacierMIP – A model intercomparison of global-scale glacier mass-balance models and projections. *Journal of Glaciology*. https://doi.org/10.1017/jog.2019.22

Huss, M., & Farinotti, D. (2012). Distributed ice thickness and volume of all glaciers around the globe. *Journal of Geophysical Research: Earth Surface*, *117*(4), F04010. https://doi.org/10.1029/2012JF002523

Huss, M., & Hock, R. (2015). A new model for global glacier change and sea-level rise. *Frontiers in Earth Science*, *3*, 1–22. https://doi.org/10.3389/feart.2015.00054

Hutter, K. (1983). *Theoretical Glaciology*. Dordrecht: Reidel Publ. Co.

Jarosch, A. H., Schoof, C. G., & Anslow, F. S. (2013). Restoring mass conservation to shallow ice flow models over complex terrain. *The Cryosphere*, *7*, 229–240. https://doi.org/10.5194/tc-7-229-2013

Jouvet, G., & Bueler, E. (2012). Steady, shallow ice sheets as obstacle problems: well-posedness and finite element approximation. *Journal of Applied Mathematics*, *72*(4), 1292–1314. https://doi.org/10.1137/110856654

Kraaijenbrink, P. D. A., Bierkens, M. F. P., Lutz, A. F., & Immerzeel, W. W. (2017). Impact of a global temperature rise of 1.5 degrees Celsius on Asia's glaciers. *Nature*, *549*, 257–260. https://doi.org/10.1038/nature23878

Leysinger Vieli, G. J.-M. C., & Gudmundsson, G. H. (2004). On estimating length fluctuations of glaciers caused by changes in climatic forcing. *Journal of Geophysical Research*, *109*(F01007), F01007. https://doi.org/10.1029/2003JF000027

Marzeion, B., Jarosch, A. H., & Hofer, M. (2012). Past and future sea-level change from the surface mass balance of glaciers. *The Cryosphere*, *6*(6), 1295–1322. https://doi.org/10.5194/tc-6-1295-2012

Maussion, F., Butenko, A., Champollion, N., Dusch, M., Eis, J., Fourteau, K., et al. (2019). The Open Global Glacier Model ( OGGM ) v1.1. *Geoscientific Model Development*, *12*, 909–931. https://doi.org/10.5194/gmd-12-909-2019

Oerlemans, J. (2001). *Glaciers and climate change*. Dordrecht: A. A. Balkema.

Radić, V., Bliss, A., Beedlow, A. C., Hock, R., Miles, E., & Cogley, J. G. (2014). Regional and global projections of twenty-first century glacier mass changes in response to climate scenarios from global climate models. *Climate Dynamics*, *42*(1–2), 37–58. https://doi.org/10.1007/s00382-013-1719-7

Radić, V., Hock, R., & Oerlemans, J. (2007). Volume-area scaling vs flowline modelling in glacier volume projections. *Annals of Glaciology*, *46*, 234–240. https://doi.org/10.3189/172756407782871288

Rounce, D., Hock, R., & Shean, D. (2020a). Glacier mass change in High Mountain Asia through 2100 using the open-source Python Glacier Evolution Model (PyGEM). *Frontiers in Earth Science*, *7*, 331. https://doi.org/10.3389/feart.2019.00331

Rounce, D. R., Khurana, T., Short, M., Hock, R., Shean, D., & Brinkerhoff, D. J. (2020b). Quantifying parameter uncertainty in a large-scale glacier evolution model with a Bayesian model: Application to High Mountain Asia. *Journal of Glaciology*.

Zekollari, H., Huss, M., & Farinotti, D. (2019). Modelling the future evolution of glaciers in the European Alps under the EURO-CORDEX RCM ensemble. *The Cryosphere*, *13*, 1125–1146. https://doi.org/10.5194/tc-13-1125-2019

Zekollari, H., & Huybrechts, P. (2015). On the climate-geometry imbalance, response time and volume-area scaling of an alpine glacier: insights from a 3-D flow model applied to Vadret da Morteratsch, Switzerland. *Annals of Glaciology*, *56*(70), 51–62. https://doi.org/10.3189/2015AoG70A921

Zemp, M., Huss, M., Thibert, E., Eckert, N., McNabb, R., Huber, J., et al. (2019). Global glacier mass balances and their contributions to sea-level rise from 1961 to 2016. *Nature*, *568*, 368–386. https://doi.org/10.1038/s41586-019-1071-0

---

## Author Comment (AC2) · 25 Mar 2020

**Reply to the comments by the anonymous reviewer**

We thank the reviewer for the critical comments that has forced us to do our numerical analysis which, as we try to argue below, have put our results on firmer ground. These criticisms will be of immense help in improving the clarity in the subsequent versions of this manuscript. We are also thankful to the reviewer for pointing us to several important references.

      Based on the reviewer's inputs, we have redone our analysis. To summarise the updated results,

1. We have now simulated the response of 810 glaciers out of the total 814 glaciers in the Ganga Basin that are larger than 2 km$^2$.

2. The transient simulation time was extended to 1000 years and the cut off on response time is set at 500 years. Only 9 glaciers, covering 2% of the total area, were omitted this time.

3. After removing glaciers with more than 50% change at the 500 year mark, our final ensemble now includes 703 glaciers.

4. The results presented in the discussion paper remain essentially even with this larger set of glaciers .

5. We checked that the biases in the scaling model for a set of idealised 1d glaciers with the same linear bedrock, mass-balance gradient and Glen's flow-law constant.

More details are provided in out point-by-point replies (in black) to the comments by the anonymous reviewer 2 (in red). For ease of reference, our replies below are marked with [1], [2], … .

In this manuscript, Banerjee and colleagues simulate the evolution of 551 glaciers from the Ganga basin (Himalaya). They start from glaciers in steady state resembling present-day glaciers and force them with a stepwise change in equilibrium line altitude until they evolve into a new steady state over a 500-yr time period. These simulations are performed with a V-A scaling method, with a linear response model and with a 2-D flow model based on the shallow ice approximation (SIA). The authors find that the V-A scaling method, when subject to a time-invariant scaling, underestimates the modelled future loss compared to the simulations performed with the SIA model. From this finding, they suggest that relying on V-A scaling is problematic for studies focusing on the future sea-level contribution from glaciers as this contribution is thus likely to be underestimated. Some of the ideas put forward in this manuscript are interesting, but the manuscript has several problems (some of which are major in my opinion).

[1] At the outset, we respond to the main issue that was raised by the reviewer throughout his report. It was suggested that the biases in scaling models as pointed out in our paper, stem from an (known) inability of these models to capture the magnitude of the change in steady-state volume in response to a step change (in other words, an underestimation of the climate sensitivity of volume/area in scaling methods). The reviewer conjectured that scaling methods do not capture the effects of typically gentler slope and thicker ice near glacier termini and thus, underestimates climate sensitivity. It was suggested that alternative scaling formulations that include slope-dependent corrections should be able to take care of such biases.

First, a similar bias is seen even for a set of glaciers with the same linear bedrock in 1-d (fig. R1). All of these glaciers have the same linear mass-balance profile, and the same rate constant as well. The pattern of the deviations is, in fact, very similar to that in the main figures 1 and 6. Therefore it is unlikely that slope dependent correction to scaling can take care of the biases that we have pointed out. Second, the reviewer did not consider a critical limitation of the scaling model that it, by construction, implies an equality of area and volume response time. This too, cannot be corrected by slope dependent correction.

As demonstrated in our manuscript, the discrepancies in scaling model results (main fig 1,6 and fig R1) are due to the following general limitations:

A) An underestimation of climate sensitivity

We provided numerical evidence in favour of a higher climate sensitivity of glaciers simulated with SIA in main figure 3 and fig R1 above.

(This is something that the reviewer also agrees to. However, the reviewer's explanations based on slope effect cannot be the full explanation as is clear from fig R1.)

B) Area and volume response times are predicted to the same

We provided corresponding theoretical proof in section 2.3 (eq. 10), and numerical evidence in main fig 4.

(Due to this limitation, the transient trajectory of glaciers simulated with scaling models shows a linear trend in V-A log-log plot. In contrast, SIA-derived trajectories are non-linear due to relatively faster changes in volume right after the ELA change. This is apparent from fig R1 and main figures 1 and 6.)

Apart from demonstrating the above limitations of scaling models, the thrust of our manuscript is to come up with an alternative linear-response method that minimises above biases. Of course, we do not claim that there are no other zero-dimensional methods that can be used to achieve the same goal.

We shall include fig R1 in supplementary, and add a clear discussion of the above points.

[Figure]

Fig. R1: (top) A comparison of the scaling-based (purple line) and SIA-based (green line) evolution of 9 idealised 1-d glaciers on linear bedrock (slope 0.1) and linear mass-balance profile for 100 years, after a step change in ELA by 50 m. (Bottom) The evolution of total area and total volume with scaling and SIA based models.

**[1] State-of-the art.**

The authors compare the outcome of V-A scaling with results from a SIA model. V-A has indeed been used in some important regional-to global studies in the past (e.g. Marzeion et al., 2012; Radić et al., 2014) due its computational efficiency. Moreover, with spatial estimates of ice thickness lacking for individual glaciers at the time, V-A

methods offered a good alternative to estimate the volume of a glacier (and its changes through time). However, increasing computational performance and new glacier-specific inventories on e.g. ice thickness (Huss & Farinotti, 2012; Farinotti et al., 2019)and mass balance (e.g. Brun et al., 2017; Braun et al., 2019; Dussaillant et al., 2019; Zemp et al., 2019), now allow for far more sophisticated methods to simulate the dynamic evolution of glaciers. This includes methods based on imposing observed geometry changes in which the glacier geometry is explicitly accounted for(e.g. Huss & Hock, 2015; Rounce et al., 2020a, 2020b)and more recently also flowline models in which glacier dynamics (i.e. mass transfer within a glacier) are included when projecting glacier changes at regional to global scales (Maussion et al., 2019; Zekollari et al., 2019). When reading this manuscript, it seems like V-A scaling is a state-of-the art approach, and that you compare it to something more sophisticated (2-D SIA model). This comparison would have been very relevant a few years ago, when V-A scaling was state-of-the art (I am for instance thinking about the excellent work realized by Surendra Adhikari during his PhD; see e.g. Adhikari & Marshall, 2012), but has, in my opinion, lost some of its interest by now. With the new glacier-specific ice thickness estimates and other information derived from remote sensing becoming widely available (outlines, surface topography, ice thickness derived from this), the importance of V-A scaling methods is now strongly reducing and is likely to continue doing so so in the near future (see e.g. discussion by Haeberli, 2016). I do therefore have some reservations whether the 'The Cryosphere' is the ideal medium to share these (somewhat outdated?) findings. This concern is furthermore strengthened by my doubts about the experimental setup and the validity of your main conclusions as elaborated in the following points.

[2] We apologise for not mentioning the regional-scale glacier-change studies using various approximate descriptions of ice-flow (Rounce et al, 2020a, 2020b, Zekollari et al. 2019, Clarke et al, 2015, ...). We shall correct this error. We shall also refer to probabilistic and data-based methods of computing future sea-level rise.

As far as computation of future sea-level contribution is concerned, we stand by our statement that "The existing global-scale estimates of the mountain-glacier contribution to sea-level rise mostly rely on low-dimensional approximate parameterisations of the glacier dynamics (van de Wal and Wild, 2001; Raper and Braithwaite, 2006; Hirabayashi et al., 2010; Radić and Hock,2011; Slangen and van de Wal, 2011; Marzeion et al., 2012; Giesen and Oerlemans, 2013; Huss and Hock, 2015; Hock et al, 2019). Several of these parameterisations are based on an statistical area-volume (or area-volume-length) scaling relation for any set of mountain glaciers." A majority of the available estimates of the contribution of glaciers to future sea-level rise is scaling based as of now. The state-of-the-art large-scale glacier models referred to in the

reviewer's comment have not yet been employed for sea-level rise computations as far as we know.

We do agree with the reviewer that in the near future one/two/three-dimensional ice-flow models are likely to be used more often for sea-level rise predictions. However, we believe that investigations of the limitations/biases of scaling-based models are still needed, as most of the existing global estimates do rely on such models. For example, 5 out of the 6 models in the intercomparison study by Hock et al. (2019) use some form of scaling. Recent studies of the global-scale vulnerability to sea-level rise continue to utilise results from scaling-based models - e.g., Kulp and Strauss (2019) used scaling-based estimates of glaciers' contribution to sea-level rise by Marzeion et al. (2012). Studies like the present one may be useful in identifying biases in such studies. For example, our analysis indicates that the multi-model mean sea-level change by 2100 as presented in Hock et al (Table 3, 2019) is likely to have a negative bias (This specific point is discussed in detail later in this document).

We do not agree with the reviewer's point-of-view that simpler effective models go outdated as higher-order dynamics become computationally feasible. In several (if not most) branches of science, a hierarchy of models with varying degrees of complexity coexists. In fact, the low-complexity models are often useful in aiding theorecital understanding/development. The need for critical investigations and development of simple 0-d models of glacier dynamics, we believe, cannot be overemphasised. That is the motivation behind the present study, where we not only point out possible biases of scaling-based models, but also present an alternative linear-response model that reduces above biases.

**[2] The experimental setup:**

a. Comparing different methods and models is always quite complicated. This is especially the case when considering 'real' cases (glaciers with real geometries in your case). A study such as the one presented here would have greatly benefited from an idealized setup, which would have made comparisons more straightforward and allowed to disentangle differences between simulations obtained fromV-A scaling and those relying on2-D SIA modelling:see e.g. Leysinger Vieli and Gudmundsson (2004)and Adhikari and Marshall (2012). Here a 'selection' of glaciers is considered, due to some 'problems' occurring when considering all glaciers in the region (see point 2b), which makes it even questionable how representative these are for this given region. With idealized glacier geometries, you could have explored the effect of glacier size, surface slope,...on the discrepancies between V-A based results and SIA modelled results more carefully.

[3] The discrepancy between scaling based and SIA based models are quite general in nature, and therefore, are expected to be present for any slope, geometry, mass-balance profile, rate-constant etc. While the magnitude of the bias is likely to depend on some of these factors as suggested by the reviewer, our aim in this paper is not to analyse such dependencies. The objective of the present study is to demonstrate that,

*1) The above bias could be significant for a set of real glaciers when a certain type of scaling models are used for predicting long-term glacier change or sea-level rise.*

*2) A linear-response model calibrated using 2d SIA outputs can be a viable alternative.*

Our experimental design is more suited for the above purpose than one using idealised geometries as suggested by the reviewer.

Also, our paper contains three main results:

A) The central assumption of a time-dependent scale factor is violated by SIA-simulated glaciers (main fig. 1),

B) A scaling-based model underestimates the climate sensitivity and response time of glaciers in comparison with SIA (main figs. 2 and 4) and assumes an equality of area and volume response time, and

C) a suitably calibrated linear-response model is free of the above low bias (main fig. 6).

A simplified glacier geometry may be useful to highlight the differences between SIA and scaling-based simulations (i.e. results A and B above), but it does not help in obtaining parameterisations of the linear-response properties and testing the performance of the corresponding model for a set of realistic glaciers (result C above).

As shown in fig R1, the same deviations of scaling model is indeed seen for idealised 1-d glaciers having the same bedrock slope - just as suggested by the reviewer. This does prove the general nature of the limitations of scaling-based models and supports our results obtained from simulations of 2-d glaciers.

A discussion about the omitted glaciers and evidence in favour of the representativeness of the selected glaciers are presented later (replies [5] and [6]). Also, now we have repeated our computation for the 814 glaciers, and now have 703 glaciers in the final ensemble (reply [9]). In the updated version of main fig. 6 below (fig R2) that uses the present ensemble of 703 glaciers obtain similar results as that from the earlier set of 551 glaciers.

[Figure]

Figure R2: The evolution of the total glacier volume (A), and (B) area for the ensemble of 703 glaciers simulated with three different methods, namely, SIA, scaling and linear-response model, are shown with orange, red, and blue solid lines, respectively.

b. Several arbitrary steps and decisions are made in the manuscript. A few examples of decisions that are hard to understand / seem not well funded:

[4] We describe the rationale for each of the steps below.

ol. 181-182: you exclude glaciers with a large change in area over the 500-year time period? Why? This seems arbitrary, but you must have a reason for this. Moreover, how this this influence your results? This makes the sample less representative …

[5] By definition, a linear-response model is only applicable when fractional changes are small (e.g. Oerlamans, 2005). So to apply 1) SIA, 2) scaling, and 3) linear-response models on the same set of glaciers, we have to exclude glaciers with large changes that are not described by linear- response models.

The threshold of 50% change that is used to exclude glaciers with large change, is indeed arbitrary. However, we confirm that our results do not depend on the specific value of the threshold chosen as (Fig. R3 below).

[Figure]

Figure R3. A comparison of the volume and area evolution computed with the three models for the set of glaciers with less than 20% change.

We had checked the representativeness of the selected 551 glaciers. As shown below, the set of 551 glaciers considered has reasonably similar distributions of area, and mean slope when compared to corresponding distributions for the full set of 814 glaciers.

Motivated by the above criticism, we have rerun our simulations, and included 703 out of the total 814 glaciers in our analysis (reply [9]).

[Figure]

[Figure]

Figure R4. Frequency distribution of glacier area (left column) and slope (right column) for all the glaciers and the selected ones. The top row shows the comparison for the set of 551 selected glaciers analysed in the discussion paper. The bottom panel shows the same for the 703 glaciers that are being used now in the updated simulations.

ol. 182-183: why do you exclude glaciers with long response times? Again, this makes your sample less representative (you probably exclude a certain type of glaciers, likely those that are gently sloping: see e.g. Haeberli & Hoelzle, 1995). Is this because these glaciers are not in steady state after 500 years? If so, you should simply run your experiments for longer and not exclude these glaciers.

[6] The slope distribution of all the 814 glaciers and that of the selected 551 glaciers are quite similar as shown in fig. R4 above, indicating that the chosen glaciers do form a representative set.

We do not require the simulated glaciers to reach a steady state. However, the response time has to be smaller than the simulation period (please see fig 3 in our response to reviewer 1).

Based on the above criticism, we have now extended the runtime of the transient simulations to 1000 years. This allows for setting the response time cutoff at 500 years. This excludes 10 glaciers that cover only about 2.5% of the total area.

We shall include figs. R3 and R4 in the supplementary.

o Figure 1: you show '200 randomly chosen glaciers': why? Should show them all!

[7] The figure with all the glaciers (and without the factor-of-10 scaling) is shown below (fig R5). With all the glaciers included the figure becomes somewhat cluttered. However, based on the above suggestion we shall show all the glaciers now to avoid

confusion, and shall include another figure with a few glaciers the supplementary (Fig. R5).

[Figure]

Figure R5: (left) Updated version of figure 1b with all 703 glaciers shown. (right ) the same plot, but for a random selection of 8 glaciers. In both the plots, purple (blue) line denotes scaling (SIA) results.

o l.249-250: 'In fig. 2b, about 30 data points,...were not included in the fit': why? You mention something about possibly creating a bias in the linear fit in the next sentence, but I do not see where this would result from / what the problem could be.

[8] Based on comments from both the reviewers, we have removed all such cut-offs used while fitting for the glacier response properties (the excluded points in main figs. 2, 3, and 5). This leads to small changes in the best-fit coefficients in the expressions for linear-response properties, but does not impact our basic conclusions. With the present set of 703 glaciers our results for linear response properties are as follows,

$$\frac{\Delta V_\infty}{V} = (1.93 \pm 0.02)\frac{\Delta A_\infty}{A}; \quad \frac{\Delta V_\infty}{V} = (1.71 \pm 0.03)\alpha^*;$$

$$\tau_V = (0.687 \pm 0.004)\tau_A; \quad \tau_A = (2.56 \pm 0.04)\tau^*$$

(please see the discussion paper for the notation)

c. The Setup of your SIA model is not fully clear.

oYou mention that for > 100 cases 'our algorithm for finding a steady-state similar to present extent did not converge or the final steady state glacier geometry was not realistic': how is this possible? How can a simple SIA solution not 'converge' to steady state(in fact, even analytical solutions may exist that do not even require running the

SIA model to find the steady state: see e.g. Jouvet & Bueler, 2012)? And what do you consider 'not being realistic'?

[9] First of all, in response to the above criticism, we have now included 810 out of total 814 glaciers in our analysis as discussed below.

In several of the glaciers that were omitted in the discussion paper, the problems were due to incorrect glacier boundaries in RGI6.0. These are mostly due to mapping errors where multiple glaciers have been merged fully or partially, parts of a glacier have in excluded, or debri-covered parts have been missed. In few other cases, a noisy bedrock led to a violation of ice-conservation (fig. R5). It is a well known problem with SIA that it violates mass conservation in regions with rugged topography. We performed a 3x3 moving window smoothing of berock to avoid this issue, before starting the simulation with an ice-free initial state. A few glaciers where such non-conserving behaviour was present could be identified as we tracked ice conservation explicitly (reply [10] below). At present there are two glaciers where violation of mass conservation is seen.

For a few glaciers for which a steady-state ELA could not be found,  it was a limitation of the simulation time. For each of the 814 glaciers,  we vared ELA to find a steady state with extent similar to that of the RGI6.0 glacier outline. This step was numerically expensive as for each of the trial values of ELA we needed to run the model long enough to check if a steady state is reached. There is no intrinsic issue with our implementation of SIA that prevents steady state, it is only related to runtime of the ELA tuning algorithm.   Earlier we did not take up the task of finding here primarily because existing algorithm was able to simulate 694 out of all the 814 glaciers, yielding an ensemble that is large enough to test the performance of scaling-based and linear response models.  The fact that  the final set of 551 glaciers have very similar slope and area distributions compared to the set of all 814 glaciers supports that claim (fig R3).

Responding to the above criticism, we have now rerun the simulations for all 814 glaciers, and obtained  initial steady state for 811 out of them. In the remaining 3 glaciers, two showed conservation error due to steep bedrock and were removed (Fig R6). For the other one, a steady state could not be found even with extended run.

[Figure]

Fig R6: One of the two glaciers (RGIID_15.04060) where ice conservation was violated due to noisy bedock (top plot) near the terminus leading to thick ice there (bottom plot).

There was another glacier, where part of the debris-covered ablation zone of a tributary was absent in the RGI outline and that lead to very thick ice in the truncated tributary (fig R7). This glacier was also removed. That left us with a total of 810 glaciers where response to step-change in ELA was simulated.

[Figure]

[Figure]

Figure R7: The outline of the glacier Rgi-15.04060 overlain on googl-earth (left) and the corresponding simulated ice thickness map. The RGI outline has incorrectly truncated debris-covered tributaries on the left. This blocks the ice-flow path and thickens the ice there in the SIA simulation. This glacier is not considered in our analysis.

After applying a criterion of less than 50% change (see reply [5]) and a 500 year cut-off on response time (reply [6]), we now have a total of 703 glaciers in our final set covering 89% of the total area.

Which boundary conditions did you use to ensure mass conservation (e.g. to ensure specific ice-free regions do not become ice-covered)? You mention that mass conservation was monitored (l. 162-163): but how do you do this (this is not so straightforward to do...)? Did you check that the integrated SMB over your glacier is zero for the steady states (which it should be)?

[10] A no-flux boundary condition was used as discussed in reply [9] above. The domain boundary for each of the glaciers using the RGI 6.0 boundary.  We perform a 3x3 moving-window centrally-weighted smoothing of the berock before starting the simulation with an ice-free initial state to minimise cases where ice-conservation is violated.

   We implement a straightforward algorithm to check mass conservation over the domain which implies,

   *Total ice present =*
      *Total accumulation over the simulation period*
    *− Total melting over the glacier simulation period*

*− fluxes out of the glacier boundary into the ice-free part of the domain.*

Each of the terms in the above equation was computed numerically at every time step. It was confirmed that ice is conserved at every time step during the transient evolution up to a fractional error of ~$O(10^{-9})$ for 812 out of 814 glaciers (e.g., Fig R8). Only on two glaciers conservation was violated due to noisy bedrock (e.g., Fig R6). These two were left out of our analysis.

We confirm that the total accumulation equals total ablation in steady state.

[Figure]

Figure R8: The top panel shows thickness maps of the initial (left) and final (right) states of a randomly chosen glacier (RGIid-15.07168). The color scale denotes ice thickness in km. The bottom panel shows the variation of the  cumulative accumulation, the cumulative ablation and the total volume as a function of time after the step change in ELA.

We shall provide ice-thickness maps (eg., fig R8), ice-conservation plots and fits to transient area/volume evolution for all the 814 simulated glaciers in the supplementary.

Would also be good if you could consider some benchmark experiments (e.g. Jarosch et al., 2013)to make sure your model is mass conserving.

[11] As explained in reply [10], checking for ice conservation is relatively  straightforward in our implementation, and at each time step ice was conserved up to a fractional error of the order of $10^{-9}$.

o Why do you randomly pick the values for the rate factor in Glen's flow law (not 'Glenn' + add a reference to the original studies, e.g. Glen, 1955)? The value of the rate factor will have a large influence on the local ice thickness and on thus the glacier volume. By picking this randomly: could be 'off' quite a lot from the 'reference/observed' volume of the glacier. Why do you not match this to the reference volume from every glacier that you have from Kraaijenbrink et al. (2017)?

[12] A wide range of values of rate factors has been used to model Himalayan glaciers. Typically, it is used as a tuning factor to obtain a good match with the observed velocity and thickness profiles. We have not done any tuning as our objective is limited  to comparing the performance of the three models for the same set of glaciers with realistic geometries. Without such tuning our modeled volume is likely to have bias, but that does not interfere with our plan of comparing the three methods for the exact same set of model glaciers. Of course, we agree that for accurate prediction of glacier mass loss in the Himalaya such a tuning is a necessity.

Choosing the rate constant randomly from a wide range ensures that our results are not specific to a particular value of the same. The variability of rate constant, glacier geometry and mass balance profile lead to scatter in
area-volume scaling plot for the modelled glacier (main figure 1). Such scatter is expected for a set of real glaciers as well. The variability in rate factor, balance gradient and bedrock geometry allows us to test the performance of scaling and linear-response models for a more realistic situation.

Is this also not problematic when working with single values for c and g later in your analyses for all glaciers (e.g. for the best fits): you make some glaciers too thin and some too thick.

[13] As clearly explained by Bahr et al (2015), the scaling law does not apply to a single glacier. It only holds statistically for an ensemble of glaciers and over/underestimation of thickness for individual glaciers cannot be avoided. According to the authors, c may

vary with time but $\gamma$ does not for a given set of glaciers. We have followed their prescription.

d. Lack of in-depth analyses. Often you seem to be perplexed by some findings yourself and leave important questions unanswered, which is unsatisfying for the reader. This questions the thoroughness of your approach, e.g.:

[14] We have obtained several numerical results here that do not have a clear theoretical explanation in the literature. While we could explain some of those results here, we carefully pointed out each of the cases where we could not. In all the instances listed by the reviewer except the first one, we were attempting to distinguish between the numerical results that are supported by theoretical arguments, and ones that are purely numerical observations.

We shall revise the text to bring more clarity and to avoid any confusion.

ol. 186-187: '...we did not do a detailed glacier-by-glacier analysis of the reason behind the failure of the algorithm'... Well, you should do this! May be something intrinsically wrong with your setup (e.g. in terms of mass conservation, boundary conditions; see 2c). If this is the case, this is likely to have direct consequences for your results and for some of your conclusions…

[15] We apologise for omitting this important step. However, we had checked that the omission of 120 glaciers out of a total of 814 glaciers did not affect our conclusions or compromised the representativeness of the set (Fig R4).

As described in reply [9], we have now successfully simulated 810 glaciers among the total of 814.

We have provided details of boundary condition and mass conservation in reply [10]. In two glaciers violation of  mass conservation was observed due to noisy bedrock (fig R6).

ol. 247-248: 'We do not have a clear explanation of this effect as yet': ...

[16] What we mean to say here is that, while for scaling-based evolution the relation,

$$\frac{\Delta V_\infty}{V} = \gamma \frac{\Delta A_\infty}{A} \,,$$

Is supported by both theoretical arguments and numerical evidence. However, the corresponding linear relationship for SIA glaciers is a purely empirical one.

We shall clarify this in the revised version.

ol. 256: 'Again, we do not have a theoretical argument for such a power-law behavior and did not explore this further here': …

[17] We meant to say here that we stayed away from a power-law fitting form which would have led to a better fit, and used linear fits as they are supported by theoretical arguments.

We shall modify the text to state this more clearly.

ol. 304-305: '..., it remains to be investigated if the results described here depend on the regional characteristics of glaciers to some extent':...

[18] Mass-balance profile and glacier bedrock profile varies from one region to another. The statement is meant to acknowledge that our best-fit parameterisations need to be checked more thoroughly before applying it for a set of real glaciers on a global scale.

We shall reword the sentence.

[3] The **main conclusion** drawn your manuscript, and which also appears in the title, is that using V-A scaling methods (with 'time-invariant scaling') are likely to underestimate the future sea level contribution from glaciers.

a. I am not sure that the material you presented is convincing enough to support this statement and that the experimental setup is adequate (see previous point).

[19] We have already answered the criticism in our replies above and are not repeating the arguments here.

b. Another major concern that I have is: if this would be the case: why do we not see this when comparing outcomes of V-A scaling estimates compared to more sophisticated methods relying on retreat parameterizations (Huss & Hock, 2015)or flowline models (Maussion et al., 2019)? The first phase of the GlacierMIP project (Hock et al., 2019), in which future large-scale glacier simulations from the literature were compared, did not reveal a tendency for V-A scaling methods to underestimate the

contribution to sea-level rise (SLR). Also in the second phase of the GlacierMIP experiments, in which several ice dynamic (vs. V-A) were included and in which coordinated experiments were performed, no clear tendency can be seen when considering V-A scaling vs. methods in which the glacier geometry (and in some cases also ice dynamics) are explicitly considered. From the material at hand, I would rather tend to believe the outcomes from GlacierMIP than the main conclusions put forward here when it comes to the implications of using V-A scaling for future sea level projections.

[20] We are thankful to the reviewer for raising this critical issue that we should have discussed in the manuscript. Despite a more realistic and therefore a more complex experimental setup employed by Hock et al. (2019), we believe their results contain signs of the biases in scaling models as pointed out in this manuscript.

Consider Table 3 of that paper which is reproduced below. The GloGem model (the only model employed by the authors that does not use scaling) predicted the largest change in both area and volume for 14 out of 16 of their experiments (shown with red arrows in their table 3 reproduced below). This may be an indication of a systematic underestimation of glacier change by the scaling-based models. The majority of the model in the ensemble being scaling based ones, the multimodel means may, thus, have a low bias. Based on our long term simulations, it is likely that if the experiments of Hock et al (2019) was to be extended over longer periods, the above differences between scaling-based and GloGem models may grow larger. Checking the outputs of the different models for a single climate forcing may also be useful here.

**Table 3.** Modeled global glacier mass and area losses by 2100 relative to 2015 (%) for four RCP emission scenarios. For each glacier model, data refer to multi-GCM means (± 1 Std dev.). *Model mean* refers to the arithmetic mean ± 1 Std dev. of all model runs for the same RCP regardless glacier model or GCM. Not all glacier models were run for all four RCPs. Results are also shown excluding the Antarctic periphery (A), and excluding the Antarctic and Greenland periphery (A + G) since some glacier models do not cover these regions

| Glacier model | Volume loss (%) | | | Area loss (%) | | |
|---|---|---|---|---|---|---|
| | Global | Global excl. A | Global excl. A + G | Global | Global excl. A | Global excl. A + G |
| **RCP2.6** | | | | | | |
| SLA2012 | 17 ± 3 | 18 ± 4 | 19 ± 4 | – | – | – |
| MAR2012 | – | 29 ± 7 | 31 ± 7 | – | 31 ± 7 | 33 ± 7 |
| GIE2013 | 14 ± 3 | 14 ± 3 | 17 ± 4 | 18 ± 4 | 19 ± 5 | 22 ± 5 |
| GloGEM | 24 ± 7 ← | 28 ± 9 | 29 ± 9 | 29 ± 7 ← | 32 ± 9 ← | 33 ± 9 ← |
| *Model mean* | *18 ± 7* | *23 ± 9* | *24 ± 9* | *22 ± 8* | *27 ± 9* | *29 ± 9* |
| **RCP4.5** | | | | | | |
| SLA2012 | 21 ± 5 | 22 ± 5 | 23 ± 6 | – | – | – |
| MAR2012 | – | 34 ± 9 | 36 ± 9 | – | 36 ± 8 | 37 ± 8 |
| RAD2014 | 28 ± 8 | 33 ± 10 | 33 ± 10 | 31 ± 10 | 34 ± 12 | 37 ± 12 |
| GloGEM | 33 ± 8 ← | 38 ± 11 ← | 39 ± 11 ← | 39 ± 9 ← | 43 ± 10 ← | 45 ± 10 ← |
| *Model mean* | *27 ± 8* | *31 ± 11* | *32 ± 11* | *35 ± 10* | *38 ± 11* | *40 ± 10* |
| **RCP6.0** | | | | | | |
| SLA2012 | 24 ± 6 | 26 ± 8 | 27 ± 8 | – | – | – |
| MAR2012 | – | 35 ± 8 | 37 ± 9 | – | 36 ± 9 | 37 ± 8 |
| *Model mean* | *24 ± 6* | *32 ± 9* | *33 ± 10* | – | *36 ± 9* | *37 ± 8* |
| **RCP8.5** | | | | | | |
| SLA2012 | 33 ± 6 | 35 ± 7 | 36 ± 8 | – | – | – |
| MAR2012 | – | 46 ± 10 | 48 ± 10 | – | 47 ± 10 | 48 ± 10 |
| GIE2013 | 27 ± 5 | 27 ± 5 | 31 ± 6 | 30 ± 9 | 33 ± 10 | 38 ± 11 |
| HYOGA2 | – | – | 17 ± 4 | – | – | 32 ± 6 |
| RAD2014 | 40 ± 8 | 46 ± 11 | 46 ± 10 | 47 ± 10 | 53 ± 13 | 55 ± 12 |
| GloGEM | 48 ± 9 ← | 55 ± 12 ← | 55 ± 12 ← | 54 ± 9 ← | 59 ± 10 ← | 60 ± 10 ← |
| *Model mean* | *36 ± 11* | *41 ± 13* | *40 ± 14* | *43 ± 14* | *47 ± 14* | *48 ± 14* |

Table 3 of Hock et al. (2019) with red arrows added to highlight the experiments where Glogem predicted the highest loss.

We do acknowledge that there are important differences between our set-up and that of Hock et al (2019) which prevents a direct comparison. We apply all the three models on the same set of steady glaciers, use the same mass-balance profiles, and consider the same idealise ELA perturbation. In contrast, the different models in Hock et al. (2019) do not use the same prescription for computing mass balance forcing, the input climate data used for mass-balance computations are also not identical. In addition, their comparison is over a relatively short period of 85 years, whereas we look at a longer period of 500 years. While our model experiments involve an idealised step change, the authors considered slower and more realistic forcing. As a result of all these differences, our setup is more sensitive to the differences in performance of the glacier-dynamics models alone. Of course, the experimental design of Hock et al. (2019) is tailor-made for the problem of predicting sea-level rise which is the main thrust of that paper.

We shall mention in the updated manuscript that the results of Hock et al. (2019) possibly support our claim of a systematic bias in scaling-based models.

c. You draw your main conclusion (that the loss from V-A scaling with time-invariant scaling is underestimated vs. SIA) from two steady states: an initial one and a final one. You present your results like transient results (e.g. in plots, when describing response times,in section 4.1. describing that cis time-dependent and decreases with time,in section 4.4.,....etc.), but in the end, **it boils down to the fact that the volume of the final steady state with time-invariant V-A scaling is 'too large'**(compared to the SIA). Due to this, the transient volume loss when evolving to this steady state is underestimated (always with respect to SIA results). The main question that you thus need to address is: **why is the V-A scaled final steady state too big?** I am not an expert in V-A scaling, but I would find it surprising that this issue has not been addressed in other V-A scaling studies and that no solutions to this problem have been formulated. In the end, from my understanding, what happens is that many glaciers that reduce in size lose their lowest part, which are often the most gently sloping parts of the glacier and where the highest ice thickness is thus found (in most ice thickness reconstructions this clearly appears, where in the end, a large part of the reconstruction results from the negative correlation between the surface slope and the local ice thickness; see Farinotti et al., 2017). It is thus to be expected that the V-A scaling that you use to create the initial steady state does not hold for the final one. This is something that would need to be explored in more detail, and for which studies in which the volume scaling also uses information from other glacier characteristics (e.g. the glacier slope) could be useful (Grinsted, 2013; Zekollari & Huybrechts, 2015; see e.g. Fig. 9a in the latter, which summarizes the main point made here).

[21] We agree with the reviewer's assertion that an important limitation of a scaling-based model is that the scaling-based model underestimates the climate sensitivity of glaciers. However, we do not agree with his/her point of view, that is the only issue. As we have already demonstrated in the manuscript, there are two more critical issues: A. the scaling-based model underestimates glacier response time (please refer to replies [1] and [3]), B. the volume and length response times are predicted to be identical under scaling.

While there are several existing investigations of the relative performance of the scaling-based models (including those cited in our discussion paper and in the reviewer's comment), the above three limitations may not have been brought out clearly.

Also, a common issue with the existing comparisons of scaling-based model with ice-flow models is that often both $c$ and $\gamma$ were taken to be time-dependent fitting

parameters empirically. This was advised against by Bahr et al. (2015) on solid theoretical grounds.

We do agree with the reviewer's comment above that a fixed time-independent scaling form does not work. Our SIA simulation clearly shows that the scale-factor **c** is time-dependent (please see reply [38]). While the gently-sloping lower ablation zone, as suggested by the reviewer, may be one of the reasons behind this, it is not the only reason. For a set of 1-d glaciers on linear bedrock with the same bedrock slope, a similar bias in the scaling-based model is obtained (fig R1, reply [1]). So any modified scaling formulation involving glacier slope etc. may not be able to cure the bias entirely.

If a parameterisation of the time dependence of *c* in terms of slope and other factors is found, then that may give a clear answer to the questions why scaling predicts relatively smaller sensitivity (as pointed out by the reviewer) and response time.   However, even that is not going to correct the drawback of having the area and volume response times (reply [1], [3]). Therefore, in this paper, we choose to focus on obtaining an alternative linear-response model that reduces the above biases, rather than trying to investigate how scaling models can be improved.

[4] Unclarities in the manuscript. I found the text difficult to follow and quite often had to re-read sentences several times before being able to grasp their meaning. A few examples include:

[22] We shall work on improving the readability, and appropriately rewrite the following sentences.

a. l. 8-9: '..and validate them with results from scaling-based simulation of the ensemble of glacier'

b. l.84-85: '...are then empirically extended in order to obtain accurate parameterisations the linear-response properties of the SIA-simulated glaciers'

c.l.86-87: 'The linear-response model the long-term total shrinkage of glaciers as predicted by the scaling-based method (Radićet al., 2007), and the linear-response model are compared with the corresponding response'd.....etc.
See also comments on specific sections below.This makes it tedious to go through the manuscript. Furthermore, there a substantial number of grammatical errors, some of which (but not all) have already been pointed out by the first reviewer.

[23] We shall try to minimise such errors in subsequent revision.

Also, many figures cannot be interpreted/read independently, without having to refer to the caption. It would be good if all essential information (e.g. meaning of colors used, R^2 values, equations,...etc.) could be directly included in the figure. Some other comments for specific sections(non-exhaustive list and not focusing on grammatical errors)

[24] We shall update the figures mentioning the best-fit forms and $R^2$ within each of the figures.

o1. Introduction:
'methods solving the dynamical ice-flow equations' à'numerical cost of such a computation on a global scale is prohibitive': well is not really the case anymore. In general: would be good to acknowledge regional-to global studies in which ice flow is explicitly accounted for (Clarke et al., 2015; Maussion et al., 2019; Zekollari et al., 2019).

[25] We shall reword the statement, and include relevant references including the ones mentioned here.

o1.2. Motivation for the present study: difficult to follow the first paragraph: be more specific when you refer to c and gamma not continuously mix with other terminology 'time invariant scaling-based parameterisation', '...given the known violation of the time-invariant scaling assumption'.

[26] The scaling equation,

$$V = cA^{\gamma}$$

does not require *c* to be a time-independent constant (Bahr et al., 2015). However, that assumption is necessary to use this equation to predict glacier evolution. So they are not interchangeable.

o2. Quite abstract and thus very difficult to go through for someone who is not an expert in V-A scaling. Could make it less technical by for instance adding some additional information that links the various parts.

[27] This section sets the notation, and provides the mathematical derivation of various results. So it may appear a bit technical. However, we appreciate the need for making it

more accessible, and shall add a paragraph in the end summarising the results in plain language.

o3.1.: 2-dimensional SIA model:

•l.152-154: where did you get the ice thickness from? From Kraaijenbrink et al. (2017)directly? As the ice thickness is quite crucial in your story (it determines the volume...), why did you not consider the consensus estimate of Farinotti et al. (2019), which is freely available?

[28] We had used data from Kraaijenbrink et al (2017) as it has debris-cover and debris thickness information as well - which we initially planned to incorporate. Also, when we started the project Faronotti et al. (2019) was not published.  Since we do not tune our models to obtain accurate descriptions of each of the glaciers, we stick to the bedrock originally used. Any reasonable choice of bedrock serves our purpose of simulating an ensemble of glaciers with realistic geometry. We agree that accurate  bedrock and tuning the parameters of the SIA model to fit the thickness and velocity maps of each of the glaciers are necessary for any computation of future glacier change or sea-level rise.

•SIA: refer to the original work by Hutter (1983)also.

[29] We shall refer to this article.

•You neglect basal sliding (l. 161). Justification? Could refer to other studies where this is done, like e.g. Gudmundsson(1999) and Clarke et al. (2015).

[30] We had neglected sliding as the theoretical results of Bahr et al. (2015) that we built upon are derived in that limit. For a set of real Himalayan glaciers sliding would surely be important as mentioned in section 4.2 of the manuscript.

•l.168-178: this is related to the SMB, which you apply in all cases(i.e. also for the linear-response model and the V-A scaling, right?). Not sure this section is correctly placed here in the '2-dimensional SIA model' section.

[31] we shall add reference to eq. 13 in the later sub-sections describing the 0-d models.

•l.183: through several exclusion you keep 68% of the initial glaciers... How much does this represent in terms of glacier volume and glacier area when compared to the total glacier sample?

[32] With the present runs we keep 703 out of the total 814. This corresponds to a number fraction of 86% and an area fraction of 89% . Also see fig R4 above for the area and slope distribution of all the glaciers and the selected ones.

•l.188-196: you explain some simplifications related to debris cover, avalanche and sliding have been made and that this may influence your results. Well, you have made much larger simplifications than this: e.g. linear SMB profiles with strongly imposed max. SMB, steady state assumptions for glaciers,... à not even worth mentioning these more detailed simplifications in my opinion. With all these simplifications, would have been better to opt for idealized setup likely (see main comment 2a).

[33] We prefer to list out all the limitations of our model as depending on the glacier being considered, one or several of  these issues have proven to be quite critical in the Himalaya.

We have already discussed (replies [1] and [3]) why idealised glaciers do not serve the purpose for us.

•l. 194: 'These simplifications do not weaken our study': not sure you can judge on this yourself...

[34] the sentence will be modified.

o Section 3.2.:

•l.204: 'was fixed at... because...': don't understand the causality (i.e. link between cause and consequence).

[35] The sentence will be modified. We mean to say: the linear mass-balance profile implies m=1, and then, using the formula given by Bahr et al (2015), gamma = 1.286.

•Figure 1: SIA-derived volumes are scaled by a factor 10: why? Does not really make sense and unclear when just looking at the figure without reading the caption... Axes should be correct in the figure and not only for a part of the data you show..Also

illustrates the unclarity in the figures mentioned in main comment 4 (problem that figures cannot be interpreted without referring to their caption).

[36] We have modified the figure by removing the scaling by 10 (Fig R5). We shall update the other figures following the suggestion by the reviewer.

oSection 3.3.:

•l.208-210: complicated way to say that you consider e-folding time scales. Would reformulate this and add references for this to e.g. Leysinger Vieli & Gudmundsson(2004).

[37] As our aim is to find the best fit linear-response properties. Thus instead of computing e-folding time, we directly fit the following linear-response form to obtain both climate sensitivity and response time for each glacier.

$$\Delta V(t) = \Delta V_\infty (1 - e^{-t/\tau_v})$$

Only for a purely exponential decay the response time would be identical to the e-folding time.

oSection 4.1.:

•l.225: $V = cA^{1.286}$: not sure I understand. Does this statement apply for the initial and/or final steady state volumes? And can all the volumes be described with this single relationship? Is the fact that quite different rate factors are used not a problem for this (see main comment, point 2c)?

[38] As shown in main figure 1 (A) both the initial and final states obey the same scaling form with different values of *c.* This is consistent with the rigorous derivation of the scaling law (Bahr et al., 2015). As long as a large enough ensemble of glaciers with a wide range of glacier area is considered, a fixed area-volume scaling relation is a good statistical description, though it may have considerable bias if any single glacier in the set is considered (Bahr et al., 2015).
        The variability of rate factor, geometry (as long as width exponent q is the same), mass balance profile (as long as exponent m is the same) etc only add noise the scaling, and lead to scatter in the scaling plot. Such effects do not ruin the volume-area scaling. This is consistent with main figure 1A and fig. R1. In fig. R1, there is little scatter as geometry, rate factor, mass-balance gradient are exactly the same for each of the glaciers. However, in main fig 1A a considerable scatter is present due to variability of

these factors. Of course, in a set of real glaciers significant variability of the above parameters, and consequently, a significant scatter in the corresponding scaling plots are present (Bahr et al., 2015).

•l.227 + l.230 + l.232:here you mention that c is time dependent. Not sure you can say that it is time dependent: simply results from the fact that final steady state volume for V-A scaling is 'overestimated' (vs. SIA). As a result the evolution to this steady state is different. See main comment 3c for this.

[39] 1. We have shown in main figure 1A that initial and final best-fit c are different.
2. We have shown in main figure 1B that fo SIA simulated glaciers time-evolution trajectories are not linear in V-A plane with log-log scale.

The above two observations prove that c depends on time (as long as the same form $V = cA^{1.286}$ is used to describe the ensemble as argued by Bahr et al. (2015) on theoretical ground).

•l.235-237: relates to main comment 3c again. If you do not modify the V-A scaling, then problems will arise when considering the same glacier that is much smaller in a warmer climate (when rising the ELA in your case): you typically lose the lower parts where most volume is and volume will thus be 'overestimated'. Is this not accounted for in some way in future glacier evolutions based on V-A scaling? As a part of this discussion, studies in which V-A scaling is extended with other glacier characteristics (such as the surface slope; Grinsted, 2013; Zekollari & Huybrechts, 2015)would be good to include. Such relationships which could prove to remain valid over time, even without changing scaling and exponents.

[40] We agree with the reviewer that with the incorporation of appropriate additional variables like slope etc., it is possible to improve the scaling formulation. However, as shown in reply [1] above, this can not cure the bias entirely. Also, it cannot resolve the issue of equality of area and volume response time (reply [1] and [3]) that is inherent in scaling. Therefore, we took an alternative route of developing an accurate linear-response description to minimise the bias. We do not rule out that there may be other possible routes to reduce the bias.

We shall make the above point clear in our revised discussions.

oSection 4.2.:

•l. 243: 'This is exactly what is seen in Fig. 2b, which shows...': I cannot directly see this…

[41] We wanted to point out the fact that fractional volume change is 1.87 times the fractional change in area, while scaling theory predicts a smaller prefactor of factor of 1.286.

We shall rewrite the sentence to make it clear.

•l.244: 'change in c to the tune of ~13%': what does this mean?

[42] We meant to say that: SIA simulation showed a fractional change in volume that is ~50% larger than that predicted by the scaling method for a given fractional change in area. A relatively small 13% change in c is not the only factor behind this, the associated mass-balance feedback also plays a  significant role.

•l.255: 'The above figure': will depend where your figure comes in final manuscript…

[43] We shall refer to the figure number here.

oSection 4.3:•
I was wondering what the point is that you want to make with this section? It is known from literature that volume responds faster than area (e.g. Oerlemans, 2001; Leysinger Vieli & Gudmundsson, 2004).

[44] We apologise for not referring to prior work by Schmeits and Oerlemans (1997), Oerlemans (2001) and, Vieli and Gudmundsson (2004) that had discussed that volume response is faster than the length response. We shall correct that error.

The main result in this section is that a scaling-based evolution leads to the same response time for area and volume (as shown with both theoretical arguments in section 2.3 and numerical results in main fig 4), while in SIA simulations the area response time is larger. This is again a major limitation of scaling-based methods.

We shall rewrite the paragraph to make the statement clear.

• l.260-264: relationship between volume and area response times. How does this compare to the relationship others have found in the literature?

[45] We thank the reviewer for the comment.

Oerlemans ( 2001) suggested a proportionality constant of 0.74 between volume and area response time. Vieli and Gudmundsson (2004) obtained corresponding values of 0.60, 0.70 and 0.67.

The above values are comparable to our best-fit value of 0.69.

We shall discuss this point in the revised manuscript.

oSection 4.4.:

•l.271-272: 'with most of the changes taking place during the first couple of centuries': this is not a result/finding.. This directly results from the e-folding time-scale when forcing a steady state glacier with an instantaneous forcing in SMB.

[46] The sentence "Starting with an initial volume (area) of 603 km$^3$ (5144 km$^2$) the 551 glaciers simulated by SIA loses 123 km$^3$ (521 km$^2$) of volume (area) in 500 years after the step-change in ELA by 50 m, with most of the changes taking place during the first couple of centuries (fig. 6)." is a fair description of the data presented in fig. 6.

•l.273: 'underestimates the long-term change': not about reaction/response. This is direct consequence of fact that final steady state volume is too large (see main comment 3c)

[47] Irrespective of the interpretation, the statements in this paragraph is a correct description of data presented in fig 6.

•l.279-280: '...suggests that there might be significant negative biases of mountain glacier contribution to sea-level rise as computed by scaling-based methods'(+ section 4.5, l.300-302): well, do not see this in GlacierMIP phase 1+2... Is a very strong statement to make and should be sure that it is well-founded.

[48] Please refer to comment [20] where give evidence that GlacierMIP contains enough hints about such a possible bias. We shall include that discussion in the revised manuscript.

O Section 4.5:6

•l.296-297: 'More detailed studies that relaxes some of the above mentioned assumptions are needed...': not sure what you mean by this. Would also make sense that you dig into this: e.g. by focusing on real transient response vs. comparing two steady states (what you do now and then translate into an analysis of the transient response resulting from this: see main comment 3c).

[49] We are considering transient response  to step change in climate and not steady states. We are following a well established method for characterising glacier response by computing response properties of steady states (e.g. Oerlamans, 2001). If steady-state linear-response properties (eg climate sensitivity and response time) are known, the response to any arbitrary mass balance forcing can be computed as (we quote from our reply to comments by reviewer 1),

$$\Delta V(t) = \Delta V(0)e^{-t/\tau} + \frac{\Delta V_\infty}{\delta E} \int_0^t \Delta E(t')e^{-(t-t')/\tau} dt'.$$

Here, $\frac{\Delta V_\infty}{\delta E}$ is the climate sensitivity of glacier volume. $\Delta V(0)$ is the initial departure from a steady state that can be obtained from the observed rate of volume change as $\Delta V(0) = -\tau \frac{dV(0)}{dt}$. A similar expression can be written down for area evolution as well. We propose to include these details in the discussion section of the revised manuscript.

We note that a similar linear-response framework had been applied to reconstruct climate forcing from glacier length change records (Oerlmans, 2005). While we focus on idealised climate forcing to obtain linear-response properties, the response properties derived from that analysis allow predicting the response to any general forcing as long as the fractional glacier changes are small.

•l.299: 'intruding more scatter in the fits': what does this mean?

[50] This is a typographical error. "Intruding" will be replaced by "introducing".

oSummary and Conclusions:

•l.309-310: scale factor reduces over time. Well, not sure the time dimension is adequate here. Boils down to having a final steady state that would require a smaller value for c: see main comment 3c.

[51] There is probably no inconsistency between our statement and that of the reviewer. Unless c is time-dependent, c cannot be different for the initial and final states. Also, the discrepancies between SIA and scaling model during the first hundred years ( see main fig 1, fig R1) when the glaciers are in a transient state, is consistent with a time-dependent c.

•I.324: computational efficiency. OK, still important, but is not really a limitation anymore, due to which V-A scaling becomes less important (and also driven by the release of new datasets with regional-to global spatial coverage at individual glacier level: see main comment 1).

[52] Here we are only referring to the "computational advantage" and not claiming that linear-response is the only way of computing long-term glacier change.

•Code availability: for which models is the code available? Seems to suggest that the SIA code is not available. Not sure if this fully agrees with the policies of The Cryosphere: see www.the-cryosphere.net/about/data_policy.html

[65] We shall make all codes available as per the policies of the journal.

**References:**
Bahr, D. B., Pfeffer, W. T., and Kaser, G. (2015). A review of volume-area scaling of glaciers. Reviews of Geophysics, 53(1), 95-140

Clarke, G. K. C., Jarosch, A. H., Anslow, F.S., Radić, V., & Menounos, B. (2015). Projected deglaciation of western Canada in the twenty-first century. Nature Geoscience, 8, 372−377. https://doi.org/10.1038/ngeo2407

Farinotti, D., Huss, M., Fürst, J. J., Landmann, J., Machguth, H., Maussion, F., & Pandit, A. (2019). A consensus estimate for the ice thickness distribution of all glaciers on Earth. Nature Geoscience. https://doi.org/10.1038/s41561-019-0300-3

Hock, R., Bliss, A.,Marzeion, B., Giesen, R. H., Hirabayashi, Y., Huss, M., et al. (2019). GlacierMIP −A model intercomparison of global-scale glacier mass-balance models and projections. Journal of Glaciology. https://doi.org/10.1017/jog.2019.22

Huss, M., & Hock, R. (2015). A new model for global glacier change and sea-level rise. Frontiers in Earth Science, 3, 1−22. https://doi.org/10.3389/feart.2015.00054

Kraaijenbrink, P. D. A., Bierkens, M. F. P., Lutz, A. F., & Immerzeel, W. W. (2017). Impact of a global temperature rise of 1.5 degrees Celsius on Asia's glaciers. Nature, 549, 257−260. https://doi.org/10.1038/nature23878

Kulp, S. A., & Strauss, B. H. (2019). New elevation data triple estimates of global vulnerability to sea-level rise and coastal flooding. *Nature communications*, *10*(1), 1-12.

Leysinger Vieli, G. J.-M. C., & Gudmundsson, G. H. (2004). On estimating length fluctuations of glaciers caused by changes in climatic forcing. Journal of Geophysical Research, 109(F01007), F01007. https://doi.org/10.1029/2003JF000027

Marzeion, B., Jarosch, A. H., & Hofer, M. (2012). Past and future sea-level change from the surface mass balance of glaciers. The Cryosphere, 6(6), 1295−1322. https://doi.org/10.5194/tc-6-1295-2012

Oerlemans, J. (2001). Glaciers and climate change. Dordrecht: A. A. Balkema.

Oerlemans, J. (2005). Extracting a climate signal from 169 glacier records. *Science*, *308*(5722), 675-677.

Rounce, D., Hock, R., & Shean, D. (2020a). Glacier mass change in High Mountain Asia through 2100 using the open-source Python Glacier Evolution Model (PyGEM). Frontiers in Earth Science, 7, 331. https://doi.org/10.3389/feart.2019.00331

Rounce, D. R., Khurana, T., Short, M., Hock, R., Shean, D., & Brinkerhoff, D. J. (2020b). Quantifying parameter uncertainty in a large-scale glacier evolution model with a Bayesian model: Application to High Mountain Asia. Journal of Glaciology.

Schmeits, M. J., & Oerlemans, J. (1997). Simulation of the historical variations in length of Unterer Grindelwaldgletscher, Switzerland. *Journal of Glaciology*, *43*(143), 152-164.

Zekollari, H., Huss, M., & Farinotti, D. (2019). Modelling the future evolution of glaciers in the European Alps under the EURO-CORDEX RCM ensemble. The Cryosphere, 13, 1125−1146. https://doi.org/10.5194/tc-13-1125-2019

---

## Author Response (AR1)

Dear Editor,

We gratefully acknowledge critical inputs from both the reviewers. Based on their suggestions, we have made the following important changes in this revised version.

1. Spelt out our objectives at the beginning clearly in order to clarify the experimental design.

2. Repeated experiments to simulate 810 out of 814 Himalayan glaciers. We have now used 703 glaciers for our analysis.

3. Added finer details about the experimental methods and results in the main text. Also, added 10 supplementary figures to clarify various things.

4. Expanded the discussions about the theoretical/numerical results, including how gradual steepening of the retreating glaciers can contribute to a decline in c.

5. Acknowledged and emphasised the idealisation involved the set of simulated "synthetic Himalayan glaciers", and possible dependence of the results on the chosen ensemble/model.

7. Explained how the long-term biases in scaling models based on the feedback of shrinking ablation zone on net mass-balance. The scaling models assume an equality of the area and volume response times.In contrasting, the SIA model predicts a larger area response time.

7. Discussed additional results from 1-d flowline model simulation of highly idealised glaciers to support the general applicability of our results.

8. Discussed how results from the GlacierMIP support our claim of possible biases in scaling models.

9. Provided methodological details about the application of the linear-response model to real transient glaciers that are forced by an arbitrary time-dependent ELA.

10. Added a careful itemised summary of the results obtained in the end.

With these changes we hope to have addressed the issues raised by the reviewers. We look forward to further comments on the experiments, results, and conclusions as presented in the revised manuscript.

Our point-by-point replies to the reviewer's comments (in red) are appended below.

Regards, Argha

**Replies to comments by reviewer #1 (Eviatar Bach):**

**Major issues:**

1. For the linear-response model based projections, the authors write that they fit the four parameters (area and volume sensitivities and response times) for each glacier based on the SIA data. They then validate the projections obtained using these parameters on the same SIA data. This is using the same data for fitting and validation, so it is not surprising that it replicates the data fairly well. Testing this method requires validation data that is not part of the fitting. A possible way to do this would be to only use a portion of the time-series of each glacier to fit, and validate on the rest (for example, fit on the first 50 years, project into the future, and validate on the 450 remaining years).

There is this this sentence which I was not clear on: "We have verified the linear-response model obtained by fitting the SIA simulation results for the ensemble of 551central Himalayan glaciers, similarly outperforms the scaling-based method for another set of 143 glaciers from the western Himalaya (figure A2)." Were the parameters obtained for the central Himalayan glaciers somehow extrapolated to the western Himalayan ones? Or were the parameters fit for every western Himalayan glacier as well? It is not clear from the description. If the authors use an extrapolation method, it would be important to describe it.

We have calibrated the model using data from 703 Central Himalayan glaciers, and validated it (without any further calibration) on an different set of 204 glaciers in the western Himalaya. We have clarified this in L83-85 and L193-197 of the revised manuscript to avoid any confusion.

2. The linear-response method is being proposed as an alternative to scalingbased methods for projecting glacier volume evolution. However, I am not clear on how this would be implemented in practice. The climate sensitivities  $\Delta V_{\infty}$ and  $\Delta A_{\infty}$  characterize the response of an initially steady-state glacier to a perturbation in the ELA. How can this be used to project evolution of a glacier that is already transient, and in a situation where it is not a single perturbation in the ELA, but that the ELA is continually rising?

We have added a section (3.4) titled "Applying the linear-response model to real glaciers", where we explain how the model can be applied to transient glaciers that are forced by any arbitrary time-varying ELA perturbation.

3. Furthermore, it seems that the linear-response method would require a relatively long time-series of the area and volume evolution of each glacier in order to fit the parameters, which is often not available. I would like to see a discussion of the data requirements and feasibility for use in sea-level projections. Following the standard paradigm of linear-response theory, the response parameters can obtained from an SIA simulation of the step-change response of steady glaciers without any time-series data – exactly the way we have done here for the 703 glaciers here. Of course, the SIA model can be tuned using surface velocity and ice-thickness data wherever available (which we have not done here) to obtain more accurate parameterisation of the linear-response properties for any given set of glaciers. We have discussed this issue in L407-412.

**Other issues:**

1. The authors remove some glaciers from consideration in several parts of the paper, such as those that had fractional changes of more than 50% over 500 years, and those with response times higher than 300 years. Also, in another part of the paper, glaciers with large values of  $\Delta A_{\infty}/A$  are removed, and another cut-off on  $\Delta V_{\infty}/V$  is imposed. I don't see an adequate justification of why these were removed, and doing so biases the results.

We have now explained (L143-146) that is it is necessary to remove the glaciers with 'large' changes as linear response theory do not apply to them. Additionally, supplementary fig. S6 is included, where a cut off 0f 20% is used, to demonstrate that our results do not depend on the cutoff chosen.

The transient runs are extended to 1000 years, extending the cut-off on response time to 500 years so that only 9 glaciers are removed. Supplementary fig. S7 presents justification why this cutoff is needed.

In the revised version, we now consider 703 glaciers. We have compared the area and slope distributions of these 703 and 810 glaciers (supplementary fig. S8).

2. "The minor differences are due to the time-invariant scaling assumption made here." Please clarify in more detail what is the difference between your derivations and those of Harrison (2001).

The difference between the expressions is mentioned now (L 256).

3. In Fig. 1B, scaling the SIA results by 10 for visual comparison is confusing. It's also hard to distinguish which are the thick and thin lines.

We have updated fig 1B based on the above criticism.

We have also corrected the typographical/grammatical errors pointed out by the reviewer.

**Replies to the comments of reviewer #2 (Anonymous):**

**[1] State-of-the art.**

The authors compare the outcome of V-A scaling with results from a SIA model. V-A has indeed been used in some important regional-to global studies in the past (e.g. Marzeion et al., 2012; Radić et al., 2014) due its computational efficiency. Moreover, with spatial estimates of ice thickness lacking for individual glaciers at the time, V-A

methods offered a good alternative to estimate the volume of a glacier (and its changes through time). However, increasing computational performance and new glacier-specific inventories on e.g. ice thickness (Huss & Farinotti, 2012; Farinotti et al., 2019)and mass balance (e.g. Brun et al., 2017; Braun et al., 2019; Dussaillant et al., 2019; Zemp et al., 2019), now allow for far more sophisticated methods to simulate the dynamic evolution of glaciers. This includes methods based on imposing observed geometry changes in which the glacier geometry is explicitly accounted for(e.g. Huss & Hock, 2015; Rounce et al., 2020a, 2020b) and more recently also flowline models in which glacier dynamics (i.e. mass transfer within a glacier) are included when projecting glacier changes at regional to global scales (Maussion et al., 2019; Zekollari et al., 2019). When reading this manuscript, it seems like V-A scaling is a state-of-the art approach, and that you compare it to something more sophisticated (2-D SIA model). This comparison would have been very relevant a few years ago, when V-A scaling was state-of-the art (I am for instance thinking about the excellent work realized by Surendra Adhikari during his PhD; see e.g. Adhikari & Marshall, 2012), but has, in my opinion, lost some of its interest by now. With the new glacier-specific ice thickness estimates and other information derived from remote sensing becoming widely available (outlines, surface topography, ice thickness derived from this), the importance of V-A scaling methods is now strongly reducing and is likely to continue doing so so in the near future (see e.g. discussion by Haeberli, 2016). I do therefore have some reservations whether the 'The Cryosphere' is the ideal medium to share these (somewhat outdated?) findings. This concern is furthermore strengthened by my doubts about the experimental setup and the validity of your main conclusions as elaborated in the following points.

We do not agree with the reviewer that scaling models have lost significant, and need not be studied. We are not aware of any global estimate of sea-level contribution of glaciers based on flow models studies. Most of the available recent estimates (e.g., a recent intercomparison study: Hock et al., 2019; a global-scale vulnerability study: Kulp and Strauss, 2019), strongly rely on scaling models. However, we do acknowledge the potential of the 1-d and 2-d ice-flow models and revised the introduction accordingly (L26).

**[2] The experimental setup:**

a. Comparing different methods and models is always quite complicated. This is especially the case when considering 'real' cases (glaciers with real geometries in your case). A study such as the one presented here would have greatly benefited from an idealized setup, which would have made comparisons more straightforward and allowed to disentangle differences between simulations obtained fromV-A scaling and those relying on2-D SIA modelling:see e.g. Leysinger Vieli and Gudmundsson (2004)and Adhikari and Marshall (2012). Here a 'selection' of glaciers is considered, due to some 'problems' occurring when considering all glaciers in the region (see point 2b), which makes it even questionable how representative these are for this given region. With idealized glacier geometries, you could have explored the effect of glacier size, surface slope,...on the discrepancies between V-A based results and SIA modelled results more carefully.

As we have explained in the revised text our main objectives are (L64-70),

1. To obtain analytical predictions for climate sensitivity and response time of glaciers in a scaling model.

2. To compare the climate sensitivity and response time of a large number of synthetic glaciers with realistic geometries, as obtained from a scaling model and a 2-d SIA model.

3. To investigate the possibility of long-term biases in scaling model estimates of changes in glacier area and volume with respect to corresponding SIA results.

4. To find convenient parameterisation of glacier response properties obtained from the SIA simulations, and develop an accurate linear-response model.

Idealised models are inadequate for the objectives 4 above (and also for 2 and 3).

However, motivated by the reviewer's suggestion, we included idealised flowline model simulations that supports our conclusions. (sect. 3.3 and supplementary fig. S10).

**b. Several arbitrary steps and decisions are made in the manuscript. A few examples of decisions that are hard to understand / seem not well funded:**

**ol. 181-182: you exclude glaciers with a large change in area over the 500-year time period? Why? This seems arbitrary, but you must have a reason for this. Moreover, how this this influence your results? This makes the sample less representative ...**

We have now explained (L143-146) that it is necessary to remove the glaciers with 'large' changes as a linear-response theory cannot be applied to them. Additional data is presented in the supplementary fig. S6 to demonstrate that our results do not depend on the cutoff chosen.

The set of selected 703 glaciers include 86% of all 814 Ganga basin glaciers (larger than 2 sq km) and cover 89% of their area. These two sets have similar distribution of slope and area as well (supplementary fig. S8) so that the selected ensemble can be considered representative.

ol. 182-183: why do you exclude glaciers with long response times? Again, this makes your sample less representative (you probably exclude a certain type of glaciers, likely those that are gently sloping: see e.g. Haeberli & Hoelzle, 1995). Is this because these glaciers are not in steady state after 500 years? If so, you should simply run your experiments for longer and not exclude these glaciers.

We have extended the runtime to 1000 years and the cut-off to 500 years so that only 9 glaciers out of the total 814 ( $\sim$ 1% number-wise and  $\sim$ 2% area-wise) are left out now. The rationale behind this is explained with the help of supplementary fig S7 (also, L146-148 in the main text).

The selected 703 glaciers have similar slope distribution as that of all the 810 glaciers as shown in Supplementary fig. S8.

o Figure 1: you show '200 randomly chosen glaciers': why? Should show them all!

o I.249-250: 'In fig. 2b, about 30 data points,...were not included in the fit': why? You mention something about possibly creating a bias in the linear fit in the next sentence, but I do not see where this would result from / what the problem could be.

In the revised version, data for all the 703 glaciers are shown in fig 1, and no data points are excluded from the fits.

c. The Setup of your SIA model is not fully clear.

oYou mention that for > 100 cases 'our algorithm for finding a steady-state similar to present extent did not converge or the final steady state glacier geometry was not realistic': how is this possible? How can a simple SIA solution not 'converge' to steady state(in fact, even analytical solutions may exist that do not even require running the

SIA model to find the steady state: see e.g. Jouvet & Bueler, 2012)? And what do you consider 'not being realistic'?

We have now updated our algorithm so that 810 out of 814 glaciers are included in the experiment. We have only excluded only 3 glaciers where bedrock noise/steepness led to violation of mass conservation – a known issue with SIA model (Jarosch et al., 2013), and another one due to a mapping error. This is discussed in the text (L134-136) and in the supplementary (fig. S2).

Which boundary conditions did you use to ensure mass conservation (e.g. to ensure specific ice-free regions do not become ice-covered)? You mention that mass conservation was monitored (l. 162-163): but how do you do this (this is not so straightforward to do...)? Did you check that the integrated SMB over your glacier is zero for the steady states (which it should be)?

Would also be good if you could consider some benchmark experiments (e.g. Jarosch et al., 2013)to make sure your model is mass conserving.

Boundary conditions are now described in the text (L127-128: noslip BC at the bedrock, and noflux BC at the domain boundary).

The algorithm used for checking conservation is given in L130-135, along with corresponding plots in supplementary fig S5. Ice conservation was explicitly verified to be satisfied up to 1 part in 109. The steady-state mass-balance was zero.

o Why do you randomly pick the values for the rate factor in Glen's flow law (not 'Glenn' + add a reference to the original studies, e.g. Glen, 1955)? The value of the rate factor will have a large influence on the local ice thickness and on thus the glacier volume. By picking this randomly: could be 'off' quite a lot from the 'reference/observed' volume of the glacier. Why do you not match this to the reference volume from every glacier that you have from Kraaijenbrink et al. (2017)?

Is this also not problematic when working with single values for c and g later in your analyses for all glaciers (e.g. for the best fits): you make some glaciers too thin and some too thick.

We did not calibrate the rate factor (or the balance gradient) to match the available volume and/or velocity estimate to avoid the associated computational cost. Tuning the rate factor to fit the thickness may not be a good idea, as it may lead to unrealistically small glacier velocities, and thus, unrealistic response properties.

We have acknowledged that the resultant ensemble is not a faithful copy of the Himalayan glaciers (L108-110, and Sect. 3.5). However, this ensemble serves the present purpose, as we are interested in a set of synthetic glaciers with realistic geometries. Plots comparing the area and ice thickness of SIA simulated steady glaciers, and the corresponding estimates from Kraaijenbrink et al. (2017) are added (fig S3, S8).

We have also added references to emphasise that the range of values of rate constant and balance gradient is realistic (L122-125).

The problem of large uncertainties in estimated volume of individual glaciers using a scaling relation with a single c is wellknown (e.g. Bahr et al., 2015). Since we are considering the scaling models that use this formulation, we stick to the above statistical interpretation of the scaling relation. This is clearly stated at the outset (L46-56).

d. Lack of in-depth analyses. Often you seem to be perplexed by some findings yourself and leave important questions unanswered, which is unsatisfying for the reader. This questions the thoroughness of your approach, e.g.:

We have tried to improve our discussions.

ol. 186-187: '...we did not do a detailed glacier-by-glacier analysis of the reason behind the failure of the algorithm'... Well, you should do this! May be something intrinsically wrong with your setup (e.g. in terms of mass conservation, boundary conditions; see 2c). If this is the case, this is likely to have direct consequences for your results and for some of your conclusions...

As described in our replies above,

\* We have now include 810 glaciers out of the total of 814 in our analysis.

\* The procedure to check mass-conservation is explained in the text, and corresponding plots are added to the supplementary.

\* Boundary condition is stated in the methods section.

\* Details of the three excluded glaciers are provided.

ol. 247-248: 'We do not have a clear explanation of this effect as yet': ...

ol. 256: 'Again, we do not have a theoretical argument for such a power-law behavior and did not explore this further here': ...

ol. 304-305: '..., it remains to be investigated if the results described here depend on the regional characteristics of glaciers to some extent':...

The above comments have been deleted/modified in the present version.

[3] The **main conclusion** drawn your manuscript, and which also appears in the title, is that using V-A scaling methods (with 'time-invariant scaling') are likely to underestimate the future sea level contribution from glaciers.

a. I am not sure that the material you presented is convincing enough to support this statement and that the experimental setup is adequate (see previous point).

We hope to have answered this criticism in the replies above.

b. Another major concern that I have is: if this would be the case: why do we not see this when comparing outcomes of V-A scaling estimates compared to more sophisticated methods relying on retreat parameterizations (Huss & Hock, 2015)or flowline models (Maussion et al., 2019)? The first phase of the GlacierMIP project (Hock et al., 2019), in which future large-scale glacier simulations from the literature were compared, did not reveal a tendency for V-A scaling methods to underestimate the

contribution to sea-level rise (SLR). Also in the second phase of the GlacierMIP experiments, in which several ice dynamic (vs. V-A) were included and in which coordinated experiments were performed, no clear tendency can be seen when considering V-A scaling vs. methods in which the glacier geometry (and in some cases also ice dynamics) are explicitly considered. From the material at hand, I would rather tend to believe the outcomes from GlacierMIP than the main conclusions put forward here when it comes to the implications of using V-A scaling for future sea level projections.

Please refer to Table 3 of Hock et al. (2019). It is clear that scaling-model estimates for changes in both area and volume are always lower than that obtained in GloGEM. Please see the discussion in L335-344.

c. You draw your main conclusion (that the loss from V-A scaling with time-invariant scaling is underestimated vs. SIA) from two steady states: an initial one and a final one. You present your results like transient results (e.g. in plots, when describing response times, in section 4.1. describing that cis time-dependent and decreases with time, in section 4.4.,....etc.), but in the end, it boils down to the fact that the volume of the final steady state with time-invariant V-A scaling is 'too large' (compared to the SIA). Due to this, the transient volume loss when evolving to this steady state is underestimated (always with respect to SIA results). The main question that you thus need to address is: why is the V-A scaled final steady state too big? I am not an expert in V-A scaling, but I would find it surprising that this issue has not been addressed in other V-A scaling studies and that no solutions to this problem have been formulated. In the end, from my understanding, what happens is that many glaciers that reduce in size lose their lowest part, which are often the most gently sloping parts of the glacier and where the highest ice thickness is thus found (in most ice thickness reconstructions this clearly appears, where in the end, a large part of the reconstruction results from the negative correlation between the surface slope and the local ice thickness; see Farinotti et al., 2017). It is thus to be expected that the V-A scaling that you use to create the initial steady state does not hold for the final one. This is something that would need to be explored in more detail, and for which studies in which the volume scaling also uses information from other glacier characteristics (e.g. the glacier slope) could be useful (Grinsted, 2013; Zekollari & Huybrechts, 2015; see e.g. Fig. 9a in the latter, which summarizes the main point made here).

1. It is inadequate to say the only problem with scaling model is that they underestimate climate sensitivity of volume as the reviewer suggested here. We have demonstrated that, the scaling model,

\* underestimate both area and volume sensitivities,

\* underestimate both volume and area response times, and

\* assume area and volume response time to be equal to each other.

We believe there are no existing prescription that allows correcting all these biases within a scaling model framework.

2. We have acknowledged that a gradual steepening of the shrinking glaciers is a major factor behind the decline in c (L276-282).

However, as we have shown using 1-d flowline models, even for transient glaciers with the same linear bedrock slope similar deviations/biases are seen (supplementary fig. S10). Slope-dependent corrections, therefore, cannot cure all the biases of scaling models.

3. We have argued that a significantly faster area response in scaling models compared to that in the SIA model, lead to a subdued volume response through the feedback of a shrinking ablation zone on the net negative balance (L242-247, and L294-304).

4. As stated clearly in the objective, instead of investigating the extended scaling model, we chose to focussed on a simple linear-response model and establish it as possible alternative which reduces the above biases.

[4] Unclarities in the manuscript. I found the text difficult to follow and quite often had to re-read sentences several times before being able to grasp their meaning. A few examples include:

a. I. 8-9: '..and validate them with results from scaling-based simulation of the ensemble of glacier'

b. I.84-85: '...are then empirically extended in order to obtain accurate parameterisations the linear-response properties of the SIA-simulated glaciers'

c.l.86-87: 'The linear-response model the long-term total shrinkage of glaciers as predicted by the scaling-based method (Radićet al., 2007), and the linear-response model are compared with the corresponding response'd.....etc. See also comments on specific sections below.This makes it tedious to go through the manuscript. Furthermore, there a substantial number of grammatical errors, some of which (but not all) have already been pointed out by the first reviewer.

Also, many figures cannot be interpreted/read independently, without having to refer to the caption. It would be good if all essential information (e.g. meaning of colors used, R^2 values, equations,...etc.) could be directly included in the figure. Some other comments for specific sections(non-exhaustive list and not focusing on grammatical errors)

We have tried to improve the language, clarity, and organisation of the revised manuscript. All the figures have been modified based on the above suggestions.

**o1. Introduction:**

'methods solving the dynamical ice-flow equations' à'numerical cost of such a computation on a global scale is prohibitive': well is not really the case anymore. In general: would be good to acknowledge regional-to global studies in which ice flow is explicitly accounted for (Clarke et al., 2015; Maussion et al., 2019; Zekollari et al., 2019).

o2. Quite abstract and thus very difficult to go through for someone who is not an expert in V-A scaling. Could make it less technical by for instance adding some additional information that links the various parts.

We have referred to these approaches in the revised version.

o1.2. Motivation for the present study: difficult to follow the first paragraph: be more specific when you refer to c and gamma not continuously mix with other terminology 'time invariant scaling-based parameterisation', '...given the known violation of the time-invariant scaling assumption'.

We have defined the scaling models in the beginning as the ones that assume a constant c and  $\gamma$  to describe an ensemble of glaciers interpreting the scaling relation statistically. We clarify that the present paper only considers this statistical interpretation of the scaling relation. We have tried to improve the Introduction section in general.

**o3.1.: 2-dimensional SIA model:**

•I.152-154: where did you get the ice thickness from? From Kraaijenbrink et al. (2017)directly? As the ice thickness is quite crucial in your story (it determines the volume...), why did you not consider the consensus estimate of Farinotti et al. (2019), which is freely available?

This is because Kraaijenbrink et al. (2017) was available when the study was initiated. It also had debris-cover information that we we intended to incorporate. Moreover, given the other idealisations in our ensemble of synthetic glaciers and our stated objectives in the study, any reasonable bed rock is fine. So we stick our original choice of bedrock.

•SIA: refer to the original work by Hutter (1983)also.

•You neglect basal sliding (I. 161). Justification? Could refer to other studies where this is done, like e.g. Gudmundsson(1999) and Clarke et al. (2015).

The available flowline model studies of Himalayan glaciers typically includes sliding. We have not performed a sensitivity analysis. So at the moment, we simply state that we dropped the sliding term for simplicity.

•I.168-178: this is related to the SMB, which you apply in all cases(i.e. also for the linear-response model and the V-A scaling, right?). Not sure this section is correctly placed here in the '2-dimensional SIA model' section.

The mass-balance function is moved out of the section describing SIA model.

•I.183: through several exclusion you keep 68% of the initial glaciers... How much does this represent in terms of glacier volume and glacier area when compared to the total glacier sample?

We have now included 86% glaciers in our study. The distribution of slope, and area of the two sets with 703 and 810 glaciers are shown in the supplementary S8.

•I.188-196: you explain some simplifications related to debris cover, avalanche and sliding have been made and that this may influence your results. Well, you have made much larger simplifications than this: e.g. linear SMB profiles with strongly imposed max. SMB, steady state assumptions for glaciers,... à not even worth mentioning these more detailed simplifications in my opinion. With all these simplifications, would have been better to opt for idealized setup likely (see main comment 2a).

We have already explained why we prefer the present setup over idealised glacier before.

At least for Himalayan glaciers, factors like avalanches and debris cover can be very important.

We have used the standard paradigm of defining system response around steady-states. In fact, most of the knowldge about glacier response is based on response properties of steady-state glaciers. A detailed discussion of this issue is added (L171-180).

We have included 1-d idealised flowline model simulation now to show that our results are not specific to the ensemble of synthetic glaciers used (Supplementary fig. S10).

•1. 194: 'These simplifications do not weaken our study': not sure you can judge on this yourself...

**oSection 3.2.:**

•1.204: 'was fixed at... because...': don't understand the causality (i.e. link between cause and consequence).

•Figure 1: SIA-derived volumes are scaled by a factor 10: why? Does not really make sense and unclear when just looking at the figure without reading the caption... Axes should be correct in the figure and not only for a part of the data you show..Also illustrates the unclarity in the figures mentioned in main comment 4 (problem that figures cannot be interpreted without referring to their caption).

We have made appropriate changes to address these concerns.

**OSection 3.3.:**

•1.208-210: complicated way to say that you consider e-folding time scales. Would reformulate this and add references for thistoe.g. Leysinger Vieli & Gudmundsson(2004).

We do not use e-folding time, but fit the linear response for directly. These two methods would give the same result for a purely exponential response. Our approach may be more suited for calibrating the linear-response model, as it minimises the RMSE between the linear-response and SIA model outputs. (L171-180)

As shown above, the differences between our best-fit response time (vertical axis) and the corresponding e-folding time (horizontal axis) is less than a couple of per cent on the average. Note that in the above plots the 1:1 lines (black).

oSection 4.1.:

•1.225: V=cA1.286: not sure I understand. Does this statement apply for the initial and/or final steady state volumes? And can all the volumes be described with this single relationship? Is the fact that quite different rate factors are used not a problem for this (see main comment, point 2c)?

As explained in L263-268, the set of 703 glaciers at any instant (intial, final, steady or nonsteady) follow this relation statistically with different best-fit c. The same is confirmed in fig 1a. This is consistent with the theory of Bahr et al. (2015).

•1.227 + 1.230 + 1.232: here you mention that c is time-dependent. Not sure you can say that it is time dependent: simply results from the fact that final steady state volume for V-A scaling is 'overestimated' (vs. SIA). As a result the evolution to this steady state is different. See main comment 3c for this.

It is clear from figure 1 that c is time-dependent. For example, the best-fit c has two different values at t=0 and t=500 years. We have already replied the other comment before.

•1.235-237: relates to main comment 3c again. If you do not modify the V-A scaling, then problems will arise when considering the same glacier that is much smallerin a warmer climate (when rising the ELA in your case): you typically lose the lower parts where most volume is and volume will thus be 'overestimated'. Is this not accounted for in some way in future glacier evolutions based on V-A scaling? As a part of this discussion, studies in which V-A scaling is extended with other glacier characteristics (such as the surface slope; Grinsted, 2013; Zekollari & Huybrechts, 2015)would be good to include. Such relationshipswhich could prove to remain valid over time, even without changing scaling and exponents.

We agree with the reviewer that scaling model description can be improved by incorporating the slope-dependence of c. We now acknowledged that a major part of the time dependence of c is due to steepening of glacier slope with time (L276-283). However, a slope-dependent correction is unlikely to get rid of all the scaling model biases pointed out in this study. For example, as shown in our idealised flowline model results (supplementary fig. S8), similar biases can be present even for a set of glaciers with the same slope.

We have now emphasised that scaling models implicitly assume,

 $\tau_{A} = \tau_{V}$ .

However, the area-response time is longer in reality (SIA). This limitation would introduce complex time-dependence in the relationship between A and V for real glaciers (Fig 1b). We have now argued that a faster area loss in the early stages of the response as simulated by scaling models, provides a feedback to the net negative balance, and leads to a subdued long-term volume loss (L242-247 and L294-305)

OSection 4.2.: •1. 243: 'This is exactly what is seen in Fig. 2b, which shows...': I cannot directly see this...

•1.244: 'change in c to the tune of  $\sim$ 13%': what does this mean?

•1.255: 'The above figure': will depend where your figure comes in final manuscript...

oSection 4.3:

•Iwas wondering what the pointis thatyou want to make with this section? It is known from literature that volume responds faster than area (e.g. Oerlemans, 2001; Leysinger Vieli & Gudmundsson, 2004).

•1.260-264: relationship between volume and area response times. How does this compare to the relationship others have found in the literature?

We have updated the text based on these comments, and added necessary discussions and references (L294-298).

OSection 4.4.:

•1.271-272: 'with most of the changes taking place during the first couple of centuries': this is not a result/finding.. This directly results from the e-folding time-scale when forcing a steady state glacier with an instantaneous forcing in SMB.

•1.273: 'underestimates the long-term change': not about reaction/response. This is direct consequence of fact that final steady state volume is too large (see main comment 3c)

•1.279-280: '...suggests that there might be significant negative biases of mountain glacier contribution to sea-level rise as computed by scaling-based methods'(+ section 4.5, 1.300-302): well, do not see this in GlacierMIP phase 1+2... Is a very strong statement to make and should be sure that it is well-founded.

We reworded the above statements, and added discussions referring to the trends seen in Hock et al., (2019) in support of the above claim. (L334-349)

**oSection 4.5:**

•1.296-297: 'More detailed studies that relaxes some of the above mentioned assumptions are needed...': not sure what you mean by this. Would also make sense that you dig into this: e.g. by focusing on real transient response vs. comparing two steady states (what you do now and then translate into an analysis of the transient response resulting from this: see main comment 3c).

We have added a section on how the steady-state response properties can be used to obtain transient response (section 3.4) to clarify the issue.

•1.299: 'intruding more scatter in the fits': what does this mean? The sentence is removed.

oSummary and Conclusions:

•1.309-310: scale factor reduces over time. Well, not sure the time dimension is adequate here. Boils down to having a final steady state that would require a smaller value for c: see main comment 3c.

If initial and final states have different best-fit c, then that implies c is time dependent.

We agree that having a smaller c and having a thinner final state are equivalent – it is not possible to assign one as the cause and the other as the effect. We have now based our explanation on the theoretical and numerical results showing that the scaling models assume the area and volume response to be equal. In contrast, within SIA, area response time is  $\sim 1.5$  times larger than the corresponding volume response time. This lead to a faster initial area loss in scaling which reduces net volume loss due to a feedback on net negative balance.

•1.324: computational efficiency. OK, still important, but is not really a limitation anymore, due to which V-A scaling becomes less important (and also driven by the release of new datasets with regional-to global spatial coverage at individual glacier level: see main comment 1).

Given that most, if not all, available model estimates for glacier contribution to sea-level rise are based on low-dimensional models, computational efficiency may be an important factor.

•Code availability: for which models is the code available? Seems to suggest that the SIA code is not available. Not sure if this fully agrees with the policies of The Cryosphere: see www.the-cryosphere.net/about/data\_policy.html

We shall make all codes available upon possible publication of the manuscript.

**Possible biases in scaling-based estimates of mountain-glacier contribution to the sea levelsea-level rise**

Argha Banerjee1, Disha Patil1, and Ajinkya Jadhav1 1ECS, IISER Pune, India

Correspondence: Argha Banerjee (argha@iiserpune.ac.in)

Abstract. Predicting mountain-glacier contribution Low-complexity glacier models are used to compute the contribution of mountain glaciers to sea-level rise involves computing global-seale glacier loss under a given given a climate-change scenario. Such calculations are usually done with low-complexity and computationally-efficient approximate models of glacier dynamics. A statistical power-law relation A majority of these models are based on statistical scaling relations between glacier

- 5 volumeand area(, area, and/or length) is the basis of several such models. We simulate transient response of an ensemble of 551 glaciers from Ganga basin, the Himalaya, using a scaling-based method and a two-dimensional ice-dynamical model based on . In this paper, the response properties of glaciers are theoretically analysed within a time-independent volume-area scaling assumption. The theoretical results are validated with a scaling model simulation of the response of 703 synthetic Himalayan glaciers from the Ganga basin to a step-change in climate. The same numerical experiment repeated with a 2-d
- 10 shallow-ice approximation (SIA) . A comparison of the model outputs suggests that the scaling-based method systematically underestimates model, obtains about three times larger climate sensitivity and response time than that predicted by the scaling model. There is a corresponding low bias in the scaling model estimates of the long-term ice loss due to a violation of the assumed time-invariant scaling. We derive expressions for the response time and climate sensitivity of glaciers simulated using a-loss of the total glacier area and volume. Also, the scaling model predicts the area and volume response times to equal
- 15 to each other, while the SIA model obtains area response time that is about 1.5 times larger than the corresponding volume response time. Consequently, the transient glaciers simulated with SIA exhibit a systematic violation of time-invariant scalingassumption, and validate them with results from the scaling-based simulation of the ensemble of glacier. These expressions are modified empirically to obtain similar parameterisations of the response properties of glacierssimulated with SIA. These new parameterisation yields. The SIA results are used to obtain parameterisations of climate sensitivity and response time
- 20 of glaciers, leading to a linear-response model which significantly reduces the above biases, while retaining the advantage of numerical efficiency outperforms the scaling model in reproducing the SIA results. This is confirmed by an experiment on an independent set of 204 glaciers from the Western Himalaya. This linear-response model may be useful for predicting the sea-level contribution from shrinking mountain glaciers.

**1 Introduction**

25 Shrinking mountain glaciers have contributed significantly to the global eustatic sea-level rise in the recent past, and this trend is expected to continue for the next hundred years or so (Meier, 1984; van de Wal and Wild, 2001; Raper and Braithwaite, 2006; Cogley, 2009; The reliability of the predicted global sea-level change is, thus, intimately tied to the accuracy of the predicted total ice-loss from mountain glaciersglobally.

Instantaneous (annual) glacier surface mass balance can be calculated readily using data from climate model simulations.

- 30 However, an accurate climate model outputs. In contrast, any prediction of the long-term evolution of a glacier would require simulating the decadal-scale requires simulating the slow (decadal) changes in glacier area and hypsometry (Raper and Braithwaite, 2006; F ideally geometry. Ideally, this is to be done by solving the dynamical ice-flow equations (Oerlemans, 2001; Cuffey and Patterson, 2010)(e.g However, the numerical cost of such a computation on a global scale is prohibitivecreates a bottleneck, even if a simplified approximate description of the full simplified approximate descriptions of the ice-flow equationslike, like, shallow-ice approx-
- 35 imation (SIA) (Hutter, 1983) or its higher order variants were to be used Further(Egholm et al., 2011; Clarke et al., 2015). One-dimensional SIA-based modelling tools are promising developments in this regard (Maussion et al., 2019; Zekollari et al., 2019; Roun The uncertainties associated with various input parameters, e.g., an uncertain glacier bedrocklimits-, limit the benefit of using the physically-based ice-flow models (Farinotti et al., 2016). The as well (Farinotti et al., 2016).

Due to the above difficulties, the existing global-scale estimates of the mountain-glacier contribution contributions of

- 40 shrinking mountain-glaciers to sea-level rise mostly rely on low-dimensional approximate parameterisations of the glacier dynamics (van de Wal and Wild, 2001; Raper and Braithwaite, 2006; Hirabayashi et al., 2010; Radié and Hock, 2011; Slangen and van de Several-glacier dynamics (Radić et al., 2014). The results from these simplified models provide critical inputs for assessing regional to global vulnerability to sea-level rise (e.g., Kulp and Strauss, 2019). While some of these parameterisations are based on an statistical area-volume (or area-volume-length) scaling relation for any set of mountain glaciers (Chen and Ohmura, 1990; Bahr et al.,
- 45 Empirical prescriptions for distributing the annual ice-loss over glacier surface, or equivalently, empirical prescriptions for adjusting the hypsometry of the transient glaciers are also used (Raper and Braithwaite, 2006; Radić et al., 2008; Hirabayashi et al., 2010; Hu

**1.1 Area-volume scaling for mountain glaciers**

The statistical power-law relationship between glacier area and volume was established empirically (e.g., Chen and Ohmura, 1990),

50

**$V = cA^{\gamma},$**

where, (Raper and Braithwaite, 2006; Huss et al., 2010; Huss and Hock, 2015), a majority of them are primarily based on a statistical volume-area (or volume-area-length) scaling relation. This volume-area scaling equation relates glacier volume V and to glacier area A are glacier area (km2) and volume (km3), respectively. as

55  $V = cA^{\gamma}$ ,

(1)

[revised manuscript text omitted]

165
$$\underline{\dot{A}} \equiv \frac{\delta b}{\gamma h} A$$

$$\equiv \quad \frac{\delta b}{\gamma c} A^{2-\gamma}.$$

Thus, a consequence of the time-invariant scaling hypothesis is that the rate of area change must scale with glacier area with an exponent (2 – γ). This is consistent with empirical observations (Banerjee and Kumari, 2019). As the scale factor of this power-law relation is proportional to the mean mass balance, Eq. 5 may be a convenient way of obtaining mean regional
thinning rates from relatively straightforward remote-sensing measurements of the rate of area change. However, this relation is accurate only to the extent the assumption of the time-independence of the scale-factor *c* is valid. scaling models of glacier evolution (e.g., Radić et al., 2007). We have derived analytical expressions for glacier response time and climate sensitivity starting from this equation, essentially following the line of arguments by Harrison et al. (2001).

**2.2 The area response time**

175 As response time is defined with respect to a steady-state glacier, let us consider a steady glacier and apply a constant perturbation for time t > 0 (i.e. a step change in ELA ). Let's take the corresponding instantaneous net negative balance of the glacier at t = 0 to be  $\delta b_0 A$ . The perturbation would asymptotically  $(t \to \infty)$  lead to a shrinkage of glacier area by  $\Delta A_{\infty} = A(0) - A(t \to \infty)$  and ice volume by  $\Delta V_{\infty}$  such that (Harrison et al., 2001),

$$\Delta A_{\infty}b_t + \beta \Delta V_{\infty} \approx -\delta b_0 A.$$

**180 Here, $b_t$ is the ablation rate near the terminus and $\beta$ is the balance gradient. The area response time of the glacier can be estimated as $\tau_A \approx \Delta A_{\infty}/\dot{A}$ . Therefore, using the expressions for $\dot{A}$ (Eq. 5)**

**2.2 Numerical methods**

We simulated the response of an ensemble of synthetic clean glaciers with realistic geometries to a hypothetical step-change in ELA using three different methods (scaling, SIA, and  $\Delta A_{\infty}$  (Eq. 7), respectively, we obtain,

(

$$\underline{\gamma h} + \beta)^{-1} = \tau^*$$

Here, the symbol  $\tau^*$  is a convenient shorthand notation for the time scale  $({}^{b_t}\gamma h + \beta)^{-1}$ . In the above derivation,  $\Delta V_{\infty}$  that appears in eq. 7 is eliminated with the help of eq. 2. The resultant expression for response time (Eq. 8 is comparable with that derived here there is a comparable of the time is a comparable of the time scale of the time is a comparable of the time scale of tim

[revised manuscript text omitted]

$$\underline{\gamma h} + \underline{\beta})^{-1} \equiv \tau^*.$$

(8)

400 Here, the symbol  $\tau^*$  is a convenient shorthand notation for the time scale  $-(\frac{b_t}{2}\gamma h + \beta)^{-1}$ . In the above derivation,  $\Delta V_{\infty}$  that appears in eq. 7 is eliminated with the help of eq. 2. Eq. 8 is comparable with the expression of area response time as given by Harrison et al. (2001), or Lüthi (2009).

**3.1.3 Volume response time**

The instantaneous change in volume ( $\Delta V(t)$ ) for a steady glacier perturbed by a small step change in ELA at t = 0 and t = 500405 years. The corresponding linear fits in log-log scale have  $R^2$  value of 0.90 is given by,

$$\Delta V(t) = \Delta V_{\infty} (1 - e^{-t/\tau_v}), \tag{9}$$

where,  $\tau_v$  is the volume response time and  $\Delta V_{\infty}$  is the volume sensitivity (e.g., Lüthi, 2009). Now, V(t), V(0), and  $V(t \to \infty)$ appearing in eq. 9 can be expressed in terms of A(t), A(0), and  $A(t \to \infty)$ , respectively, with the help of corresponding scaling relations (eq. 1). This, in the limit of a small fractional changes in area, yields,

410
$$\Delta A(t) = \Delta A_{\infty}(1 - e^{-t/\tau_v}).$$
(10)

Comparing the above two equations, and using eq. 8 one obtains,

$$\tau_A = \tau_V = \tau^*. \tag{11}$$

This implies that all scaling models implicitly assume the area and volume response times of a glacier to be equal to each other. However, it is known that for mountain glaciers area response time is larger than the volume response time within a

- 415 SIA model (Oerlemans, 2001; Vieli and Gudmundsson, 2004). Therefore, the assumed equality of the two response times in scaling models (eq. 11) contradicts the existing SIA results. This is an intrinsic bias that is present in any scaling model. Another interesting trend that is evident from figAfter a step change in ELA, as the ablation zone shrinks, the initial net negative balance of a glacier gradually decays to zero over a period determined by the corresponding response time. A longer area response time in SIA implies that this reduction in the ablation zone is slower here than that in a scaling model. A
- 420 corresponding feedback of a larger ablation zone on the net mass balance should then lead to a higher long-term volume loss in a SIA model than that in a scaling model. This indicates the possibility of a low bias in scaling model estimates of the climate sensitivity of volume, or equivalently, that in the long-term changes in glacier volume due to any rise in ELA.

**3.1.4 Climate sensitivity of area and volume**

An expression for the climate sensitivity of glacier area ( $\Delta A_{\infty}$ ), which is the asymptotic change in area due a change in ELA 425 by  $\delta E$ , is obtained by eliminating  $\Delta V_{\infty}$  from eq. 1a is that the 7 using eq. 2.

**$\underbrace{\tau^*\beta\delta E\gamma h}_{\sim\!\sim\!\sim\!\sim}\underbrace{\alpha^*}_{\sim\!\sim\!\sim}.$**

(12)

430 Here, we have used the definition of  $\tau^*$  (Eq. 8), and that  $\delta b_0 \approx \beta \delta E$  for a step change in ELA by  $\delta E$ . The RHS of the above equation is denoted by  $\alpha^*$  for convenience.

The corresponding expression for  $\frac{\Delta V_{\infty}}{K}$  is then obtained using Eq. 2,

$$\frac{\Delta V_{\infty}}{V} = \gamma \alpha^*. \tag{13}$$

Again, Eq. 13 is comparable to the expression of volume sensitivity as derived by (Harrison et al., 2001), where the authors used an arbitrary thickness scale H, instead of the denominator of  $\gamma h$  appearing in the definition of  $\alpha^*$  above.

Please note that strictly speaking, the climate sensitivity of area and volume with respect to a change in ELA should be defined as  $\frac{\Delta A_{\infty}}{\delta E}$  and  $\frac{\Delta V_{\infty}}{\delta E}$ , respectively. However, in this paper, we use  $\Delta A_{\infty}$  and  $\Delta V_{\infty}$  as the corresponding sensitivities to simplify the notation.

---

## Referee Report (RR1)

**Second round of review of 'Possible biases in scaling-based estimates of mountain-glacier contribution to the sea level' by Banerjee et al.**

Submitted to 'The Cryosphere Discussions', discussion started on 9 January 2020
Second round of review: May 2020

Banerjee and colleagues have put a substantial effort in updating their manuscript in order to answer the issues raised by both reviewers. Through this, they have addressed some issues raised (e.g. more clarity on mass conservation, some unclear statements were removed and clarified). However, some important foundations of the story are still problematic.

- Scaling methods are losing significance now that methods arise in which the glacier geometry is explicitly accounted for. The authors state that they are not aware of any studies in which sea-level contribution is calculated based on flow models. OGGM was applied globally (Maussion et al., 2019), and was used to project the future contribution to sea level from all glaciers (Marzeion et al., 2020).
- The argument that an idealized setup is not useful to compare various methods does not hold in my opinion. To make claims about the suitability of scaling-based methods for sea level contributions based on Himalayan glaciers (or any mountain glaciers) does not really make sense. When it comes to sea-level contribution studies, one should focus on ice caps and big Arctic glaciers (i.e. not mountain glaciers), which contain the almost entirety of the worldwide glacier volume (see e.g. Table 1 in Farinotti et al., 2019). i.e. you investigate the effect of scaling-based methods on the future evolution of mountain glaciers: 'sea level' is out of the context here.
- In their answers Banerjee and colleagues suggest that there is a clear evidence of underestimation of glaciers relying on V-A scaling arguments in GlacierMIP (Hock et al., 2019). This is not very clear, and moreover, the problem with GlacierMIP is that the setup was very different for the various models (i.e. it is difficult to compare a V-A scaling and another type of model if the forcing is totally different). The good news is that this has been partly been solved in the second phase of GlacierMIP (Marzeion et al., 2020). Another advantage of this new study is that there are more models to compare and that the comparisons can also be made at the regional level. All the data is freely available. The authors would have to look into this, but I'm afraid that also here there is no clear sign of V-A scaling based methods vs. others.
- Some parts remain vague. It is for instance unclear how your algorithm failed for many glaciers, and now that you have performed an 'update' of your algorithm this is solved…
- It remains problematic to see that the model was not calibrated for individual glaciers. The argument 'to avoid the associated computational cost' is not a very solid one… Such models are computationally cheap to run and given the relatively limited sample of glaciers considered (within the framework of regional- to global studies), this should not be a problem. By having a realistic geometry, the velocities will automatically also be relatively close to the observations (and the argument 'Tuning the rate factor to fit the thickness may not be a good idea, as it may lead to unrealistically small glacier velocities, and thus, unrealistic response properties' does therefore not hold).
- The conclusions are in the end still based on a comparison of the evolution between steady states (which boils down to comparing steady states). The lack of real transient analyses makes it difficult to support any claims related to validity of transient models (which are used for sea level rise studies).

In conclusion, I am still not convinced by the statement that V-A scaling methods are likely to underestimate the future sea level contribution from glaciers. And even with a good modelling setup and clear presentation of your results, I do not think that any conclusions on sea level contribution validity of different models can be obtained from a study on Himalayan glaciers (or any other mountain glaciers, given the limited total volume of ice stored in these ice bodies).

**References**

Farinotti, D., Huss, M., Fürst, J. J., Landmann, J., Machguth, H., Maussion, F., & Pandit, A. (2019). A consensus estimate for the ice thickness distribution of all glaciers on Earth. *Nature Geoscience*. https://doi.org/10.1038/s41561-019-0300-3

Hock, R., Bliss, A., Marzeion, B., Giesen, R. H., Hirabayashi, Y., Huss, M., et al. (2019). GlacierMIP – A model intercomparison of global-scale glacier mass-balance models and projections. *Journal of Glaciology*. https://doi.org/10.1017/jog.2019.22

Marzeion, B., Hock, R., Anderson, B., Bliss, A., Champollion, N., Fujita, K., et al. (2020). Partitioning the Uncertainty of Ensemble Projections of Global Glacier Mass Change. *Earth's Future*, in press. https://doi.org/10.1029/2019EF001470

Maussion, F., Butenko, A., Champollion, N., Dusch, M., Eis, J., Fourteau, K., et al. (2019). The Open Global Glacier Model ( OGGM ) v1.1. *Geoscientific Model Development*, *12*, 909–931. https://doi.org/10.5194/gmd-12-909-2019

---

## Author Response (AR2)

Dear Dr Radic,

Thank you for providing us with an opportunity to submit a revised version of the manuscipt. We have modified our manuscript based on the referees' comments, and the changes are as follows.

1. We have provided clear description of the methodology adopted for the linear-response model simluation of the western Himalayan glaciers.

2. The discussion of the intercomparison studies have been expanded which now include reference to Marzeion et al. (2020) as well. We have acknowldeged the future potential of flow-based models while discussing the motivation behind the present work.

In the present version, we have corrected a couple of errors in eq. 16 and 18, and fixed a bug that was present in the scaling model code. We have updated the figures and text describing linear-response and scaling model results accordingly. Please note that updated results did not lead to any changes in the basic results, disccussions, and conclusions.

Also, while we had simulated 204 glaciers from the western Himalaya, results from only 164 were used for the comparison shown in fig S9. The remaining glaciers were not considered due to more 50% long-term changes taking place on these glaciers. We have corrected the text and captions accordingly.

Please note that apart from the experiments with step-changes in ELA to study the glacier response properties, we have also done a comparison of the three models for continuous linear changes in ELA as disccussed later in this document. This was in response to one of the comments by the anonymous referee. The biases of the scaling model and the improved performance of the linear response model can be seen in these experiments as well.

We look forward to your response on the revised manuscript.

Best regards,
Argha Banerjee

**Reply to the comments by editor Valentina Radic**

Your revised manuscript received two reviews. While the referees acknowledge the efforts you have put into the revisions they still remain critical about the study. Please see their responses below. I am willing to provide you a chance to submit your responses to these comments and revise the manuscript accordingly. In particular, pay attention to the referees' comments regarding the (1) usefulness of V-A scaling in the light of existing more advanced approaches, and (2) the clarity on how the parameters in the V-A scaling are calibrated over the Himalayan glaciers. It is likely that your revised manuscript, once received, will be sent for another round of reviews.

We thank the editor for providing us with an opportunity to submit a revised manuscipt. We have modified our manuscript based on the referees' comments to the best of our ability.

Replying to the comment 1) by the editor, we would like to argue that the present study, which deals with glacier response properties within scaling theory and possible biases of corresponding long-term glacier change estimates, is relevant due to the folllowing reasons.

There are numerous examples of coexistence of 'simple' and 'adavnaced' models in most branches of physical sciences, including in glaciology. For example, all the ice-flow models in the Marzeion et al (2020) uses simple temperature-index models for mass-balance computation, and not more advanced energy-balance models that have been there for decades. We are confident that the low-complexity glacier-evolution models would remain in use in the future, and in that context our results are relevant and valuable.

The reliance on multi-model ensemble averages imply that a variety of glacier model would continue to be used even as flowline models capable of simulating glaciers on a global scale are available. The scaling models are lilkely to contribute significantly to such multi model averages in the future. As we have pointed out in the text, about 2/3 and 1/2 of all the models used in latest sea-level rise estimates by Hock et al (2019) and Marzeion et al (2020), respectively, were scaling models. They are also being used for hydrological simulations. This makes it important to develope and critical investigate of low-complexity models, rather than focusing exclusively on SIA models.

There are two other contributions presented in the paper that may be important. First, the linear-response model presented here is an accurate low-complexity alternative to model glacier evolution. As such, this may be an useful addition to the ensemble of low-complexity models used for computing sea-leve rise. Second, our systematic derivation of the response properties of glaciers under a scaling assumption, and parameterisations for SIA derived response properties, we believe, are useful additions to the exisitng literature on glacier response properties.

In response to the comment 2) by the editor, we have now included additional discussions about the calibration of the linear-response model using SIA results for the central Himalayan glaciers, the application of the model to simulate western Himalayan glaciers, and possible application of the linear-response model to other glacierised regions. Hope this has led to improved clarity of our presentation of the linear-response model. We thank referee Eviator Bach for pointing this out.

Our detailed (in **black**) to the comments by the referees (in **red**), along with details of the corresponding changes in the manuscript are given below.

**Reply to the comments by referee Eviator Bach**

However, I am still unclear on two issues, which I believe have not been adequately addressed from the last revision:

1. How is the linear response model applied in projecting the volume loss? In the paper, the authors write that "For each of the 703 glaciers, the time series of volume and area as obtained using the SIA and scaling models, were separately fitted to linear-response forms (e.g., eq. 9 below) to obtain the corresponding best-fit values of the four linear-response parameters (the climate sensitivities and the response times of area and volume) for each of them". If the linear-response model is fit individually to each glacier, how was it then applied to the "204 glaciers from the western Himalaya without any further calibration"? Was some sort of average chosen for the linear-response parameters?

We apologise for the confusion. We have now tried to clarify the procedure as follows.

L194
"The best-fit empirical parameterisations for climate sensitivity and response time obtained by fitting the SIA results as described above (given in eqs. 14−17 later), were used to run a linear-response model simulation for any given glacier. This model was applied to simulate the response of the above 703 synthetic Himalayan glaciers to a 50 m step-change in ELA at t = 0. We emphasise that for the linear-response model, we do not use the best-fit the response properties of the individual glacier derived from the SIA simulations. Rather, the parameterisations of the same obtained by fitting the SIA-derived response properties (given in eqs. 14−17 later) were utilised. These parameterisations thus allow the model to be applied to any other set of Himalayan glaciers without the need for simulating them with SIA first."

L 204
"To test the applicability of the above linear-response model that was calibrated using SIA results for the 703 central Himalayan glaciers, the same model was applied to a different set of 204 glaciers from the western Himalaya. The parameterisations developed for the central Himalayan glaciers as discussed above (given in eqs. 14−17 later) were used to estimate the response properties of each of these western Himalayan glaciers using input values of corresponding mass-balance gradient, mean thickness and ablation rate near the terminus. For these western Himalayan glaciers, SIA and scaling model simulations were also performed following the procedures as detailed above."

L374
"To confirm the improved performance of the linear-response model compared to that of the scaling mode we applied both the models to simulate a different set of 164 glaciers in the western Himalaya (supplementary fig. S1). The best-fit linear-response properties obtained from SIA simulation of the 703 central Himlayan glaciers were first fitted to obtain four equations (eqs. 14−17) that relates the response properties to $\beta$, $\gamma$, h and $b_t$  as described before. The same equations were used to estimate response properties of each of the 164 western Himalayan glaciers as required for the linear-response model simulations."

2. How you are proposing to apply the linear-response model for a real glacier? Are you suggesting that, for each glacier, an SIA model should be run to generate a time-series, fit the linear-response parameters, and then use the linear-response model to project into the future? If so, what is the advantage of this approach over just running the SIA model into the future for each glacier? This

should be made explicit.

The parameterisation provided in eqs. 14-17 can be used to estimate the response properties of any glacier as long as estimates of mean thickness, balance gradient, gamma, and ablation rate near the terminus are available - Just as we have demonstrated for the western Himalayan glaciers. However, since the above paramterisations were derived for conditions typical of Himalayan glaciers, it may be necessary to test the parameterisations by SIA simulation of a few tens of glaciers spanning a wide range of area and slope values, before applying it to any other region. We have now discussed this in the manuscript (L422):

"Since the above parameterisation of linear-response perperties (eqs. 14−17) are derived from SIA simulations of an ensemble of Himalayan glaciers, when applying them to any other glacierised region in world, it may be necessary to simulate a few tens of glaciers (having a representative range of area and slope) from that region using SIA first, and confirm the accuracy of the above parameterisations."

Typographical issues:
– Bach et al. is cited as 2019 in the text, but it is from 2018
– paramterisations -> parameterisations

We have corrected these errors.

**Reply to comments by anonymous reviewer**

Banerjee and colleagues have put a substantial effort in updating their manuscript in order to answer the issues raised by both reviewers. Through this, they have addressed some issues raised (e.g. more clarity on mass conservation, some unclear statements were removed and clarified). However, some important foundations of the story are still problematic.

We thank the reviewer for the kind comments. Our replies to his/her specific criticisms are given below.

Scaling methods are losing significance now that methods arise in which the glacier geometry is explicitly accounted for. The authors state that they are not aware of any studies in which sea-level contribution is calculated based on flow models. OGGM was applied globally (Maussion et al., 2019), and was used to project the future contribution to sea level from all glaciers (Marzeion et al., 2020). The argument that an idealized setup is not useful to compare various methods does not hold in my opinion.

Despite the advent flow-based models, the scaling models continue contribute significantly to the multi-model mean. The recent intercomparison studies by Hock et al (2019) and Marzeion et al (2020), report multimodel ensembles where 45 - 66% members were scaling-based models. Thus, it is important to probe the scaling model assumptions, and possible prediction biases in such models. We have refered to the flowline model studies appropriately now. Please refer to our replies to the comment (1) by the editor above for our detailed reponse on this issue.

We never claimed that idealised setup is not useful for model comparisons. They certainly are, and that is why we have used them in this study to confirm the general nature of our results based on the earlier suggestions by the reviewer (please refer to the highly ideliased 1-d setup presented in supplementary fig. S10). However, an idealised setup is inadequate to obtain parameterisation of response properties (e.g., eqs. 14-17) that can be applied to real glaciers. Our objective of establishing a linear-response model necessitates the realistic geometries as explained before.

To make claims about the suitability of scaling-based
methods for sea level contributions based on Himalayan glaciers (or any mountain
glaciers) does not really make sense. When it comes to sea-level contribution
studies, one should focus on ice caps and big Arctic glaciers (i.e. not mountain
glaciers), which contain the almost entirety of the worldwide glacier volume (see
e.g. Table 1 in Farinotti et al., 2019). i.e. you investigate the effect of scaling-based
methods on the future evolution of mountain glaciers: sea level is out of the
context here.

We beg to disagree with the above point of view. First, the realtive contribution of any set of glaciers to sea-level rise over a given span of time depends on both their climate sensitivity and response time, apart from their volume. Typically glaciers across all size classes are considered in the exisiting sea-level rise computations. Second, all of our discussions are relevant and useful to model any glacier (excluding ice-sheets and ice caps) independent of its size and location, including the ones refered to by the referee.

In their answers Banerjee and colleagues suggest that there is a clear evidence of
underestimation of glaciers relying on V-A scaling arguments in GlacierMIP (Hock
et al., 2019). This is not very clear, and moreover, the problem with GlacierMIP is
that the setup was very different for the various models (i.e. it is difficult to
compare a V-A scaling and another type of model if the forcing is totally different).
The good news is that this has been partly been solved in the second phase of
GlacierMIP (Marzeion et al., 2020). Another advantage of this new study is that
there are more models to compare and that the comparisons can also be made at
the regional level. All the data is freely available. The authors would have to look
into this, but I m afraid that also here there is no clear sign of V-A scaling based
methods vs. Others.

We agree with the referee that it is hard to establish systematic biases among different models unless the models are all applied to the same set of glaciers, with the same initial volume, and forced by the same climate forcing. Though we have done exactly that this in our study of the sythetic glaciers, the strategies in the above intercomparison studies are different. So it is not expected that the above studies would help in concluding about the scaling model biases that we present in out manuscript. We have acknowledged that in the present manuscript (L356-L363).

However, as we have argued in the text, the intercomparison studies referred to above do not rule out the possibility of a systematic bias in scaling models. For example, even as some of the scaling models start with an initial volume that is ~200% larger than that used in GLoGEM model (see the relavant portion of fig3 of Hock et al. (2019) reproduced below), more often than not GloGEM yielded higher estimates of fractional change on a global scale.

[Figure]

Similarly, the percentage changes under three clmiate scenarios presented in Fig 5 and 11 of Hock et al. (2019) showed larger estimated global change wer obtaine with GloGEM model. The trend is particularly clear in figure 11 which compared runs forced with the same GCM outputs.

[Figure]

**Fig. 5.** Projected mass losses by 2100 in percent of the glacier mass in year 2015 for 19 RGI regions from six glacier models using three RCP emission scenarios. Dots mark the multi-GCM means for each glacier model connected by gray bars, and triangles show their arithmetic mean. Regional results are sorted by the glacier models' mean mass loss according to the RCP8.5 scenario. Results are also shown for all regions combined (global), and all regions excluding the Antarctica periphery (A), and excluding the Antarctica and Greenland periphery (A + G). Note that not all glacier models compute all regions or use all three emission scenarios. The data are available in the Supplementary Material.

[Figure]

**Fig. 11.** Projected mass losses by 2100 in percent of the glacier mass in year 2015 for 19 RGI regions and globally excluding Antarctic and Greenland periphery (A + G) based on the four GCMs that were used by all six glaciers models. Results are based on RCP8.5. Dots mark the results for each glacier model connected by gray bars. Regional results are sorted as in Figure 5.

Very similar trends are seen in estimated global-scale fractional glacier loss by 2100 under different climate scenario as given in Marzeion et al. (2020). The relevant subfigures of fig. S17, S18 and S20 reproduced below showed higher changes in GloGEM compared to the scaling-based GLIMB, WAL2001 and RAD2014 models. However, we agree with the referee that at regional scales, the above trend is not always seen.

[Figure]

[Figure]

Overall, the above intercomparison studies at least does not rule out the possiblity of a systematic bias in the scaling model. And as mentioned above, to reach a definite conclusion, the different models should be applied to the same set of glaciers that are initialised with the same volume and hypsometry, and are forced by the same climate data.

Some parts remain vague. It is for instance unclear how your algorithm failed for many glaciers, and now that you have performed an update of your algorithm this is solved...

In the earlier version of the paper the steady-state criterion was that the aboslute fractional changes in mean glacier thickness has to be less than $10^{-6}$. In the present run, we have changed it to a criterion on specific mass balance: any state with an abolute mass balance less than $10^{-4}$ m/yr is assumed to be steady. This reduced the computation time required to find the steady state - particularly for intermediate states with mean ice thickness less than 100 m. This helped in finding the right steady state for most of the glaciers within a reasonable computation time. Please note that tuning the ELA to obtain a steady glacier having extent similar to the present one is the most expensive piece in our computation. For each trial ELA value, the model needs to be run for several hundreds to a few thousand year until the corresponding steady state is reached.

It remains problematic to see that the model was not calibrated for individual glaciers. The argument to avoid the associated computational cost is not a very

solid one... Such models are computationally cheap to run and given the relatively
limited sample of glaciers considered (within the framework of regional- to global
studies), this should not be a problem. By having a realistic geometry, the velocities
will automatically also be relatively close to the observations (and the argument
Tuning the rate factor to fit the thickness may not be a good idea, as it may lead
to unrealistically small glacier velocities, and thus, unrealistic response properties
does therefore not hold).

As we have argued in our previous reply, while a calibration of the SIA model is necessary to
obtain accurate glacier-change predictions for the Himalaya under climate change, it is not needed
to investigate relative biases between models. This, in fact, is consistent with the reviewer's
assertion above about the effectiveness of idealised setup. What we need here is a set of synthetic
glaciers with realistic geometries and other physical properties. Details of the modeled glaciers
provided in the supplementary demonstrate that the ensemble studied here serves that purpose
adequately.

While we agree that realisitc geometry gets the shape of the velocity profile right, it would not get
the corresponding scale right without an appropriately chosen glacier-specific rate factor. And
varying the rate factor would also impact the ice thickness. Therefore, we believe, it is important to
calibrate for both the velocity and thickness data.

The conclusions are in the end still based on a comparison of the evolution
between steady states (which boils down to comparing steady states). The lack of
real transient analyses makes it difficult to support any claims related to validity of
transient models (which are used for sea level rise studies).

To our knowledge, all discussions of response properties of mountain glaciers in the literature are
based on studying transitions from one steady state to another. The trasient response of a system is
assumed to be determined by the response properties that are defined with respect to the initial and
final steady states, which is then calculated using eq. 18 or its analogs. We have simply followed
that standard paradigm here.
However, based on the reviewer's suggestion we have performed some additional transient analysis.
The figures below show the computed transient response of the glaciers in the western Himalaya to
a linear rise in ELA at the rate of 50m per century  (top row), and 10 m per century (bottom row),
resepectively. The ELA change were applied continuously over the simulation period of 500 years.
(As in the experiments discussed in the main text and the supplementary material, for each of these
experiments below we leave out the glaciers with more than 50% change at 500 years mark).

[Figure]

As expected, the outcomes of these transient experiments are consistent with the general results obtained in the manuscript based on an analysis of the step response of steady-state glaciers as simulated by the three different models.

In conclusion, I am still not convinced by the statement that V-A scaling methods are likely to underestimate the future sea level contribution from glaciers. And even with a good modelling setup and clear presentation of your results, I do not think that any conclusions on sea level contribution validity of different models can be obtained from a study on Himalayan glaciers (or any other mountain glaciers, given the limited total volume of ice stored in these ice bodies).'

We have provided the following convincing evidence to support the claim of possible biases in scaling model.

> 1) The theoretical results on response properties of scaling-model glaciers,
> 2) The numerical comparison scaling model resutls with that from a 2-d SIA model simulation for a set of 703 and 164 synthetic Himalayan glaciers responding to step (and linear) change in ELA,
> 3) numerical evidence from a comparison of 1-d (SIA) flowline model and a scaling model wihtin a highly idealised setup,

They all support our conclusions. As we have discussed in the manuscript, and in our replies above, the intercomparison studies may also be consistent with our result. However, it is also true that these intercomparison studies are not ideally suited to make any strong conclusion on this debate due to the differences in initial conditions, climate forcing and so on, between the models.

The past contribution of mountain glaciers to sea-level rise has been estimated to be 30% since 1900, and their contribution to potential sea-level rise is expected to remain significant for this century at least. And of course, our results are not restricted to the central Himalayan glaciers studied, and of relevance to any glaciers (other than Ice-sheets and Ice caps) that follow the volume-area scaling that is discussed in the text.

**References**

[revised manuscript text omitted]

---

## Author Response (AR3)

**Replies** to the editor's **comments**:

Thank you for the revised manuscript and line-by-line response to referees' comments.

I agree with most of your responses to the referee's criticism, especially on the need for simple models.

Thank you for your kind comments. We are particularly glad to know that you share our views on the importance of simple models.

However, I did find that the key criticism of Reviewer #1 still holds: 'I do not think that any conclusions on sea level contribution validity of different models can be obtained from a study on Himalayan glaciers (or any other mountain glaciers, given the limited total volume of ice stored in these ice bodies).' As I understood this comment, it's not about whether your analysis and findings are valid, but whether you should stretch your findings to global scale while your study is on a regional scale or more precisely, on a suite of glaciers. I do think you should 'scale down' your conclusions, especially in the Abstract section. Effectively, you provide the analysis on synthetic glaciers and a subset of Himalayan glaciers. Thus you results (and objective) concerns glacier evolution on a regional scale. I don't see any need to 'sell ' your study as one on glacier contribution to sea level rise as this would imply 'global' scale' modeling application which you did not perform. I suggest to revise the text throughout the manuscript so that you focus on glacier evolution on regional scale rather than on 'sea level rise.' It is fine to mention the potential implications for the sea level rise estimates from melting glaciers (as you currently do in the Conclusions section), but you should not overreach in the objectives, i.e. presenting the study to be what it is not.

Thank you for interpreting the criticism of Reviwer #1 for us. We accept this criticism. Accordingly, we changed the previous emphasis on sea-level rise computation, and suitably modified the title, abstract, introduction, and parts of the discussions.
     Only place where the sea-level rise problem is discussed in some detail now (L355-L366) is in the context of the two intercomparison studies (Hock et al., 2019; Marzeion et al., 2020). We believe this discussion provides highly relevant examples where the biases discussed in the manuscript could be playing a role. So, we have not removed this discussion entirely. However, we have shortened this paragraph compared to the previous version.
     With the above changes, we hope to have addressed this criticism.

In the light of this criticism I also suggest to change the title to: 'Possible biases in scaling-based estimates in modeling the evolution of Himalayan glaciers.'

We have proposed a slightly modified title than you have suggested here too brings out both the general nature of our arguments, and the regional nature of the glaciers studied. The  present title is, "
[revised manuscript text omitted]

---

## Author Response (AR4)

Dear Dr Radic,

Thank you for suggestions. We have incorporated the suggested changes, and performed a careful check to rectify the gramatical/typographical errors present. We hope that the manuscript may now be considered suitable for publication.

Looking forward to your positive reponse.

Regards,

Argha

Reply to editor's comments:

Please revise your manuscript with the the following minor corrections (also check for any typos and incorrect grammar throughout the text - I have only checked the revised parts):

Line 3: change 'from such a scaling-based models' to 'from a scaling-based model'

Line 3: change 'with that' to 'with those'

Line 7: change ' a 2-d SIA model, obtains' to 'the SIA model, yields'

Line 7: change 'time than that predicted' to 'time than those predicted'

Line 10: 'The SIA results yield parameterisations of climate sensitivity and response time of the glaciers in terms of the corresponding ablation rate near the terminus, mass-balance gradient, and mean thickness'. The sentence is a bot convoluted. How about: 'According to the SIA model, the climate sensitivity and glacier response time are best parameterized as functions of ablation rate near the terminus, mass-balance gradient, and mean thickness.'

Line 12: Change 'A linear-response model based on these parameterisations outperforms the scaling model in reproducing glacier response as simulated with SIA.' to 'Using a linear-response model based on these parameterizations, we find that the linear-response model outperforms the scaling model in reproducing the glacier response simulated by the SIA model.'

Line 58: change 'of the evolution mountain glaciers' to 'of glacier evolution'

Line 348: change 'between the SIA and scaling models' to 'between the SIA and the scaling model'

Line 350: change 'Please note that' to 'Note that, '

Line 357: change ' from that obtained here' to 'from these here'

We have accepted all of the above suggestions and updated the manuscript accordingly.

[revised manuscript text omitted]